# Retrieving Vertical Profiles of Cloud Droplet Effective Radius using Multispectral Measurements from MODIS: Examples and Limitations

Andrew J. Buggee<sup>1,2</sup>, Peter Pilewskie<sup>1,2</sup>

<sup>1</sup>Laboratory for Atmospheric and Space Physics, Boulder, 80303, United States of America <sup>2</sup>Atmospheric and Oceanic Sciences, University of Colorado Boulder, Boulder, 80303, United States of America *Correspondence to*: Andrew J. Buggee (andrew.buggee@lasp.colorado.edu)

Abstract. With the coming launch of the Climate Absolute Radiance and Refractivity Earth Observatory (CLARREO) Pathfinder (CPF) comes an opportunity to develop a new retrieval for warm, non-precipitating clouds from spectral reflectance measurements. With continuous coverage across the shortwave spectrum and a factor of 5 to 10 lower radiometric uncertainty than the Moderate Resolution Imaging Spectroradiometer (MODIS), CPF facilitates the retrieval of a vertical profile of droplet size, providing insight into the internal structure of a cloud. Measurements from MODIS coincident with in situ observations provide the foundation for developing an optimal estimation technique. Solution constraints were required to ensure consistency with forward model assumptions. The limited unique information in the MODIS bands used in this analysis resulted in a non-unique solution, with many droplet profiles leading to convergence. Droplet size at cloud bottom is difficult to constrain because visible and shortwave infrared reflectances have an average penetration depth near cloud top. The region of convergence within the solution space decreased along the cloud bottom radius dimension by 1  $\mu m$  when increasing the number of wavelengths used in the retrieval from seven to 35, and by 3.75  $\mu m$  when reducing the total uncertainty from 3% to 1%. The enhanced accuracy and, to a lesser degree, the enhanced spectral sampling provided by CPF measurements are essential to extracting vertically resolved droplet size information from moderately thick, warm clouds.

#### 1 Introduction

Clouds affect Earth's climate in complex, pivotal ways by modulating incoming and outgoing radiation. They affect weather on short time scales and climate on long time scales. In situ cloud measurements provide thermodynamic and microphysical information over small spatial scales, but the cost of scaling these observations daily and globally is prohibitive. Remote sensing of clouds from space provides the means of acquiring regional to global and seasonal to longer-term information on cloud microphysics and the global distribution and evolution of water in the atmosphere. Monitoring cloud properties from space has improved our understanding of the impacts of clouds on Earth's climate, but cloud feedbacks remain a critical challenge to predicting future climate states.

Passive optical remote sensing of clouds uses measured spectral reflectance of solar radiation to retrieve cloud optical depth (the number of photon mean free paths over the vertical geometric depth of a cloud layer) and the photon-penetration weighted cloud effective droplet radius. These cloud optical properties "...are both a consequence of and an expression for the solar radiative transfer characteristics of clouds (Stephens et al., 2019)." Cloud optical depth plays a fundamental role in cloud radiative feedbacks (Stephens, 2005), and cloud reflectivity (Bohren and Clothiaux, 2006). The fraction of incident light absorbed by optically thick warm clouds is proportional to the effective droplet radius over the solar spectrum (Twomey and Bohren, 1980). The effective radius is an important parameter in the study of cloud condensation nuclei (Twomey, 1977), droplet number concentration (Grosvenor et al., 2018) and precipitation (Chen et al., 2007). From cloud optical depth and effective droplet radius, liquid water path (mass of liquid water in a column of air) and droplet number concentration (number of droplets in a unit of volume) can be derived. Liquid water path is related to cloud droplet growth processes and the onset of precipitation (Miller et al., 2016), and has been used to verify the representation of clouds in climate models (Stephens et al., 2019). Droplet number concentration is used as a proxy for cloud condensation nuclei to study the aerosol indirect effect (Feingold et al., 2006).

Scattered solar radiation from clouds has been used to derive effective droplet radius, cloud optical thickness, and cloud phase since the 1960s. Sagan and Pollack (1967) used spectrally varying reflectance measurements to study the clouds of Venus. Hansen and Pollack (1970) applied the same techniques to terrestrial clouds using measurements taken by a shortwave infrared spectrometer on board a high-altitude U-2 plane. Twomey and Seton (1980) expanded on this work by outlining what is now considered the standard method for deriving cloud optical properties with spectral measurements in the visible and shortwave infrared (often referred to as the bispectral method). Throughout the 1980s and 1990s, several methods of reliably determining droplet size and optical depth (Nakajima and King, 1990; Twomey and Cocks, 1982) as well as cloud phase (Pilewskie and Twomey, 1987) from remote measurements were developed. Twomey and Cocks (1989) and Rawlins and Foot (1990) tested the retrieval theory using five and two wavelengths, respectively, from airborne radiometer measurements to retrieve effective radius and optical depth by comparing measurements with computed reflectances. Beginning in the early 2000s, the afternoon constellation of satellites, called the A-Train, put decades worth of research to the test by implementing these retrieval algorithms on a global, daily basis. The Moderate Resolution Imaging Spectroradiometers (MODIS) on the Aqua and Terra satellites have measured scattered solar radiation and emitted terrestrial radiation in discrete spectral bands for over two decades (Platnick et al., 2003). These measurements were used to derive effective cloud droplet radius, cloud optical thickness, cloud phase, liquid water path, and droplet number concentration, for which there now exists an extensive data record.

The bispectral method of cloud optical remote sensing can be applied to measured reflectance in as few as two spectral bands, one at a wavelength where absorption by water is negligible and the other at a wavelength where water weakly absorbs, defined by the product of droplet size and bulk absorption coefficient being much less than unity (Nakajima and King, 1990; Twomey and Cocks, 1982). Reflectances in these two spectral regions are nearly independent from one another, especially for clouds

with an optical thickness greater than about ten; at non-absorbing wavelengths reflectance is proportional to cloud optical thickness, and at wavelengths where liquid water weakly absorbs reflectance is proportional to effective droplet radius. This bispectral method is employed to compute the MODIS Collection 6 cloud products by computing extensive lookup tables of cloud reflectance with varying solar and viewing geometry, effective cloud droplet radius, cloud optical depth and various surface spectral reflectance assumptions (Amarasinghe et al., 2017). Cloud optical depth and effective droplet radius are retrieved by calculating the minimum  $\ell^2$ -norm, the root-sum-square, of the difference between two MODIS spectral measurements of reflectance and the lookup table estimates.

While the bispectral method is straightforward to implement, it assumes that droplet size within the pixel under observation is vertically and horizontally homogenous (Amarasinghe et al., 2017). Theoretical analysis of warm, non-precipitating adiabatic clouds predicts a vertical structure of droplet size that increases from cloud base to cloud top (Yau and Rogers, 1996). Many in situ measurements of warm, non-precipitating clouds have verified this prediction; the opposite behavior has been found in precipitating clouds and clouds containing drizzle (King et al., 2013; Miles et al., 2000; Painemal and Zuidema, 2011). King et al. (2013) suggested that the assumptions within the MODIS cloud products algorithm for warm, non-precipitating clouds may lead to an overestimation of liquid water path by as much as 25%.

The bispectral retrieval method results in a wavelength-dependent effective radius due to the variability of liquid (and ice) water absorption in the shortwave infrared, specifically defined as the region between 1  $\mu m$  and 2.5  $\mu m$  for this study. This was explained by Platnick (2000) who showed that photons at different wavelengths penetrate to different depths within clouds due to the spectral dependence of single scattering albedo. Thus, the retrieved droplet radius represents a weighted average over the vertical extent of the cloud, with the largest weighting occurring at cloud top (Platnick, 2000). Platnick (2000) also performed an information content study showing that the three retrievals of effective radius using three MODIS spectral channels centered at 1.6  $\mu m$ , 2.1  $\mu m$ , and 3.7  $\mu m$  were found to provide only two pieces of information. The reason these three measurements do not provide three unique pieces of information is that the difference between the retrieval at the 1.6  $\mu m$  channel,  $r_{1.6}$ , and the retrieval at the 2.1  $\mu m$  channel,  $r_{2.1}$ , is less than the retrieval uncertainties for each (Platnick, 2000). Platnick (2000) determined that the relative retrieval uncertainty needs to be at most 5% for the three MODIS retrievals,  $r_{1.6}$ ,  $r_{2.1}$ , and  $r_{3.7}$  to provide three unique pieces of information.

85

Following Platnick (2000), several studies were motivated to retrieve droplet profiles leveraging the information available from MODIS measurements. Chang and Li (2002) proposed using MODIS measurements at three shortwave infrared spectral bands to retrieve the vertical dependence of effective droplet radius. Their method assumed a linear relationship between effective droplet radius and cloud depth, and, like MODIS Collection 6, they computed lookup tables of reflectance at each wavelength to retrieve a droplet profile. Subsequent analysis by (Chang and Li, 2003) used MODIS measurements to solve for the effective droplet radius at cloud top and bottom using a pair of shortwave infrared wavelengths. Repeating this for a

different pair of shortwave infrared wavelengths, the authors retrieved a droplet profile by taking an average of the two linear retrievals. The authors concluded that creating lookup tables for more than two wavelengths and at least six free variables was too memory-intensive for practical use with real data (Chang and Li, 2003). Using the method outlined by Chang and Li (2003), Chen et al. (2007) suggested the vertical structure of droplet size can be used to discern between clouds with and without precipitation-sized droplets.

An early example of applying the optimal estimation method to retrieve cloud optical properties was Heidinger (2003), who retrieved effective radius and optical depth using measurements at  $0.63 \mu m$ ,  $1.6 \mu m$ ,  $3.8 \mu m$ ,  $11 \mu m$ , and  $12 \mu m$ . Minnis et al. (2011) also developed an iterative technique to retrieve cloud phase, optical depth, and effective radius using observations from MODIS and the Visible Infrared Imaging Radiometer Suite (VIIRS) to support the Clouds and the Earth's Radiant Energy System (CERES) data products using measurements at  $0.65 \mu m$ ,  $3.8 \mu m$ , and  $11 \mu m$ . Poulsen et al. (2012) developed a multispectral optimal estimation retrieval method named the Oxford-RAL retrieval of Aerosol and Cloud (ORAC), used to retrieve effective radius, optical depth, cloud top pressure, cloud fraction and surface temperature. Sayer et al. (2011) applied the ORAC algorithm to the data record of the Along Track Scanning Radiometers ATSR-2 and AASTR, creating an extensive retrieved cloud properties data set. The ORAC-retrieved effective radius was found to be  $3.8 \mu m$  smaller, on average, than the bispectral retrieval using MODIS measurements (Sayer et al., 2011).

Kokhanovsky and Rozanov (2012) outlined the mathematical framework for applying an optimal estimation technique to infer a vertical droplet profile using spectral measurements. They showed that four MODIS wavelengths could be used simultaneously with less computational cost than the lookup table method to solve for three variables: the effective radii at cloud top and cloud bottom and cloud optical depth. The authors demonstrated their method with synthetic and real MODIS measurements. Coddington et al. (2012) computed the gain of Shannon information content with respect to the retrieval of effective droplet radius and cloud optical depth using hundreds of measurements across the solar spectrum. The authors found that beyond the traditional method of using two wavelengths, there is additional information within 100 spectral measurements that can meaningfully alter the retrieval of droplet size and optical depth. King and Vaughan (2012) applied an optimal estimation technique to hundreds of synthetic spectral measurements throughout the visible and shortwave infrared. The use of synthetic data enabled a systematic study of the impact of measurement uncertainty on the retrieval uncertainty of cloud optical depth and the effective radii at cloud top and cloud bottom. King and Vaughan (2012) concluded that a measurement uncertainty of 1% would result in a retrieval uncertainty of less than 2  $\mu$ m for the effective radius at cloud bottom and less than 0.1  $\mu$ m at cloud top. It's important to note that this result depends on cloud optical depth (King and Vaughan, 2012). For the retrieved radius at cloud bottom, the authors found the minimum retrieval uncertainty for an optical depth of 10 (King and Vaughan, 2012).

The Climate Absolute Radiance and Refractivity Earth Observatory (CLARREO) Pathfinder (CPF) is an upcoming spaceborne hyperspectral imaging spectrometer that will deploy on the International Space Station, which occupies a near-circular orbit about 400 km above the Earth with an inclination of 51.6° (Shea et al., 2020). The CPF Hyperspectral Imager for Climate Science (HySICS) will make measurements of scattered radiation contiguously from 350 nm to 2300 nm with a spectral sampling and resolution of 3 nm and 6 nm, respectively (Shea et al., 2020). HySICS radiometric uncertainty is 0.3%, and its nadir spatial resolution is 0.5 km after three pixel binning. The full swath width will be 70 km, comprised of 480 measurement pixels (Shea et al., 2020). We have developed new methods that utilize the enhanced radiometric accuracy and spectral resolution of CPF to retrieve vertical profiles of cloud droplet size. The research herein builds upon previous studies in several ways. First, we developed an optimal estimation technique that constrains the set of possible solutions by maintaining a retrieved droplet profile consistent with the forward model assumptions. Second, we apply this optimal estimation method to MODIS data coincident in time and space with in situ measurements from the Variability of the American Monsoon Systems Ocean-Cloud-Atmosphere-Land Study Regional Experiment (VOCALS-REx) field campaign to provide a means of validation (Platnick et al., 2017a; Wood et al., 2011). For decades, researchers have investigated the inherent challenges with comparing in situ measurements and remote retrievals (Feingold et al., 2006; Nakajima et al., 1991; Painemal and Zuidema, 2011; Platnick and Valero, 1995; Stephens and Tsay, 1990; Twomey and Cocks, 1989). We discuss how comparisons between in situ and remote measurements provide support for algorithmic development, but differences in sampling volumes reveal substantial limitations. Lastly, we demonstrate how improved radiometric accuracy and, to a lesser degree, an increase in the number of spectral measurements used in the retrieval decreases the set of acceptable solutions. For this analysis, we simulated top-ofatmosphere reflectance spectra sampled by HySICS.

Section 2 provides an overview of passive optical remote sensing of clouds from space, reviews current methods of deriving cloud optical properties from satellite measurements and introduces the optimal estimation method used in this analysis. Section 3 describes the data and forward model assumptions. Section 4 presents results with comparisons between the retrieved vertical profiles and the in situ data and highlights the dependence on radiometric accuracy. Section 4 also discusses challenges comparing in situ and remote measurements and the effects of increasing the number of wavelengths used in the retrieval. Section 5 provides an interpretation of the results and discusses potential future work to improve the methods.

# 2 Passive Optical Remote Sensing of Clouds

#### 2.1 The Bispectral Method

Deriving cloud optical properties from spectral reflectance measurements constitutes an inverse problem. As with any inverse problem, the solutions are highly dependent on the assumptions made in the forward model. When setting up a retrieval of cloud effective radius and cloud optical depth, the fundamental question is: What combination(s) of these variables would lead

to the set of observations measured? Let x be the state vector that contains the variables we seek to retrieve, thus  $x = (r_e, \tau_c)$ . To solve for x, we define a forward model, R, which maps our state vector to a set of spectral reflectance measurements, m, such that R(x) = m. The relationship between the desired state vector and spectral reflectance is non-linear.

The MODIS collection 6 cloud retrieval uses the bispectral method, relying on an extensive library of forward model calculations to retrieve the effective droplet radius,  $r_e$ , and cloud optical depth,  $\tau_c$  (Platnick et al., 2017b). The effective radius is defined mathematically as the ratio of the third moment of the droplet size distribution, n(r), to the second moment (Hansen and Travis, 1974):

$$r_e = \frac{\int_0^\infty r \, \pi \, r^2 \, n(r) \, dr}{\int_0^\infty \pi \, r^2 \, n(r) \, dr} \tag{1}$$

In addition to the desired state vector, each reflectance calculation depends on the solar and viewing geometry, the surface albedo, wavelength, and molecular and aerosol scattering and absorption. Note that these independent variables are not included in our equations. Lookup tables are created by computing reflectance over ranges of each these independent variables. The desired variables  $r_e$  and  $\tau_c$  are determined by computing the minimum  $\ell^2$ -norm difference between the measured reflectances, m, and the forward model estimates of reflectance, R(x).

#### 2.2 Monte Carlo Derived Weighting Functions

175

180

Unless droplet size is uniform throughout a cloud, the bispectral retrieval of effective radius depends on the two wavelengths chosen because average photon penetration depth within a cloud depends on the wavelength-dependent single scattering albedo (Platnick, 2000). Using a Monte Carlo model, we derived the weighting functions for the first seven spectral channels of MODIS to determine the average penetration depth for a vertically inhomogeneous cloud. A Monte Carlo model can simulate radiative transfer by treating photon-particle interactions stochastically. The critical element of this model is to define the processes of scattering and absorption probabilistically and then map each of these distributions onto a uniform probability distribution that can be sampled with a random number generator.

Clouds were modelled as horizontally infinite plane-parallel layers with a finite optical thickness and a vertical profile of effective radius. Liquid water content, *LWC*, is defined as the total mass of liquid water per unit volume of air:

195 
$$LWC = \int_0^\infty \rho \frac{4}{3} \pi r^3 n(r) dr$$
 (2)

where  $\rho$  is the density of liquid water. Assuming a parcel of air rises adiabatically, *LWC* increases linearly with geometric height. A linear relationship between liquid water content and height can be defined as:

$$200 \quad LWC(z) = LWC(0) + \left(LWC(H) - LWC(0)\right) \frac{z}{H} \tag{3}$$

where H is the total geometric depth of the cloud such that z = 0 at cloud base and z = H at cloud top. If we assume that total number concentration,  $N_c(z)$ , is constant with height, and we define the droplet distribution as consisting of a single radius,  $r_e$ , then we can remove the integral in Eq. (2) and use Eq. (3) to solve for the effective radius under the adiabatic assumption:

$$r_e(z) = \left(\frac{3}{4\pi N_c \rho} \left(LWC(0) + \left(LWC(H) - LWC(0)\right)\frac{z}{H}\right)\right)^{\frac{1}{3}} = \left(r_{bot}^3 + \left(r_{top}^3 - r_{bot}^3\right)\frac{z}{H}\right)^{\frac{1}{3}}$$
(4)

where  $r_{top}$  and  $r_{bot}$  are the effective radii at cloud top and cloud base, respectively (Platnick, 2000). We note that this adiabatic model is consistent with the commonly used Bennartz adiabatic model for a non-zero liquid water content value at cloud base (Bennartz, 2007). This droplet profile was used for all Monte Carlo simulations. Clouds were comprised of 100 plane-parallel layers with droplet size following a narrow gamma distribution in each layer with an effective variance,  $v_{eff}$ , of 0.077 (equivalent to libRadtran's width parameter,  $\alpha = \frac{1}{v_{eff}} - 3 = 10$ ) (Deirmendjian, 1964). Figure 1 shows normalized weighting functions for a vertically inhomogeneous cloud. Each weighting function represents the conditional probability of a photon scattered in the upward direction at cloud top, given that it penetrated to a max depth of  $\tau$ .

The wavelength-dependent column-weighted retrieved effective radius is approximated by:

205

215

$$r_e^* = \int_0^{\tau_c} r_e(\tau) \, w_\lambda(\tau) \, d\tau \tag{5}$$

where w<sub>λ</sub>(τ) is the wavelength-dependent weighting function (Platnick, 2000). For a non-constant droplet profile, Eq. (5) represents the retrieved effective radius for a given wavelength. From Fig. 1, it is evident that reflectance at different shortwave infrared wavelengths depend on the droplet profile. Since single scattering albedo, ω<sub>0</sub>, and to a lesser extent the asymmetry parameter, varies with wavelength, measurements at different wavelengths probe different depths within a cloud. In general, droplet absorption, defined by 1 – ω<sub>0</sub>, controls the vertically dependent weighting functions since photons that are more likely to be absorbed are less likely to penetrate deep into cloud layers. Figure 1 shows that on average, reflectance is dominated by scattering from the cloud top due to a greater proportion of photons reaching a maximum penetration depth in the upper region of the cloud.

Figure 1: Weighting functions of the MODIS instrument's first seven spectral channels. Model parameters are shown in the lower right corner.  $\mu_0$  is the cosine of the solar zenith angle,  $A_0$  is the surface albedo below the cloud layer,  $N_{photons}$  represents the number of photons used to compute each weighting function,  $N_{Layers}$  represents the number of homogeneous, plane-parallel layers, and  $\tau_0$  is the total optical depth of the cloud. Horizontal dashed lines represent the optical depth associated with the retrieved effective radius using the wavelength specified (Eq. (5)).

The development of a Monte Carlo simulation to model radiative transfer within clouds provided insight into how wavelength-dependent reflectance samples different layers of clouds. If  $r_e$  were constant with height the structure of each weighting function and the depth of average penetration would be irrelevant. Figure 1 shows that weighting functions at all seven MODIS wavelengths used in this analysis reach a similar maximum optical depth of about one. Furthermore, these weighting functions are broad and have considerable overlap, signifying considerable correlation between reflectances at different wavelengths.

Ideally, a set of orthogonal weighting functions that probe different depths of the cloud would be preferred. While this is not achieved with wavelengths in the visible and shortwave infrared region, measurements at many wavelengths can still be used to increase the retrieval signal-to-noise ratio.

# 2.3 The Optimal Estimation Method

245

Kokhanovsky and Rozanov (2012) applied an optimal estimation technique to retrieve a state vector that included droplet size at cloud top and base:  $\mathbf{x} = (r_{top}, r_{bot}, \tau_c)$ . Importantly, since only upper and lower values of the droplet profile are retrieved, this technique requires an assumption about the dependence of droplet size with altitude within cloud. Once droplet size is

retrieved at the top and base,  $r_e(\tau)$  can be determined continuously across the domain  $\tau = [0, \tau_c]$ . We assumed the droplet profile was adiabatic according to Eq. (4).

The Gauss-Newton iterative method, a technique used to solve non-linear least-squares problems, is used to solve for the state vector (Rodgers, 2000). At each iteration, the new state vector estimate is:

$$\mathbf{x}_{i+1} = \mathbf{x}_i + (\mathbf{S}_a^{-1} + \mathbf{K}_i^T \mathbf{S}_{\epsilon}^{-1} \mathbf{K}_i)^{-1} [\mathbf{K}_i^T \mathbf{S}_{\epsilon}^{-1} (\mathbf{m} - R(\mathbf{x}_i)) + \mathbf{S}_a(\mathbf{x}_i - \mathbf{x}_a)]$$
 (6)

where matrices are indicated in capitalized boldface, and vectors are indicated in lowercase, italicized boldface.  $\mathbf{x}_i$  is the state vector estimate of the  $i^{th}$  iteration,  $\mathbf{x}_a$  is the a priori state vector,  $\mathbf{S}_a$  is the a priori covariance matrix,  $\mathbf{K}_i$  is the Jacobian matrix of  $R(\mathbf{x}_i)$ , and  $\mathbf{S}_\epsilon$  is the measurement covariance matrix. The a priori state vector represents the best guess of the values of each retrieved variable before the Gauss-Newton iterative solution is derived. The a priori covariance matrix accounts for the uncertainty in the a priori guess and the relationship between each state variable. The measurement covariance matrix is the sum of measurement and forward model uncertainties:  $\mathbf{S}_\epsilon = \mathbf{S}_m + \mathbf{S}_{fm}$  (Poulsen et al., 2012).  $\mathbf{S}_m$  defines the measurement uncertainty at each wavelength and the correlation between measurements at different wavelengths. Two measurements with a non-zero covariance are at least partially redundant with respect to retrieving the desired variables.  $\mathbf{S}_{fm}$  defines the forward model uncertainty, which can be separated into two categories: sources proportional to the measured signal and the uncertainties in surface reflectance (Poulsen et al., 2012). The Jacobian is defined as:

$$\mathbf{K}_{i} = \nabla R(\mathbf{x}_{i}) = \begin{bmatrix} \frac{\partial R(\mathbf{x}_{i}\lambda_{1})}{\partial r_{top}} & \frac{\partial R(\mathbf{x}_{i}\lambda_{1})}{\partial r_{bot}} & \frac{\partial R(\mathbf{x}_{i}\lambda_{1})}{\partial \tau_{c}} \\ \frac{\partial R(\mathbf{x}_{i}\lambda_{2})}{\partial r_{top}} & \frac{\partial R(\mathbf{x}_{i}\lambda_{2})}{\partial r_{bot}} & \frac{\partial R(\mathbf{x}_{i}\lambda_{2})}{\partial \tau_{c}} \\ \vdots & \vdots & \ddots & \ddots \end{bmatrix}$$

$$(7)$$

The forward model, R, is used to compute reflectance at a set of wavelengths for some cloud state,  $x_i$ . The Jacobian represents the change in reflectance due to a perturbation in each state variable. Equation 6 balances several competing factors during each iteration: the difference between the measured and computed reflectances  $(m - R(x_i))$ , the difference between the current state estimate and the a priori  $(x_i - x_a)$ , and the rate of change of the estimated measurements with respect to the current state variable  $(K_i = \nabla R(x_i))$ .

To construct Eq. (7), we compute the change in reflectance due to a small change in one of the state variables. For example:

$$\frac{\partial R(x_i, \lambda_1)}{\partial r_i^{top}} \approx \frac{\Delta R\left((r_i^{top} + \Delta r^{top}, r_i^{bot}, \tau_{c_i}), \lambda_1\right)}{\Delta r^{top}}$$
(8)

is the change in reflectance due to a change in the radius at cloud top. We defined the change in the state variables as a fraction of the current iteration state vector. However, the magnitude of the change in reflectance depends on the initial values of the state variables. In addition, we need the change in reflectance to be greater than the measurement uncertainty. To ensure these conditions for all cases analyzed, the Jacobian was computed using the following fractions to estimate the partial derivatives:  $\Delta \mathbf{x}_i = \left[0.1 r_i^{top}, \ 0.35 r_i^{bot}, 0.1 \tau_{c_i}\right]$ . These values, derived using MODIS measurements and determined through trial and error, 285 ensured that the reflectance change exceeded the measurement uncertainty when the state vector was outside of a local minimum. These fractions need to precisely estimate the Jacobian, defined as the rate of change of reflectance with respect to an infinitesimal change in one of the state variables, and account for the MODIS measurement uncertainty. For example, we found if  $\Delta r_{bot}$  was too small, then  $\Delta R(x, \lambda)$  was dominated by measurement uncertainty. If  $\Delta r_{bot}$  was too large, we no longer accurately estimated the local slope. These fractions were used for all seven spectral channels because the MODIS 290 measurement uncertainty at these channels is roughly constant (adjustments should be made for use with other instruments). We note that the lower radiometric uncertainty of HySICS enables the detection of smaller changes in reflectance, enabling better estimates of the partial derivatives of the Jacobian.

During our analysis, we needed to constrain the solution space of the retrieved variables when using the Gauss-Newton iterative technique. We adopted the bound-constraint method by Doicu et al. (2003) to ensure the following constraints were satisfied:

$$r_{bot} < r_{top}$$

$$1 < r_{bot} < 25$$

$$1 < r_{top} < 25$$

$$(9)$$

The first constraint is required because we assumed an adiabatic droplet profile. If the first constraint is not satisfied, the adiabatic forward model assumption is invalidated. The second and third constraints are required because the pre-computed table of Mie calculations used to convert cloud properties to optical properties has an effective radius upper limit of  $25 \mu m$  (Emde et al., 2016). These constraints exclude the retrieval of drizzle or precipitation sized droplets. Future iterations of this work will use an expanded lookup table with a larger effective radius upper limit. For each iteration, we defined a new direction as:

$$\mathbf{p}_{i} = (\mathbf{S}_{a}^{-1} + \mathbf{K}_{i}^{T} \mathbf{S}_{\epsilon}^{-1} \mathbf{K}_{i})^{-1} [\mathbf{K}_{i}^{T} \mathbf{S}_{\epsilon}^{-1} (\mathbf{m} - R(\mathbf{x}_{i})) + \mathbf{S}_{a}(\mathbf{x}_{i} - \mathbf{x}_{a})]$$
(10)

such that the updated state vector guess was:  $\mathbf{x}_{i+1} = \mathbf{x}_i + \mathbf{p}_i$  (Doicu et al., 2003). We then solved for the maximum scalar value, a, that resulted in a new state vector,  $\mathbf{x}_{i+1} = \mathbf{x}_i + a\mathbf{p}_i$ , that met our state variable constraints and resulted in a lower  $\ell^2$ -norm between the estimate and true measurements:

$$\sqrt{\sum (R(\boldsymbol{x}_i + a\boldsymbol{p}_i) - \boldsymbol{m})^2} < \sqrt{\sum (R(\boldsymbol{x}_i) - \boldsymbol{m})^2}$$
(11)

From hereon we will use the term cost function, commonly noted as J, to refer to the left side of Eq. (11), the  $\ell^2$ -norm of the difference between the forward model reflectances and the true measurements. This was repeated until one of two convergence metrics was met. If the percent difference of the cost function between two successive iterations was less than 3%, the process was terminated. This value was adopted from an extensive number of retrievals. Values lower than 3% were the result of a local minima and further iterations never led to significant changes in the retrieved state vector. The other convergence criteria terminated the iterative process if the cost function was less than or equal to the  $\ell^2$ -norm of the total uncertainty (measurement and forward model),  $\delta m$ , and the previous iteration (Doicu et al., 2003):

$$\sqrt{\sum (R(\boldsymbol{x}_{i+1}) - \boldsymbol{m})^2} \le \sqrt{\sum (\delta \boldsymbol{m})^2} < \sqrt{\sum (R(\boldsymbol{x}_i) - \boldsymbol{m})^2}$$
(12)

Once convergence occurred, the posterior covariance matrix was computed. The uncertainties of the retrieved variables are the square root of the main diagonal (Rodgers, 2000).

$$S_{x} = (K^{T} S_{\epsilon}^{-1} K + S_{a}^{-1})^{-1}$$
(13)

#### 3 Data Used and Forward Model Assumptions

We applied our optimal estimation algorithm outlined in Sect. 2.3 to real data using multispectral measurements from MODIS. We used the MODIS spectral response functions to simulate top-of-atmosphere reflectance for the first seven spectral channels listed in Table 1, which reports the bandwidth and spectral resolution of each channel (MODIS Aqua and Terra Relative Spectral Response Functions, 2025). The resolution was estimated by computing the full-width at half-max. These seven spectral channels were used because they avoid water vapor absorption, simplifying the forward model. The following analysis only considered MODIS observations of liquid water clouds over the ocean with an optical depth of at least three. We chose this threshold because visible and shortwave infrared reflectances become more independent from one another with increasing optical depth (Nakajima and King, 1990; Twomey and Cocks, 1989). The uncertainties reported in Table 1 are average values for all pixels meeting the aforementioned constraints from the three MODIS swaths used in Fig. 3 (MODIS Characterization Support Team (MCST), 2017).

In this analysis, libRadtran (Emde et al., 2016) was used to run 1D DISORT (Stamnes et al., 2000) to compute forward modeled spectral reflectance. All clouds were defined as they were in the Monte Carlo simulations (Sect. 2.2) with an adiabatic droplet profile, 100 plane-parallel layers, and a gamma droplet distribution with a vertically constant effective variance of 0.077. This effective variance value was chosen based on analysis of in situ measurements of non-precipitating marine stratocumulus clouds from the VOCALS-REx flight campaign. We note that the assumption of a narrow monomodal droplet distribution is a simplification that is not valid for all clouds. In situ measurements have found that the droplet distribution tends to widen towards cloud top (Meyer et al., 2025). Furthermore, the presence of drizzle-sized droplets leads to a tail in the droplet distribution (Pörtge et al., 2023; Zinner et al., 2010). Future applications will explore alternate droplet distribution assumptions. We used the MODIS retrieval of cloud top height to define the upper boundary of the cloud, but this value is likely to be imperfectly aligned with the cloud top effective radius that we retrieved due to retrieval uncertainties in both. Cloud geometric thickness was set to 0.5 km, following our own analysis showing negligible impacts of cloud geometric thickness on reflectance for the wavelengths used. We used the U.S. 1976 standard atmosphere to define vertical profiles of all atmospheric gases (Anderson et al., 1986). Several forward model assumptions mirrored the forward model used in the MODIS collection 6 cloud optical properties retrieval algorithm (Amarasinghe et al., 2017). Maritime aerosols were assumed since only cloudy scenes over ocean were considered. Aerosol optical depth was defined as 0.1 for all cases (Amarasinghe et al., 2017). The Cox-Munk surface bidirectional reflectance model was used to account for the impact of wind speed and direction of the ocean surface (Amarasinghe et al., 2017; Cox and Munk, 1954).

Table 1: First seven spectral channels of the MODIS instrument. The reflectance uncertainty represents the average for water cloud observations over ocean with an optical thickness of at least three.

| Band | Bandwidth (nm) | Resolution (nm) | Reflectance<br>Uncertainty (%) |  |
|------|----------------|-----------------|--------------------------------|--|
| 1    | 614 - 681      | 47.5            | 1.95                           |  |
| 2    | 820 - 899      | 38.3            | 2.03                           |  |
| 3    | 452 - 481      | 19              | 1.91                           |  |
| 4    | 593 - 569      | 19.8            | 1.77                           |  |
| 5    | 1214 - 1271    | 23.5            | 1.65                           |  |
| 6    | 1596 - 1660    | 27.7            | 1.58                           |  |
| 7    | 2058 - 2175    | 52.3            | 1.65                           |  |

It is worth noting that, while an accurate forward model is desired, the primary function of the forward model and algorithm developed for this research was a proof-of-concept for retrieving vertical droplet profiles. Nevertheless, forward model uncertainty exists, and several works have detailed the many sources related to the retrieval of cloud optical properties (Platnick et al., 2017b; Poulsen et al., 2012; Watts et al., 1998). Our analysis considered clouds over ocean with an optical thickness of at least three. Therefore, we have ignored uncertainty related to surface reflectance. Sources of uncertainty most relevant to our retrieval and the limited cases we investigated include the precipitable column water amount above cloud, cloud top height, the effective variance of the droplet size distribution, vertical profiles of atmospheric gasses and aerosols, the plane-parallel assumption, and the instrument response (Platnick et al., 2017b; Poulsen et al., 2012).

We used measurements of relatively homogenous clouds to avoid the impacts of 3-D radiative effects on our retrieval. However, our assumption of a plane-parallel cloud does not accurately represent all cloud structures. Horizontally inhomogeneous clouds can lead to 3-D radiative effects, such as illumination and shadowing that result from a net horizontal radiative energy transport. Previous studies have shown that sub-pixel inhomogeneity leads to an increase in the uncertainty of retrieved effective radius (Zhang et al., 2012; Zhang and Platnick, 2011). We limited our analysis to relatively homogeneous marine stratus clouds, which have been shown to have modest 3-D biases on the retrieval of effective radius (Zinner et al., 2010). In addition, Zhang and Platnick (2011) showed the sub-pixel inhomogeneity index, the ratio of the standard deviation of the 16 sub-pixels of MOIDS-measured reflectance at 250 m spatial resolution to the mean, was a strong indicator of whether horizontal inhomogeneity affected the retrieval of effective radius. The authors concluded that effective radius retrievals were biased from 3-D radiative effects when the cloud had an inhomogeneity index greater than 0.3 (Zhang and Platnick, 2011). 3-D radiative effects of relatively homogenous clouds with an inhomogeneity index of less than 0.1 are likely minor, and, therefore, it may be possible to determine the cloud vertical structure using different shortwave infrared measurements (Zhang and Platnick, 2011). All pixels used in the development of our algorithm, including the three cases shown in Fig. 3, had an inhomogeneity index of less than 0.1. We will further address horizontal cloud structure and sub-pixel inhomogeneity in Sect. 5.

During VOCALS-REx, aircraft measurements of cloud droplet profiles were acquired from 14 flights conducted from 15 October to 15 November 2008. Some of the flight paths were spatially and temporally coincident with overpasses of the Terra and Aqua satellites (Wood et al., 2011). Over the entire duration of VOCALS-Rex, three vertical profiles were sampled within 5 minutes of a MODIS overpass, providing the best opportunities for comparison with remote retrievals. The Cloud Droplet Probe (CDP) manufactured by Droplet Measurement Technologies (Lance et al., 2010) measured forward scattering from a laser source to determine droplet diameters between 2 and 52 μm. The two-dimensional cloud optical array probe (2DC) by Particle Measurement Systems (Strapp et al., 2001) similarly measured droplet diameters between 25 and 1560 μm. To avoid redundancy, we ignored the 2DC data for droplet diameters less than 52 μm. These two data sets are distinct in that one consists primarily of typical cloud droplet sizes (~10 μm), whereas the other contains drizzle and precipitation-sized droplets (>100

μm). These two measurement systems enabled us to segregate clouds between those with and without drizzle by using a liquid water path threshold of 1 g  $m^{-2}$  as measured by the 2DC instrument, a slightly lower threshold than was used by Painemal and Zuidema (2011). This effectively removed any sampled clouds with droplets larger than 52  $\mu$ m from our data set. Painemal and Zuidema (2011) found a positive bias for the CDP LWC measurements compared to those from a hot wire probe. We applied their prescribed correction using a simple linear regression to the CDP droplet size distribution. In defining cloud top and bottom within the in situ data, we followed Painemal and Zuidema (2011), who defined the minimum liquid water content threshold of 0.03 g  $m^{-3}$  and a minimum total droplet number concentration threshold of 1  $cm^{-3}$ . Therefore, the cloud top and bottom were identified as the minimum and maximum altitudes where both criteria were satisfied.

Using over 100 VOCALS-REx in situ vertical profiles without drizzle or precipitation-size droplets, we computed the median profiles of effective radius, liquid water content, and number concentration by normalizing the vertical dimension, discretizing it into 30 bins and computing the median value for each. For each vertical bin, we found that a log-normal distribution best fit the measurements of effective radius and liquid water content, whereas a normal distribution was the best fit for number concentration. The shading in Fig. 2, which represents the average deviation from the median value, reflects these distributions: the shading is symmetric for number concentration and asymmetric for the effective radius and liquid water content. Figure 2 shows that the median profiles of effective radius and liquid water content closely resemble the theoretical adiabatic profiles overlaid in black. Figure 2 shows that the median profile of droplet effective radius was found to increase with altitude within cloud. We found the median effective radius at cloud top was about 37% larger than the value at cloud base for non-precipitating marine stratocumulus. These results justify the adiabatic assumption that results in a linear increase in liquid water content with altitude within cloud. We also note that the median profile of droplet number concentration is roughly constant with altitude, another assumption in the forward model.

The Gauss-Newton method assumes a Gaussian prior with symmetric uncertainty about the a priori value. The a priori value for the radius at cloud top and the optical depth was defined as the bispectral retrieval of effective radius and optical thickness, respectively, using MODIS measurements at  $0.65 \, \mu m$  and  $2.13 \, \mu m$  (Table 1). The a priori value for the radius at cloud bottom was defined as 70% of the retrieved effective radius. This percentage was derived from the median in situ vertical profile of effective radius (Fig. 2), which shows that the value of cloud bottom radius was 70% of the value of cloud top effective radius. The a priori uncertainties for cloud top radius and optical depth were set to their respective MODIS collection 6 bispectral retrieval uncertainties (Platnick et al., 2017b). For the three MODIS scenes analyzed in this study, the mean retrieval uncertainty of cloud effective radius for liquid water clouds over ocean with an optical depth of at least three was 8.2% ( $\sim 0.89 \, \mu m$ ). For optical thickness, the mean retrieval uncertainty was 5.1% ( $\sim 0.57$ ). We should note that the retrieved effective radius does not represent the droplet size at cloud top, as the weighting functions in Figure 1 demonstrate. Nevertheless, we consistently retrieved droplet sizes at cloud top close to the in situ measured values, proving the bispectral retrieval to be an effective value for the a priori at cloud top.

For the a priori uncertainty of droplet effective radius at cloud bottom, we scaled the bispectral retrieval uncertainty of effective radius using the weighting function for 2.13  $\mu m$ . Most photons at 2.13  $\mu m$  are scattered near cloud top. Therefore, we need to express a higher uncertainty for the a priori value for the radius at cloud bottom. We used the 2.13  $\mu m$  weighting function to determine the portion of the total measured signal with a maximum penetration depth within the upper and lower quartiles of the cloud. For the example cloud in Fig. 1, which has a similar droplet profile as the median effective radius profile found during the VOCALS-Rex campaign (Fig. 2), over 50% of the measured signal comes from the upper quartile of the cloud. Only 8% of the total signal comes from the lowest quartile. Thus, we adopted a cloud bottom uncertainty of a factor 6 larger than retrieved effective radius uncertainty. For the measurement covariance matrix,  $S_{\epsilon}$ , the measurement component,  $S_m$ , was defined as the uncertainty for the seven spectral channels of MODIS used in this analysis. Average reflectance uncertainty values are shown in Table 2. For the forward model component,  $S_{fm}$ , we assumed a value of 2.5% for all channels, adopting a similar used by Poulsen et al. (2012) over the same wavelength range. The different spectral measurements and the retrieved variables were assumed to be independent from one another. While the use of diagonal covariance matrices is common (King and Vaughan, 2012; Kokhanovsky and Rozanov, 2012), it does not reflect the true nature of the problem (see Sect. 5).

Figure 2: Median vertical profiles of effective radius, liquid water content, and droplet number concentration for non-precipitating clouds measured during the VOCALS-Rex flight campaign. The green line shows the median value of the distribution as a function of normalized cloud depth. The green-shaded area represents the average deviation above and below the median line. The black lines in the left and middle panels show the theoretical adiabatic profile using the boundary values found by the median profile. The vertical line in the right panel highlights the near-constant number concentration.

Section 4 shows retrievals for the three vertical profiles sampled within 5 minutes of a MODIS overpass. To account for the temporal displacement of cloud location, we applied a simple advection model using horizontal wind speed and direction measured on the aircraft. Using the median wind speed and direction from within the cloud, we computed the distance the cloud would have travelled during the time between MODIS and VOCALS-REx. The location was either projected forward or backward depending on whether the in situ sampling occurred before or after the MODIS overpass. The horizontal distance travelled by plane during in situ sampling exceeded the MODIS pixel sampling distance for all cases shown in this study. None of the droplet profiles shown in Sect. 4 were contained within a single pixel. After applying our advection model, the MODIS pixel closest to the newly projected location was used for the retrieval.

It is important to quantify the uncertainty of the in situ measurements since they were used to validate our retrieval algorithm. However, the CDP droplet size uncertainty estimate is attributed to several factors that make it difficult to quantify (Lance et al., 2010). Droplets that pass through the edges of the sampling area tend to have much higher uncertainty than droplets that pass through the center. Uncertainty due to coincidence, where multiple droplets pass through the sampling area within the sampling time of the detecting optics, is challenging to estimate because it depends on droplet size, particle concentration, and transit location within the sampling area. There are also limitations to the size resolution of the instrument due to the non-monotonic relationship between droplet size and the scattered laser light signal (Lance et al., 2010). Lance et al. (2010) used a water droplet generating system to determine the sizing accuracy of the CDP instrument. Using their results, we simplified the CDP measurement uncertainty for this analysis by defining an uncertainty of 20% for effective radii below 5  $\mu m$ , and an uncertainty of 10% for those above 5  $\mu m$ .

Table 2: The MODIS pixel observation time, location and sub-pixel inhomogeneity for each observation shown in Fig. 3, along with the corresponding VOCALS-REx in situ sample duration and the time difference between the two measurements.

| Figure | MODIS Observation time (UTC) | MODIS        | MODIS Sub-         | VOCALS-     | VOCALS-     | Time<br>difference |
|--------|------------------------------|--------------|--------------------|-------------|-------------|--------------------|
|        |                              | Observation  | pixel              | REx in situ | REx in situ |                    |
|        |                              | latitude and | inhomogeneity      | start time  | end time    | (min)              |
|        |                              | longitude    | index $H_{\sigma}$ | (UTC)       | (UTC)       | (111111)           |
| 3a     | Nov 11 2008                  | -24.0986,    | 0.09               | 18:45:20    | 18:45:50    | 8.88               |
|        | 18:54:28                     | -75.0013     |                    |             |             | 0.00               |
| 3b     | Nov 11 2008                  | -22.8188,    | 0.07               | 14:40:59    | 14:41:38    | 1.18               |
|        | 14:42:29                     | -73.0008     | 0.07               |             |             | 1.10               |
| 3c     | Nov 9 2008                   | -22.8970,    | 0.08               | 14:33:33    | 14:34:23    | 3.62               |
|        | 14:30:20                     | -73.0036     | 3.00               |             |             | 2.02               |

#### 4 Results

505

510

- Figures 3a, 3b, and 3c, show results applying the algorithm described in Sect. 2.3 for the retrievals of  $r_e(\tau)$  for clouds with in situ derived optical depths of 6.5, 11, and 19.5, respectively. Each figure also shows the MODIS Collection 6 bispectral retrieval of  $r_e$  and  $\tau_c$  using measurements at 0.65  $\mu m$  and 2.1  $\mu m$ . The time, location, and sub-pixel inhomogeneity of each MODIS observation, along with the VOCALS-REx sampling time and the time difference between each measurement is listed in Table 2. The bispectral retrieval of effective radius was within range of the cloud top in situ measurement for each case, 485 demonstrating consistency with its use as the a priori value for the radius at cloud top. The estimated liquid water path from the retrieved profile was closer to the in situ measured value than that derived from the bispectral retrieval for two of the three cases. The absolute difference between the multispectral estimate of liquid water path and that derived from the bispectral method for Fig. 3a, 3b, and 3c are 1.5, 0.7, and 12.5  $g\ m^{-2}$ , respectively. There are several factors contributing to these results. While the retrieval of the radius at cloud top was close to the in situ measurements in all cases, the retrieval of the radius at 490 cloud bottom was consistently larger than the in situ measurement. Second, we showed in Fig. 2 that the median vertical profile of droplet size of over 100 in situ measurements was close to adiabatic. This provided the basis for assuming an adiabatic droplet profile in the forward model, but this does not mean all in situ measured profiles were adiabatic, as evidenced by the large spread in the observations.
- The retrieved droplet profiles in Fig 3a and 3b follow a similar pattern to their respective in situ measurements, but both are larger than the in situ at nearly all levels within the cloud. This clearly affects the liquid water path comparisons. In particular, the retrieved effective radius at cloud base in Fig. 3b did not match the in situ measurements as well as the other two cases. As such, the estimated liquid water path using the retrieved profile was nearly identical to the value estimated by the bispectral retrieval. It proved difficult to determine exactly why this case fared worse than the other two, and it appears at odds with King and Vaughan (2012) who found uncertainty of the effective radius at cloud base to be at a minimum for a cloud optical depth of about 10 when using synthetic data.

We investigated the uniqueness of the retrieved solutions and found that the constraints applied to the Gauss-Newton technique outlined in Sect. 2.3 were required to retrieve droplet profiles that consistently resembled in situ measurements. The Gauss-Newton solver is not designed to find the global minimum. Instead, it converges towards a local minimum, which depends on the initial state vector estimate and the a priori (Rodgers, 2000). Indeed, there are many state vectors that will result in a set of spectral measurements within the MODIS measurement uncertainty because of the low relative weights near cloud base for the seven spectral channels used in this analysis. In our analysis we found that without constraints on the solution space, even an a priori close to the in situ values for the radius at cloud top and bottom could still lead to a solution with  $r_{top} 

Figure 3: Comparison between the effective radius calculated from in situ measurements (black circles), the MODIS bispectral retrieval of effective radius and optical depth (dotted vertical and horizontal blue lines, respectively), and the retrieved vertical profile using the optimal estimation method (pink dashed curve). The liquid water path estimate using in situ data, the MODIS retrievals, and our retrieved vertical profile are stated in the bolded box. Retrieval uncertainty for the effective radius at cloud top and bottom are shown as pink horizontal bars. Optical thickness retrieval uncertainty is represented by the pink vertical bar. In situ uncertainties are shown as black horizontal bars. The MODIS and in situ data were recorded on 9 Nov., 2008 (b) and 11 Nov., 2008 (a & c).

To provide insight into the sensitivity of the multispectral retrieval of  $r_{bot}$  with cloud optical depth we analyzed the components of the Jacobian. Figure 4 shows the change in estimated spectral reflectance,  $R(x_i)$ , due to a change in the cloud bottom radius for three clouds with different optical thicknesses but identical droplet profiles equal to the median droplet profile found in Fig. 1. For the  $i^{th}$  iteration and the  $j^{th}$  spectral channel, we estimate the change in reflectance using the following equation:  $\Delta R(x_i, \lambda_j) = R((r_{top,i}, r_{bot,i} + \Delta r_{bot,i}, \tau_{c_i}), \lambda_j) - R((r_{top,i}, r_{bot,i}, \tau_{c_i}), \lambda_j)$ . The y-axis of Fig. 4 shows this change for the seven spectral channels used in our multispectral retrieval. The behavior observed in Fig. 4 matches our expectations defined by the bispectral method. The change in estimated reflectance due to a change in  $r_{bot}$  is small in the visible where the droplet single scattering albedo is close to 1. In the shortwave infrared, water droplet absorption is proportional to the droplet radius. Thus, we expected a greater change in reflectance in the shortwave infrared spectral channels as the cloud bottom radius increases due to decreasing single scattering albedo. However, as optical depth increased, fewer photons penetrated the cloud's full depth, and eventually there was no change in reflectance.

The black circles and squares in Fig. 4 show the measurement uncertainty for the MODIS and HySICS instrument, respectively. These uncertainties are also displayed in absolute reflectance. We multiplied the reported percentage radiometric uncertainties of each instrument with the original reflectance,  $R((r_{top,i}, r_{bot,i}, \tau_{c_i}), \lambda_j)$ , from each spectral channel. While the change in reflectance and the corresponding uncertainties depend on the current state vector, the overall behavior remains the same, with the change in reflectance at non-absorbing wavelengths remaining below the measurement uncertainty, and the absorbing wavelengths exceeding it. For moderately thin clouds with an optical depth of less than 10, the change in reflectance typically exceeds the measurement uncertainty at wavelengths 1.64  $\mu m$  and 2.13  $\mu m$ . Changes in estimated reflectance when optical depth was 20 were equivalent or less than the measurement uncertainty. This represents an upper threshold in optical depth over which this retrieval is valid. Figure 4 also emphasizes expected improvements in this method from utilizing CPF measurements with radiometric uncertainty of 0.3% (Shea et al., 2020).

#### 4.1 Comparing in situ measurements with remote retrievals

It is important to acknowledge the difficulty in comparing remote retrievals of droplet size with their in situ measured counterparts. We used the in situ measurements as a guide while developing our algorithm, but it would be incorrect to treat them as absolute truth. Many previous studies found retrieved effective radius to be systematically larger than the corresponding in situ measured values (Meyer et al., 2025; Nakajima and Nakajma, 1995; Painemal and Zuidema, 2011; Twomey and Cocks, 1989). In a recent example, Meyer et al. (2025) found two different remote estimates of cloud effective radius, one using the bispectral technique, and the other using polarized reflectance measurements at scattering angles near the cloud bow, disagreed by 1-3  $\mu$ m. Both remote retrievals of effective radius were found to be larger, on average, than the coincident in situ derived value (Meyer et al., 2025). In addition, the two in situ cloud probes used in the analysis disagreed with one another by over 1  $\mu$ m (Meyer et al., 2025).

At nadir, the area sampled on the ground by a single MODIS pixel is approximately  $1 \ km^2$ . With a near-circular orbit, the Terra and Aqua satellites have a roughly constant height above Earth's surface of about 709 km. We estimate the sampling volume of a plane parallel cloud with a  $0.5 \ km$  thickness viewed by a single nadir-looking pixel to be about  $0.167 \ km^3$ . The sampling volume of the CDP laser probe is the product of the distance traveled by the plane over the sampling time with the sampling area of the instrument, which is about  $0.3 \ mm^2$  (Lance et al., 2010). The C130 aircraft that carried the CDP flew at an average speed of  $107 \ m \ s^{-1}$ . With a  $1 \ Hz$  sampling rate, the sampling volume of the CDP instrument was about  $32 \ cm^3$ , or  $3.2 \cdot 10^{-14} \ km^3$ . Therefore, the volumes sampled by the aircraft instruments and the MODIS spectrometer differ by 13 orders of magnitude. The enormous difference requires a discussion about the spatial variability of droplet size within marine stratocumulus clouds.

Figure 4: The change in our estimate of spectral reflectance due to a change in  $r_{bot}$ . The black circles show the MODIS measurement uncertainty in reflectance for each spectral channel. Three different cloud optical depths were compared to determine optical depth limits. The black squares show the measurement uncertainty for CPF is below the change in reflectance for each spectral channel.

Throughout the VOCALS-Rex flight campaign, numerous horizontal flight paths were conducted at a near-constant altitude. We used these legs to investigate the horizontal variability of effective radius in non-precipitating clouds. We constrained this analysis to horizontal legs where the plane had a maximum vertical displacement of 10 m during sampling. Figure 5b shows three representative samples of effective droplet radius, which showcases the two common regimes of behavior: steadily increasing or decreasing, and a quasi-stable mean. The range of these three horizontal legs conveys how much change in droplet size is possible. These ranges were calculated to be  $1.1 \mu m$  (red),  $5.5 \mu m$  (blue), and  $6.4 \mu m$  (yellow). The blue curve in Fig. 5b shows a stable effective radius over a horizontal range of 42 km. However, there are two sharp deviations near 2 and 4 km from the quasi-stable. The corresponding liquid water content measurements in Fig. 5a, and the droplet number concentrations in Fig. 5b, show sharp decreases. If these two outliers are removed, the range of the blue curve is  $2.9 \mu m$ , and the standard deviation is  $0.45 \mu m$ .

Using 50 horizontal in situ legs from VOCALS-REx, we computed the standard deviation of effective radius over three spatial scales representing the smallest and largest cross-track MODIS pixel sampling distances on the ground, and the HySICS sampling distance at nadir. For MODIS, the cross-track sampling distance is  $1 \, km$ , and at a scan angle of  $55^{\circ}$ , it is about  $5 \, km$  long (Nishihama et al., 1997). The sampling width at nadir for the HySICS instrument is  $0.5 \, km$ . We computed the standard deviation of droplet size over each length-scale by sliding windows over all 50 horizontal legs assuming the variability was invariant with direction within the horizontal plane. Figure 6 shows the histogram of standard deviations for the three length

scales. The median variability for  $0.5 \ km$ ,  $1 \ km$  and  $5 \ km$  was  $0.31 \ \mu m$ ,  $0.37 \ \mu m$ , and  $0.47 \ \mu m$ , respectively, and is represented as vertical dotted lines in Fig. 6. Thus, as scan angle increases, the pixel ground sampling area captures larger variations in droplet size. Figure 6 clearly demonstrates that for the marine stratus clouds sampled during VOCALS-Rex, the median variability of effective radius with respect to the horizontal plane decreases with decreasing sampling distance.

Figure 5: Three horizontal legs of (a) liquid water content, (b) effective radius and (c) droplet number concentration from three different non-precipitating marine stratus clouds. These measurements were made at a near-constant altitude during the VOCALS-Rex field campaign on 9 Nov. 2008. The standard deviations of effective radius over each profile are shown in the legend.

For comparing remote sensing with in situ measurements, it is important to recognize that the in situ profile represents a very small portion of the MODIS sampling volume. The retrieval of droplet size from MODIS measured radiance over a single pixel represents an integral over the sampled volume, which accounts for the contribution to reflectance at a given time, depth, and horizontal location (Feingold et al., 2006). In addition to the retrieval and in situ measurement uncertainties, the horizontal variability of droplet size is another ambiguity to consider when comparing remote retrievals with in situ measurements. Other factors that may contribute to this discrepancy are the assumed droplet size distribution effective variance and the imaginary index of refraction of liquid water (Meyer et al., 2025). Meyer et al. (2025) adjusted these two parameters in their forward radiative transfer model and found better agreement between in situ measurements and remote retrievals in some cases.

605

595

Figure 6: Histograms of the standard deviation of effective radius for horizontal legs of non-precipitating clouds from the VOCALS-REx flight campaign. The standard deviations were calculated over three pixel lengths. Horizontal dotted lines represent the median value for each length scale.

It should be noted that our classification of horizontal and vertical profiles is non-ideal but a necessary byproduct of airborne sampling. Every vertical profile sampled by VOCALS-REx spanned far more horizontal distance than vertical. Our intention with Fig. 2 was to show a representation of the distribution of droplet sizes sampled along the vertical dimension of a cloud; however, droplet horizontal variability is inevitably part of airborne vertical sampling.

Temporal variability also contributes to a discrepancy between in situ measurements and retrievals. The three vertical profiles shown in this study are those closest in time between a MODIS and in situ measurement for the entire VOCALS-Rex field campaign. The time difference between the MODIS observation and the VOCALS-REx measurements was defined as the difference between the recorded time of the MODIS pixel used in the retrieval with the time of the in situ sampling halfway through the vertical droplet profile. The time differences were 8.88 (Fig. 3a), 1.18 (Fig. 3b), and 3.62 minutes (Fig. 3c). We attempted to account for advection within our retrieval algorithm, but this does not account for the variability of cloud droplet size over time.

### 4.2 Simulated HySICS Spectra

615

620

We retrieved droplet profiles using the lookup table method introduced in Sect. 2.1 with simulated HySICS spectra to investigate two aspects that impact the solution space: the number of wavelengths used in the retrieval and the total uncertainty.

Simulated reflectance spectra were generated in a similar manner to the synthetic data generated by King and Vaughan (2012). libRadtran was used to compute top-of-atmosphere reflected radiance spectra for plane-parallel clouds over ocean with an adiabatic droplet profile using 1D DISORT (Emde et al., 2016; Stamnes et al., 2000). Reflectance at each HySICS spectral channel was estimated by convolving the radiance spectrum with the HySICS spectral response functions and normalizing with the incident solar irradiance. For this analysis, we assumed a uniform radiometric uncertainty of 0.3% across all spectral channels. Rather than estimating the uncertainty due to each component in the forward model, we investigated two scenarios: one with 2.7% forward model uncertainty, which is consistent with values used by Poulsen et al. (2012), and one with 0.7% forward model uncertainty. This resulted in total uncertainty values of 3% and 1%, respectively. The later scenario may be difficult to achieve, but future applications may be able to leverage the full contiguous spectrum sampled by HySICS to simultaneously retrieve properties that are usually assumed in the forward model. We will discuss this further in Sect. 5. We generated simulated spectra with varying uncertainty by sampling from a Gaussian distribution with zero mean. The lookup table method took about 50 times longer to compute than the iterative Gauss-Newton method, but once completed, we created a map from state space to measurement space. We repeated this process for different sets of spectral channels and for different values of measurement uncertainty to study how these two aspects affect the retrieval of droplet size at cloud base.

Figure 7: Simulated top-of-atmosphere HySICS reflectance spectrum for a cloudy scene with the 35 wavelengths used for the retrievals in Figs. 8 and 9 shown in pink.

To quantify how the number of spectral channels used in the retrieval affects the solution, we solved for the state vector with two different sets of wavelengths. The simulated reflectances were computed for a vertically inhomogeneous cloudy scene using the same solar-viewing geometry as the MODIS measurement shown in Figure 3.a, and a similar state vector to the one sampled by VOCALS-REx shown in the same figure. Forward modeled reflectance was computed for different combinations of the three state variables,  $\mathbf{x} = (r_{top}, r_{bot}, \tau_c)$ . Figure 8 shows the contours of the relative cost function, the fraction of the cost function with respect to the  $\ell^2$ -norm of the total uncertainty:  $\sqrt{\sum (R(\mathbf{x}_i) - \mathbf{m})^2} / \sqrt{\sum (\delta \mathbf{m})^2}$ . The left side of Fig. 8 was generated using seven spectral channels aligned with the seven MODIS spectral channels used in the multispectral retrieval

(Table 1). The right side of Fig. 8 was generated using 35 spectral channels across the visible and shortwave infrared that avoided water vapor and other gaseous absorption (Fig. 7). Both panels in Fig. 8 assumed a total uncertainty of 3%. According to the convergence criteria outlined in Sect. 2.3, the iterative algorithm terminates when the cost function is less than or equal to the  $\ell^2$ -norm of the total uncertainty (Eq. (12)). This region of cost function minima is located within the isopleth of one. State vectors within this isopleth lead to forward model reflectances within the uncertainty of the measurements. The solution space occupies three dimensions corresponding to the three retrieved variables. Figure 8 collapses the solution space into two dimensions by taking the difference between the cloud bottom radius dimension and the radius at cloud top associated with the global minimum relative cost function value. When we increased the number of spectral bands from seven to 35, the region of cost function minima decreased along the cloud bottom radius dimension by about 1  $\mu m$ . The light shading in both panels of Fig. 8 represent state vectors with a negative value of  $r_{top} - r_{bot}$ , which invalidates our forward model assumption. Figure 8 demonstrates that when using 35 spectral channels with a total uncertainty of 3%, the number of state vectors within the isopleth of one with a larger radius at cloud bottom than cloud top is reduced.

Figure 8: Contours of the relative cost function between the libRadtran-estimated reflectance and simulated HySICS top-of-atmosphere reflectance. The left panel was generated using 7 spectral channels that align with the first seven MODIS spectral channels (Table 1) and the right panel with 35 spectral channels throughout the visible and shortwave infrared (Fig. 7). The y-axis is the difference between the cloud top radius value associated with the global minimum relative cost function and the cloud bottom radius. The light shading highlights the negative  $r_{top} - r_{bot}$  region. The green x represents the state vector value used to generate the simulated HySICS measurements.

Figure 9 illustrates the impact of total uncertainty on the solution space. We computed forward modeled reflectances for the same scene described above for Fig. 8, but we kept the number of wavelengths used in the retrieval constant, using the same

35 spectral channels as the right side of Fig. 8. The left side of Fig. 9 shows the relative cost function using synthetic spectra with 3% total uncertainty, whereas the right side used synthetic spectra with 1% total uncertainty. Unlike the small reduction of the cost function minima region with increasing wavelengths, Fig. 9 shows a significant reduction with decreasing total uncertainty. The region within the isopleth of one decreased along the cloud bottom radius dimension by about  $3.75 \mu m$ . The right panel of Fig. 9 shows a steeper solution space for retrievals with a total uncertainty of 1% as compared to 3. We consistently found the gradient to be large outside the convergence region, but once inside, the gradient was quite small. Even if we allow the iterations to continue within the isopleth of one, the slopes are small enough that the algorithm quickly converges at a local minima. It's important to note that the shape of the contours depends on the state vector and varies with each simulated HySICS spectra because of the addition of Gaussian noise. Both figures show the mean state for the particular state vector used.

Figure 9: Contours of the relative cost function between the libRadtran-estimated reflectance and simulated HySICS top-of-atmosphere reflectance. The left and right panels used simulated HySICS reflectances with 2% and 0.3% measurement uncertainty, respectively. Both panels used 35 spectral channels throughout the visible and shortwave infrared (Fig. 7). The y-axis is the difference between the cloud top radius value associated with the global minimum relative cost function, and the cloud bottom radius. The light shading highlights the negative  $r_{top} - r_{bot}$  region. The green x represents the state vector value used to generate the simulated HySICS measurements.

The widths of the contours in Figs. 8 and 9 represent the retrieval uncertainty for the radius at cloud bottom (y-axis) and cloud optical thickness (x-axis). When the total uncertainty is reduced from 3% to 1%, Fig. 9 shows that the uncertainty of the radius at cloud bottom decreases from about 8.5  $\mu m$  to about 4.75  $\mu m$ , and the uncertainty for the retrieved cloud optical thickness decreases from 1.1 to 0.2. The average MODIS bispectral retrieval of optical thickness for warm clouds over the ocean with

an optical thickness of at least three falls within the two retrieval uncertainties for a total uncertainty of 3% and 1% (0.51). Due to the uncertainty in the retrieved cloud bottom radius, our values are higher than the average MODIS bispectral retrieval of effective radius (0.89  $\mu$ m), which was expected given the lack of signal from the lower portion of the cloud. Our results appear to align with the results of King and Vaughan (2012), who calculated droplet profile retrieval uncertainties for different total uncertainties.

#### 5 Discussion and Conclusions

To prepare for upcoming high-accuracy, full-spectral space-borne hyperspectral measurements, we have developed new methods to retrieve vertical profiles of cloud droplet size. We extended the results of King and Vaughan (2012) by developing an iterative Gauss-Newton technique that was applied to real data. Using the first seven spectral channels of MODIS and coincident in situ measurements from the VOCALS-REx flight campaign, we showed that retrieving a profile of effective radius is possible, but solving for the effective radius at cloud base is problematic because of the similarity of weighting functions at various visible and shortwave infrared wavelengths. Other studies have retrieved vertical profiles of effective 715 radius from MODIS data without addressing solution uniqueness (Chang and Li, 2003; King and Vaughan, 2012; Kokhanovsky and Rozanov, 2012). Chang and Li (2003) outlined methods to retrieve droplet profiles, applied these methods to real data, and investigated changes in retrieved variables due to reflectance uncertainty. Kokhanovsky and Rozanov (2012) used the Gauss-Newton optimal estimation method to retrieve droplet profiles, demonstrating that their method worked on real 720 data. King and Vaughan (2012) investigated the impact of measurement uncertainty on retrieval uncertainty using synthetic hyperspectral data but did not address solution uniqueness. The limited unique information in the MODIS bands used in our analysis led to a non-unique solution, with many droplet profiles leading to a set of spectral measurements within the MODIS measurement uncertainty. We implemented a constrained form of the algorithm, which reduced the solution space to a set consistent with the forward model assumptions, leading to state vectors that more closely matched the in situ measurements.

Coincident in situ measurements were used to validate the retrieval. Algorithmic parameters described in Sect. 2.3 were tuned such that the retrieved droplet profile closely matched the in situ measurements. However, in situ measurements cannot be treated as absolute truth because the sampling volumes of VOCALS-REx and the MODIS measurements differ by 13 orders of magnitude. Using VOCALS-REx in situ data, we found the median horizontal variability of effective radius to be between 0.31  $\mu$ m and 0.47  $\mu$ m for the three pixel ground sampling distances of 0.5, 1, and 5 km. The retrieved droplet size is representative of a radiatively-weighted mean over the sampling volume. The in situ measurement is considered a point measurement, which is more susceptible to spatial perturbations. Horizontal variability of effective radius over the MODIS pixel sampling area should be taken into account, along with the in situ measurement and retrieval uncertainty, when making these comparisons.

The three in situ vertical profiles analyzed in this study spanned multiple MODIS pixels. Unfortunately, there was never a scenario where a vertical profile was completely contained within a single pixel. We found that the overlapping pixel with an optical depth closest to the in situ measurement performed best in the retrieval. This result demonstrates the important interdependence between the retrieved variables: we required an accurate a priori of optical depth to retrieve droplet sizes that more closely matched the in situ measurements. Indeed, Figs. 8 and 9 demonstrate the importance of an accurate a priori and initial guess because these values help define the approach to the convergence region. Future work will explore the use of a non-diagonal a priori covariance matrix and the interdependence between the retrieved variables.

The first seven spectral channels of MODIS were used in this analysis because they avoid water vapor absorption. We also simulated reflectance for 35 HySICS spectral channels that are relatively free of water vapor and other gaseous absorption to investigate how the number of spectral measurements and reduced total uncertainty affect the retrieval. We assumed radiometric uncertainty for both instruments was uncorrelated, a simplification of their true nature. The measurement uncertainty of HySICS reported by Kopp et al. (2017) indicated that neighboring spectral channels strongly covary with one another. Future iterations of this work will leverage these results to define the off-diagonal elements of the measurement covariance matrix. When we increased the number of wavelengths from seven to 35 using simulated HySICS spectra, we found that the region of cost function minima within the solution space decreased along the cloud bottom radius dimension by about 1  $\mu m$ . Future applications with hyperspectral measurements from CPF will consider hundreds of spectral bands, including those in the wings of shortwave infrared water vapor absorption features. Perhaps this additional information will enhance the modest improvements to the retrieval of droplet size at cloud base shown in Fig. 8 by increasing the retrieval signal-to-noise ratio.

Minimizing forward model uncertainty leads to a measurement-limited solution that may be unachievable with CPF's unprecedented accuracy. Indeed, reducing total uncertainty to 1% may prove difficult, but future work should strive for more accurate forward models. Assuming a droplet profile is just one assumption that reduces forward model uncertainty because the assumption of a vertically homogeneous droplet profile is known to be a simplification for certain types of clouds (Platnick, 2000). For example, Meyer et al. (2025) explored adjusting the assumed droplet size distribution effective variance and the imaginary index of refraction of liquid water and found better agreement between in situ and remote measurements in some cases. In the future, an optimal estimation algorithm may be able to leverage the full spectrum of CPF to estimate cloud phase (Pilewskie and Twomey, 1987), cloud top height (Rozanov and Kokhanovsky, 2004), above-cloud column water vapor (Albert et al., 2001), carbon dioxide column amount (Buchwitz and Burrows, 2004), and aerosol optical depth (Mauceri et al., 2019), reducing forward model uncertainty for the droplet profile retrieval by limiting the number of assumptions.

For this analysis, we assumed plane-parallel clouds with a vertically inhomogeneous droplet profile. However, real cloud structures often exhibit horizontal variation within a single pixel, likely impacting the retrieval of an effective radius profile.

Several previous studies have investigated the impact of sub-pixel horizontal inhomogeneity on the bispectral retrieval of effective radius using Large Eddy Simulations (LES) to generate horizontally and vertically inhomogeneous cloud fields. An LES model can account for turbulent mixing in the boundary layer, heat and moisture transport, and can resolve cloud droplet size distributions, which provide more realistic 3-D cloud structures than a 1-D model. Zhang et al. (2012) used the full details of LES-simulated cloud microphysics and a MODIS cloud properties retrieval simulator to investigate the systematic difference between effective radius retrievals  $r_{3.7}$  and  $r_{2.1}$ . The authors found that the difference between  $r_{3.7}$  and  $r_{2.1}$  increases as the sub-pixel inhomogeneity index increases, and attribute sub-pixel variations of cloud optical depth as the primary cause of these differences (Zhang et al., 2012). A similar study by Zinner et al. (2010) also used LES-generated cloud fields to investigate the impact of 3-D radiative effects on retrievals of effective radius and found them to be pronounced only for scattered cumulus scenes, whereas the effects for marine stratus were small.

We performed single-pixel analysis on real MODIS measurements, which precludes any knowledge of sub-pixel information other than the sub-pixel inhomogeneity index. We limited the observations used in our analysis to MODIS observations of marine stratus clouds with a sub-pixel inhomogeneity index of less than 0.1 to reduce the impact of 3-D radiative effects (Zhang and Platnick, 2011). While we attempted to minimize potential 3-D impacts on our retrieval, to broaden the applicability of similar approaches, future work using LES-generated cloud fields will be necessary to investigate the impacts of sub-pixel inhomogeneity and 3-D radiative biases on the retrieval of a droplet profile. Previous work by Zhang et al. (2016) established a mathematical framework for estimating the retrieval uncertainty for scenes with large sub-pixel reflectance variations. We expect similar biases to those found in previous studies for effective radius when using the first seven MODIS spectral channels or measurements from CPF with high spatial inhomogeneity to retrieve a droplet profile. Mitigation of 3-D effects on traditional 1-D retrievals is an ongoing field of research. Several promising studies have shown machine learning techniques trained on LES data are capable of overcoming some 3-D biases (Nataraja et al., 2022; Okamura et al., 2017). The applicability of machine learning to overcome 3-D biases impacting droplet profile retrievals should be explored.

In addition to 3-D biases, other factors can affect droplet profile retrievals of 1-D clouds that require further investigation. Our analysis considered relatively homogenous marine stratus clouds over ocean with an optical thickness of at least three. Thus, we could ignore uncertainty related to surface albedo because the portion of the top-of-atmosphere signal due to surface reflectance is negligible. Future work observing clouds over land or optically thin or broken clouds over ocean will need to investigate how surface reflectance impacts the retrieval of a droplet profile. Without polarized radiance measurements, we were unable to retrieve the effective variance and, therefore, assumed a narrow monomodal distribution. While this likely leads to uncertainty in the retrieved effective radius, a narrow distribution is a reasonable assumption based on the results of multiple studies that showed that the presence of drizzle has only modest impacts on the retrieval of effective radius at different wavelengths (Painemal and Zuidema, 2011; Zhang et al., 2012; Zinner et al., 2010). Future work will also explore effective variance assumptions.

We did not investigate the retrieval sensitivity to solar and viewing geometry, but future studies should include this. Platnick (2000) demonstrated the retrieval of effective radius for vertically inhomogeneous clouds depends on the solar-viewing geometry by showing that weighting functions increasingly sample the upper region of the cloud as viewing angle increases. Accordingly, we expect our droplet profile retrieval to estimate larger values at cloud top and bottom as viewing angle increases, if the cloud under observation has a nonhomogeneous vertical droplet profile. In addition, the bispectral retrieval uncertainty of effective radius is larger for small droplets because of non-unique reflectances for the two channels used; the effect is more pronounced for low solar zenith angles. However, photons have a deeper average penetration depth with low solar zenith angles, potentially leading to a set of weighting functions with a higher degree of orthogonality. The balance between these two opposing effects should be investigated.

The optical depth over which the droplet size at cloud base can be retrieved is limited by the uncertainty of the measurements and the forward model. Changes in the spectral reflectance due to a change in droplet size at cloud base were often below the MODIS measurement uncertainty for optically thick clouds. Figure 4 illustrates that CPF measurement uncertainty, which is lower than the estimated change in reflectance at every spectral channel used in this analysis, will improve the retrieval of droplet size at cloud base. Furthermore, Fig. 9 shows a 3.75  $\mu m$  reduction in the region of cost function minima along the cloud bottom radius dimension when total uncertainty is reduced from 3% to 1%. These results underscore the importance of higher accuracy from the next generation of space-borne spectrometers, and a need for more accurate forward models. The results of this study suggest that a reduction in radiometric and forward model uncertainty is a more significant factor for retrieving droplet profiles than increasing the number of spectral bands.

Code and Data Availability. The retrieval algorithm developed for this paper is freely available on GitHub (https://github.com/andrewjbuggee/multispectral-retrieval-using-MODIS). The MODIS L1B and geolocation files (https://doi.org/10.5067/MODIS/MYD021KM.061) & https://doi.org/10.5067/MODIS/MOD021KM.061), and the L2 files (https://doi.org/10.5067/MODIS/MODIS/MODO6\_L2.061) & https://doi.org/10.5067/MODIS/MYD06\_L2.061) used for retrieving droplet profiles are described within the previously mentioned GitHub repository and freely available at NASA's Level-1 and Atmosphere Archive & Distribution System Distributed Active Archive Center (LAADS-DAAC), hosted at NASA's Goddard Space Flight Center (GSFC): https://ladsweb.modaps.eosdis.nasa.gov/. The VOCALS-REx data used for comparison with the multispectral retrievals are similarly defined within the GitHub repository. These data are maintained by the National Center for Atmospheric Research Earth Observing Laboratory Field Data Archive (NCAR EOL) and are freely available at: https://doi.org/10.5065/D60863M8.

*Author Contributions*. AJB led the study, developed the retrieval algorithm, and wrote the paper. PP made extensive paper edits and provided ideas, comments, and suggestions throughout the project.

Competing Interests. None of the authors of this paper have competing interests.

Acknowledgements. The authors would like to thank Dr. Odele Coddington, Dr. Yolanda Shea, Dr. Kevin McGouldrick, and Dr. Zhien Wang for reading early drafts and providing feedback. Their comments significantly improved this project. We would like to thank Dr. Greg Kopp for discussions on the HySICS instrument and for providing the spectral response functions. The authors would also like to thank the reviewers, whose thorough reading and thoughtful much improved our study. This material is based upon work supported by the National Science Foundation Graduate Research Fellowship under Grant No. 2040434.

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
