# Peer review of "Retrieving Vertical Profiles of Cloud Droplet Effective Radius using Multispectral Measurements from MODIS: Examples and Limitations"

_EGUsphere, 2025_

## Referee Comment (RC4)

The manuscript discusses the retrieval of vertical droplet size profiles from multispectral solar reflectance observations with high radiometric accuracy using a constrained optimal estimation inversion technique applied to MODIS observations. The study leverages VOCALS-REx field campaign data to develop the forward model constraints, which improves retrievals from, in particular, the lower optical depth levels of moderately thick liquid phase clouds. The high radiometric accuracy and spectral sampling follows from the design specifications for the upcoming CLARREO Pathfinder (CPF) instrument to be flown on the ISS in the 2026-27 timeframe. The findings highlight the value of high radiometric accuracy compared with current state of the art satellite imagers, as well as the challenges in comparing retrievals against *in situ* measurements in heterogenous clouds due to the profound differences in sampling volume.

The study contributes to a better understanding of future cloud microphysical profile retrieval information content from solar reflectance observations and nicely expands on previous efforts. The manuscript is very well-written, successfully capturing the history of previous studies as well as appropriate details of the author's work. I characterized one comment as major but all others are minor.

Major comment

Fig. 4: This is a very important figure in terms of the study findings but I was confused.

(1) I interpret the y-axis to be absolute reflectance, not relative reflectance like the accuracy specs for MODIS or CPF. If that's correct, the choice of r_bottom and dr_bottom will scale the y-axis value Jacobians without changing the MODIS or CPF lines. If that's the case, I don't know how to interpret the results (e.g., if dr_bottom is effectively zero, all bars will be zero on the y-axis). If not the case, please elaborate.
(2) While the text mentions the spectral dependence of the Jacobians, it's not clear which channel(s) are being used in the figure.
(3) The y-axis for the Jacobians should be labeled delta reflectance, delta reflectance/drbottom, or something similar unless I'm mistaken about (1).

Minor comments

L34: "effective droplet radius" or "effective droplet absorption" is proportional to 1-ssa? While it's true that $kr_e$ ~ 1-ssa for an absorbing wavelength, it's an ill-defined definition for $r_e$ when ssa=unity (i.e., reduces to $r_e$ = 0/0).

L49, 60: While Twomey and Cocks (1982) provides a nice overview of the retrieval theory, a more focused retrieval study was done in the follow-up Twomey and Cocks (1989, *Beitr. Phys. Atmosph.),* which used 5 spectral channels simultaneously in the retrieval (not bi-spectral) and presented the solution space in terms of residual contour plots similar to your Figs. 7, 8.

I'm not suggesting you include the following relevant historic $\tau$, $r_e$ retrievals references but just for awareness: Other airborne retrievals (Foot (1988), Rawlins and Foot (1990)); AVHRR (Arking and Childs, 1985), Platnick and Twomey (1994).

L60: suggest adding the qualifier "nearly independent from one another for optically thicker clouds …"

L64: "… radius, cloud optical depth, and various surface spectral reflectance assumptions."

L123, Sect. 4.2: As a simulation, it doesn't make a difference for present purposes, but I'm curious why the simulations were done for EMIT spectra instead of CPF, which is mentioned prominently as the motivation for the study (including the abstract). Was it in anticipation of doing EMIT retrievals as a follow-on? It would be useful to explain the rationale.

L184: What effective variance ($v_e$) is used? The alpha "width parameter" is mentioned on L294 but would be helpful to put it in terms of $v_e$. Are the same value(s) used for all 100 layers?

L188, 193: Eq. 5 is an approximation, though a reasonably good one, for the retrieved $r_e$ since an exact weighting function is confounded by multiple scattering. I.e., suggest "represents the approximate retrieved …"

L91: A nice summary of the previous work. Platnick (2000) also did an information content study for MODIS-like imager, including the effect of calibration uncertainty, to help understand the number of independent parameters that can be retrieved for vertical profile inversions. Hard to make apple-to-apple comparisons but do your results seem somewhat consistent? Similar question with respect Fig. 8 accuracy sensitivity.

Fig. 1: Please try to add some contrast to the line plot colors as some are hard to distinguish (esp. for color blind readers).

L253, 254: Good idea.

L295: The MODIS retrieval wouldn't correspond exactly to the upper boundary $r_e$ according to Fig. 1. Likely a small difference but worth a comment.

L361/Sect. 4.1: For further context on the confounding effects that uncertainties in situ probes have on retrieval validation, including sampling issues associated with vertical and horizontal heterogeneity, I suggest looking at the recent Meyer et al. ORACLES study (amt.copernicus.org/articles/18/981/2025/). The paper discusses airborne spectral retrievals compared against two in situ cloud probes (CAS, PDI) having different measurement approaches in addition to some retrieval forward model errors. Retrieval

evaluation with airborne probes continues to be an inherently challenging problem for the community. Nice discussion here and in Sect. 4.1.

L377: Not sure that the cloud-top $r_e$ retrievals "validate" use of the 2.1 μm MODIS bi-spectral retreival as a prior as much as demonstrates consistency with its use as a prior. I.e., much of the upper cloud $r_e$ information content is coming from the 2.1 μm channel, regardless of which algorithm is used.

Fig. 3a and 3c have the same MODIS retrieval values (blue dashed lines). One must be incorrect.

L409: I think this often gets lost on those who use gradient searches as part of inversion algorithms, especially in higher dimensional spaces. So, good to make this point, as obvious as it may seem. Is there an example solution contour plot associated with Fig. 3 that you could show to illustrate this point (i.e., similar to Figs. 7, 8)?

L440: suggest "… approximately 1 km$^2$". The effective pixel shape in the across track direction suffers from the finite integration time and so has a ~2 km triangular wide spatial weighting function for most MODIS channels though a bit less so for "1 km" channels aggregated from the native 250 m (bands 1, 2) and 500 m (bands 3-7) detector arrays. That said, L462 is correct that the across track sampling is 1 km.

L446: Interesting number. Thanks for making the calculation.

Fig. 5 caption, L454, 455, and later text/captions.: Constant altitude flight lines aren't usually considered a "profile" in airborne sampling vernacular (at least in the cloud and aerosol community). Also, elsewhere in the manuscript profiles is used, without qualification, to describer vertical sizes only so it will be a source of confusion. Try "horizontal legs" or just "legs". I realize that constant altitude across three different clouds during the campaign may end up sampling different depths relative to cloud top and so have some verical profile information (e.g., the yellow curve in Fig. 5).

L458: "… and 6 μm (yellow)"

Figs. 7, 8: Nice demonstration of more channels v. better accuracy, with the latter being the only way to dramatically reduce the delta radius solution space uncertainty. That's an important result. (1) Initially, I didn't notice that the y-axis had both positive and negative values. Would be helpful to add a horizontal line to the zero value so readers can quickly appreciate that a large region of the space is outside the constraint. Or add a slight shading to the negative regions. (2) Add a point on the plots to indicate the modeled cloud optical depth and delta effective radius that was used in the simulation (didn't see it mentioned in the text, nor the cloud top effective radius).

*Data Availability*: If MODIS L2 cloud data was used, please also mention that these files were obtained from LAADS. I strongly suggest providing a doi for both the L1B and L2 files, which should be available on the LAADS product information pages.

---

## Author Comment (AC2)

**Authors' response to comments from anonymous reviewer 1**

Andrew J. Buggee & Peter Pilewskie

May 2025

We thank the reviewer for their thorough reading of our paper and for providing thoughtful comments. We have addressed each one below.

**Reviewer's General Comments**

1. This paper is a study on the potential of the upcoming CLARREO Pathfinder (CPF) mission to provide more detailed retrievals of cloud properties than heritage (MODIS-like) imaging sensors thanks to a combination of decreased radiometric uncertainty and increased spectral sampling. The specific geophysical situation studied is joint retrieval of cloud optical depth (COD) and effective radius at the top and bottom of the cloud, for marine stratocumulus scenes. In contrast, one of the major current large-scale approaches (MODIS-like bispectral) retrieves COD and near-top effective radius using a pair of bands, and makes multiple bispectral retrievals with their differences being semi-informative on cloud structure.

2. There are two main parts to the analysis. First is use of VOCALS-REX field campaign data to set up some case studies for the proposed retrieval method using MODIS. This has the advantage of being something which can be tested now. The second is a sensitivity study, comparing the capabilities of a MODIS-like sensor with the EMIT instrument as a surrogate for CPF. Together these provide a starting point for moving towards this next level of detail in passive imager cloud retrieval algorithms.

3. The manuscript is in scope for AMT. There is a lot to like about this paper: it tackles an important problem, is clearly written, and has some nuance to the discussion. I particularly appreciated the discussion of sampling scales in the VOCALS-REX part of the discussion. The quality of writing and presentation are good. It is (mostly) well-referenced. I also appreciate the authors quickly noticing and fixing the incorrect panel of Figure 3. I think it is worth consideration for an AMT science highlight.

4.  That said, there are points where I think clarification and deeper discussion with respect to realistic performance are needed. As the paper does not claim to be a fully operational approach it does not need to be the final word on the matter, but as a case study and example of what can be done, I think there are sections where more caveats should be discussed, and there are a few things I was not certain about. I recommend minor revisions before publication. I would be willing to review the revision, if the Editor would like.

**Reviewer's Specific Comments**

1.  Line 31: I'm not sure I'd seen COD described as mean photon free paths through the cloud before, although I can see this framing makes sense as it is the integral of extinction coefficient which is units e.g. km$^{-1}$ (extinction events per unit distance). Normally it is just referred to as vertical integral of extinction coefficient. I'm curious if there's a reason the authors picked this particular framing for COD.

    a.  *Authors' Response:* We find it useful to describe optical depth as the number of photon mean free paths because it is an intuitive way to think about a quantity that depends on how the number density of particles and their scattering and absorption cross sections vary along a particular path. Bohren and Clothiaux (2006) show that the probability of a photon traveling a geometric distance $x$ before being scattered or absorbed (assuming no multiple scattering), is $p(x) = (\kappa + \beta) \exp(-(\kappa + \beta)x)$, where $\kappa$ is the bulk absorption coefficient and $\beta$ is the bulk scattering coefficient. The mean free path is then $< x > = \int_0^\infty x \, p(x) dx = \frac{1}{\kappa + \beta} = \ell$. Since $\tau_\lambda = \int (\kappa + \beta) dz$, we can also define optical depth as $\tau_\lambda = \int \frac{dz}{\ell}$, the number of mean free paths.

    b.  *Proposed changes to the manuscript:* None.

2.  Line 31: not sure I'd describe effective radius retrievals as "extinction-weighted" but maybe "photon-penetration-weighted"? For a really deep convective cloud, for example, the photons seen from space are still mostly coming from near the cloud top even if the water/extinction would be somewhat further down. And this is in line with e.g. the Platnick (2000) reference cited and weighting functions shown in the paper. To me "extinction-weighted" implies an optical center of mass.

a. *Authors' Response:* We used the term 'extinction-weighted' to convey that the retrieved effective radius is a vertically-weighted average that depends on the extinction properties of liquid water at the set of wavelengths used in the retrieval. However, it is clear that our term may lead to some confusion, while the reviewer's proposed term is strictly correct.

b. *Proposed changes to the manuscript:* We will use the term "photon-penetration-weighted" instead of "extinction-weighted".

Commented [AB1]: It felt odd to say we would use 'optical path weighted' instead of their suggested 'photon-penetration-weighted' without a reason. So I just said we would use their suggestion. Do you have objections?

Commented [PP2R1]: okay

3. Introduction, general: I like the historical discussion, but there are a few omissions that I think are quite relevant. One is the ORAC retrieval which came out of the same lab as Clive Rodgers who put down the Optimal Estimation (OE) formalism used here, applied mostly to European sensors (ATSRs and successors). See Sayer et al (2011) and Poulsen et al (2012). This isn't an explicitly bispectral approach (uses all bands together) but only retrieved a single effective radius (sensitivity from 1.6 and 3.7 micron bands) as opposed to attempting a profile. Another is the VISST algorithm applied to cloud properties from MODIS observations within CERES pixels (as part of the CERES data processing chain), which is also not bispectral but again retrieving a single effective radius from visible and multiple SWIR bands (0.65, 1.6, 2.1, 3.7 micron). The reference I use for this is Minnis et al (2011) – that paper cites some earlier AVHRR work using that algorithm from the late 1990s, but it's in conference proceedings that don't seem to be broadly available, so I can't say for sure what was done. There is also earlier OE work by e.g. Heidinger (2003) applied to the AVHRRs (a lot of later work from that NOAA team focuses on the infrared, but the above algorithm also used solar radiances and is more conceptually similar to bispectral). All of these approaches (ORAC, CERES, AVHRR) have been applied to multi-decadal multi-sensor records and approach the question of effective radius parameterization a bit differently from either the bispectral method or the profiling method, so I think merit some discussion in the manuscript. Also, I think all of these methods were applied somewhat earlier than the publications describing them were written (otherwise mostly documented in proceedings and technical reports) so they are not such newcomers as the paper dates might imply.

a. *Authors' Response:* We appreciate the reviewer sharing these relevant papers. Poulsen et al. (2012) was particularly illuminating with its thorough outline of the ORAC retrieval methodology and the description of forward model uncertainty. The results of Sayer et al. (2011) suggest that the multispectral retrieval of effective radius estimates effective radii deeper in the cloud where droplet sizes tend to be smaller. The paper concludes with an endorsement for the retrieval of vertical droplet profiles. Heidinger (2003) applied the Rodgers optimal estimation technique to retrieve effective radius and optical depth using one channel in the

visible, one in the near-infrared, and two in the infrared. Minnis et al. (2011) describes an iterative technique to retrieve cloud phase, optical depth, and effective radius from MODIS and VIIRS observations to support the CERES data products. We will highlight all of these papers in the introduction.

b. *Proposed changes to the manuscript:* We found the suggested papers relevant to our historical overview and will add them to our introduction. In addition, we will rely on the work of Poulsen et al. (2012) in section 2.3, where we will include a description of forward model uncertainties.

4. Line 106: I see there is a paper reference there but for ease it would be good to detail the expected pixel size, orbital geometry, swath width, and spectral sampling/bandwidth of the CPF mission as well. This should be recapped in the conclusion as well, where relevant (e.g. in the discussion of scales of variability in marine stratocumulus clouds).

a. *Authors' Response:* We agree.

b. *Proposed changes to the manuscript:* Spectral sampling and resolution, orbital geometry, spatial resolution, and swath width will be added to line 108. We will also include the CPF spatial sampling and swath width information in Section 4.1 to provide context to our discussion on comparing sampling volumes between in situ and remote measurements. Lastly, we updated Figure 6 to include an additional histogram with a length scale of $0.5km$, the spatial sampling of the CPF instrument at nadir. While the result is similar to the 1 $km$ spatial sampling of MODIS when looking nadir, we found it useful to show that, as pixel size decreases, the average variability of effective radius with respect to the horizontal plane decreases.

5. Section 2.1: I would suggest renaming this "the bispectral method" instead of "the standard method". What does "standard" mean? From a polar-orbiting viewpoint, yes, this method has been applied routinely to MODIS and VIIRS. But that in my view implies it's the only way things are done, despite e.g. the ATSR, AVHRR, CERES references I provided which have similar (or longer) time series of data.

a. *Authors' Response:* We agree and will adopt "bispectral method".

b. *Proposed changes to the manuscript:* We will change section 2.1 heading to 'The bispectral method'. We will also update our phrasing throughout the paper to use 'bispectral' instead of 'standard'.

6. Line 233: In practical terms $S_\varepsilon$ tends to be used not just for measurement uncertainty but the combination of measurement plus forward model uncertainty covariance. This may be worth noting. Mathematically, it doesn't make a difference whether one puts only measurement error in $S_\varepsilon$ (in which forward model parameterization uncertainty is normally put in another matrix often called $S_b$ in Rodgers notation), or combines both measurement and forward model uncertainty. This is omitted from the equations and discussions here. See also my comment 11, which is my main issue with the paper as written.

   a. *Authors' Response:* We agree with the reviewer's comment that there should be some discussion on forward model errors. Indeed, our forward model deviates from the true nature of clouds and the atmosphere due to the many simplifications, which deserve scrutiny. The recommended paper by Poulsen et al. (2012) was particularly illuminating in this regard, thanks to its thorough discussion of forward model uncertainties.

   b. *Proposed changes to the manuscript:* We will update line 233 and section 2.3 to explicitly state that the covariance matrix, $S_\epsilon$, includes measurement and forward model uncertainty. We will define the various sources of forward model uncertainty that are relevant to our problem in section 3, citing Poulsen et al. (2012).

7. Line 260 and elsewhere: the paper often refers to the "constrained" OE approach, kind of making it seem like the constraints are unusual or an innovation. In reality though every algorithm (including OE ones) are putting in constraints similar to this (state bounds). I'm not sure that the word "constrained" needs to be emphasized in the paper very much as it makes the reader focus more on that while in my view the novel aspect is getting at radius profiles in adiabatic clouds.

   a. *Authors' Response:* We do not claim that constrained optimal estimation is an innovation of our own. We chose to repeat that phrase to emphasize the importance of the constraint. Without it, using MOIDS measurements with ~ 2% measurement uncertainty can lead to retrieved profiles where the droplet size at cloud top is smaller than cloud bottom, violating our forward model assumption.

That being said, we appreciate the reviewer's comment because we do not wish to distract readers from the more important result of retrieving droplet profiles.

    b. *Proposed changes to the manuscript:* We will remove the qualifier *constrained* from each mention of the optimal estimation method. Where the constraints are first introduced, we will adjust our wording to emphasize their importance, particularly for spectral measurements with MODIS-like uncertainty.

8. Line 276 and elsewhere: the residual/left side of $L^2$ norm is most commonly referred to as the "cost function" and often denoted capital italic $J$ in the Rodgers formalism. For ease of readers comparing different references, I think it would be good to note these notation/terminology differences somewhere around here.

    a. *Authors' Response:* We will adopt the terminology and notation that are commonly used in the retrieval community. However, we would like to point out that the left side of equation 11 in our manuscript is not the cost function in the sense outlined by Rodgers (the first two terms on the right-hand side of equation 5.3 in Rodgers, 2000) or Poulsen (equation 1 of Poulsen et al. (2012)). This is why originally defined the $L^2$-norm of the difference between the forward-modeled reflectances and the measurements as the 'residual'.

    b. *Proposed changes to the manuscript:* We will define the cost function as the $L^2$-norm of the difference between the forward-modeled reflectances and the measurements. Instead of continually referring to the $L^2$-norm throughout the paper, we will change the phrasing to either 'cost function', or simply $J$.

9. Line 286: for completeness, I'd add the equation for uncertainty estimate on the retrieved state here. Unless I missed it, it seems to not be included, and as part of the paper is talking about expected improvements from CPF I think it is worth including explicitly how this is calculated.

    a. *Authors' Response:* We agree.

    b. *Proposed changes to the manuscript:* We will add the equation for computing the posterior covariance matrix after line 286.

10. Line 335: the MODIS retrieval uncertainties used as the a priori uncertainty should be stated here, and a citation to where they came from added.

   a. *Authors' Response:* The a priori uncertainty for optical depth and effective radius at cloud top was defined as the MODIS retrieval uncertainty for optical depth and effective radius, respectively. The MODIS retrievals and their respective uncertainties vary between pixels. Therefore, there is no single number to report. For the three MODIS scenes used in our paper, the mean retrieval uncertainty for cloud effective radius over ocean with an optical depth of at least three was 10.6% ($\sim 0.89\ \mu m$). For optical depth, the mean retrieval uncertainty was 5.9% ($\sim 0.56$). These values align with the expected retrieval uncertainty of the MODIS Collection 6 cloud products (Platnick et al., 2017).

   b. *Proposed changes to the manuscript:* We will add a citation to line 335 for the retrieval uncertainty of MODIS Collection 6 cloud products (Platnick et al., 2017). We will also include the statistics mentioned above for retrieval uncertainty of cloudy pixels over ocean with an optical depth of at least three to provide readers with an idea of the values used in our analysis.

11. Sections 3 onwards: my main technical issue with the MODIS retrievals and simulated CPF uncertainties is that they are a realistic "best case" performance and this is kind of skirted over. The discussion more or less takes the only relevant uncertainty source as radiometric (sensor absolute calibration uncertainty and shot noise). Even if that were true, from my reading the calibration uncertainty is taken as spectrally independent. In reality it may be spectrally correlated (based on experiences with various space-based sensors) which affects downstream uncertainty characterization. But really, the main issue is the implicit assumption that the forward model (including its numerical implementation) is perfect which is inherently false (and semi-acknowledged by the fact the section 3 title includes "forward model assumptions"). These assumptions, as well as e.g. factors like lookup table interpolation precision, uncertainties in ancillary data (surface reflectance/albedo, gas columns), and non-calibration image artefacts (e.g. 3D radiative transfer effects, image ghosts, delayed impulse response after bright pixels), are often similar to or larger than absolute calibration uncertainty. And these can all have e.g. angular dependence and spectral covariation as well. So this is a big reason why retrievals are never as good as idealized sensitivity studies (as they rarely can take into account these factors). I understand this paper is a proof of concept and not a full operational algorithm. But I think it is necessary to acknowledge these issues seriously (I really doubt we can make our forward models good enough to take advantage of CPF's radiometric calibration quality). Otherwise it feels like it is misleadingly over-hyping the

CPF mission as folks who don't work in algorithm development may well not be aware that radiometric quality is only one of the determining factors in retrieval quality. I wonder if somehow this discussion could be tied into the existing sensitivity studies (or new sensitivity studies). Maybe this could involve comparing MODIS retrieval uncertainties with the width of contours in figures 7 and 8 – I will leave this to the authors to decide how best to respond.

a. *Authors' Response:* We acknowledge the lack of discussion on sources of forward model uncertainty. Forward model uncertainty is difficult to quantify but should not be ignored. We agree with the reviewer that our discussion in sections 4.2 and 5 should focus on total uncertainty. Minimizing forward model uncertainty leads to a measurement-limited solution that, the reviewer points out, may be unachievable with CPF measurements. Assuming a droplet profile is just one assumption that reduces forward model uncertainty because the assumption of a vertically homogeneous cloud is known to be a simplification for certain types of clouds (Platnick, 2000). In the future, an optimal estimation algorithm may be able to leverage the full spectrum of CPF to simultaneously estimate cloud phase (Pilewskie and Twomey, 1987), cloud top height (Rozanov and Kokhanovsky, 2004), above-cloud column water vapor (Albert et al., 2001), $CO_2$ column amount (Buchwitz and Burrows, 2004), and aerosol optical depth (Mauceri et al., 2019) , reducing forward model uncertainty by limiting the number of assumptions.

We assumed the radiometric uncertainty of the instrument was uncorrelated, and the reviewer is correct in noting that this is not the best representation of real space-based spectrometers. Kopp et al. (2017) computed the relative total radiometric uncertainty for the CPF instrument, HySICS, as a function of spectral channel for bright (cloud-filled) Earth viewing scenes. Flat field uncertainty dominates at short wavelengths, while shot noise and brightness offset dominate at longer wavelengths. For each channel, the total relative uncertainty appears to strongly covary with neighboring channels (Kopp et al., 2017). Future iterations of this work will leverage these findings to define the off-diagonal elements of the measurement covariance matrix. That said, we will emphasize that the assumption of uncorrelated measurement uncertainty between spectral channels is a simplification of the true instrument.

Lastly, we do not want the framing of our results to overstate our findings. We will adjust the wording in sections 4.2 and 5 to provide the necessary context for our results. We appreciate the reviewer's suggestion to include a discussion on how our multi-spectral retrieval uncertainty compares with the well-documented MODIS Collection 6 effective radius retrieval uncertainty (Platnick et al., 2017). We will incorporate this into section 4.2.

b. *Proposed changes to the manuscript:* We will update section 3 to include a description on sources of forward model uncertainty, following previous work by Poulsen et al. (2012). In section 4.2, we will adjust the uncertainty added to the simulated TOA reflectance spectra to include both measurement and forward model uncertainty. Instead of explicitly estimating the uncertainty of each source within the forward model, we leverage previous work by Watts et al. (1998) and Platnick et al. (2017) to describe the fraction of the total uncertainty due to forward model uncertainty. We also make it clear that forward model uncertainty can never be reduced entirely. Additionally, we will expand section 4.2 with a comparison of our multi-spectral retrieval uncertainty estimate using simulated CPF TOA reflectances with the MODIS collection 6 cloud products retrieval uncertainty.

**References**

Albert, P., Bennartz, R., and Fischer, J.: Remote Sensing of Atmospheric Water Vapor from Backscattered Sunlight in Cloudy Atmospheres, 2001.

Bohren, C. F. and Clothiaux, E. E.: Fundamentals of Atmospheric Radiation, Wiley-VCH, Darmstadt, 2006.

Buchwitz, M. and Burrows, J. P.: Retrieval of CH4, CO, and CO2 total column amounts from SCIAMACHY near-infrared nadir spectra: retrieval algorithm and first results, in: Remote Sensing of Clouds and the Atmosphere VIII, Remote Sensing of Clouds and the Atmosphere VIII, 375–388, https://doi.org/10.1117/12.514219, 2004.

Heidinger, A. K.: Rapid Daytime Estimation of Cloud Properties over a Large Area from Radiance Distributions, 2003.

King, N. J. and Vaughan, G.: Using passive remote sensing to retrieve the vertical variation of cloud droplet size in marine stratocumulus: An assessment of information content and the potential for improved retrievals from hyperspectral measurements, J. Geophys. Res. Atmospheres, 117, https://doi.org/10.1029/2012JD017896, 2012.

Kopp, G., Smith, P., Belting, C., Castleman, Z., Drake, G., Espejo, J., Heuerman, K., Lanzi, J., and Stuchlik, D.: Radiometric flight results from the HyperSpectral Imager for Climate Science (HySICS), Geosci. Instrum. Methods Data Syst., 6, 169–191, https://doi.org/10.5194/gi-6-169-2017, 2017.

Mauceri, S., Kindel, B., Massie, S., and Pilewskie, P.: Neural network for aerosol retrieval from hyperspectral imagery, Atmospheric Meas. Tech., 12, 6017–6036, https://doi.org/10.5194/amt-12-6017-2019, 2019.

Minnis, P., Sun-Mack, S., Young, D. F., Heck, P. W., Garber, D. P., Chen, Y., Spangenberg, D. A., Arduini, R. F., Trepte, Q. Z., Smith, W. L., Ayers, J. K., Gibson, S. C., Miller, W. F., Hong, G., Chakrapani, V.,

Takano, Y., Liou, K.-N., Xie, Y., and Yang, P.: CERES Edition-2 Cloud Property Retrievals Using TRMM VIRS and Terra and Aqua MODIS Data—Part I: Algorithms, IEEE Trans. Geosci. Remote Sens., 49, 4374–4400, https://doi.org/10.1109/TGRS.2011.2144601, 2011.

Nakajima, T. Y., Suzuki, K., and Stephens, G. L.: Droplet growth in warm water clouds observed by the A-Train. Part I: Sensitivity analysis of the MODIS-derived cloud droplet sizes, J. Atmospheric Sci., 67, 1884–1896, https://doi.org/10.1175/2009JAS3280.1, 2010.

Painemal, D. and Zuidema, P.: Assessment of MODIS cloud effective radius and optical thickness retrievals over the Southeast Pacific with VOCALS-REx in situ measurements, J. Geophys. Res. Atmospheres, 116, 1–16, https://doi.org/10.1029/2011JD016155, 2011.

Pilewskie, P. and Twomey, S.: Cloud Phase Discrimination by Reflectance Measurements near 1.6 and 2.2 μm, J. Atmospheric Sci., 44, 3419–3420, 1987.

Platnick, S.: Vertical photon transport in cloud remote sensing problems, J. Geophys. Res. Atmospheres, 105, 22919–22935, https://doi.org/10.1029/2000JD900333, 2000.

Platnick, S., Meyer, K. G., King, M. D., Wind, G., Amarasinghe, N., Marchant, B., Arnold, G. T., Zhang, Z., Hubanks, P. A., Holz, R. E., Yang, P., Ridgway, W. L., and Riedi, J.: The MODIS Cloud Optical and Microphysical Products: Collection 6 Updates and Examples from Terra and Aqua, IEEE Trans. Geosci. Remote Sens., 55, 502–525, https://doi.org/10.1109/TGRS.2016.2610522, 2017.

Poulsen, C. A., Siddans, R., Thomas, G. E., Sayer, A. M., Grainger, R. G., Campmany, E., Dean, S. M., Arnold, C., and Watts, P. D.: Cloud retrievals from satellite data using optimal estimation: evaluation and application to ATSR, Atmospheric Meas. Tech., 5, 1889–1910, https://doi.org/10.5194/amt-5-1889-2012, 2012.

Rodgers, C. D.: Inverse Methods for Atmospheric Souding: Theory and Practice, 2000.

Rozanov, V. V. and Kokhanovsky, A. A.: Semianalytical cloud retrieval algorithm as applied to the cloud top altitude and the cloud geometrical thickness determination from top-of-atmosphere reflectance measurements in the oxygen A band, J. Geophys. Res. Atmospheres, 109, https://doi.org/10.1029/2003JD004104, 2004.

Sayer, A. M., Poulsen, C. A., Arnold, C., Campmany, E., Dean, S., Ewen, G. B. L., Grainger, R. G., Lawrence, B. N., Siddans, R., Thomas, G. E., and Watts, P. D.: Global retrieval of ATSR cloud parameters and evaluation (GRAPE): dataset assessment, Atmospheric Chem. Phys., 11, 3913–3936, https://doi.org/10.5194/acp-11-3913-2011, 2011.

Watts, P. D., Mutlow, C. T., Baran, A. J., and Zavody, A. M.: Study on Cloud Properties derived from Meteosat Second Generation Observations, Rutherford Appleton Lab, 1998.

---

## Author Comment (AC3)

**Authors' response to comments from reviewer 3**

Andrew J. Buggee & Peter Pilewskie

May 2025

We thank the reviewer for their thorough reading of our paper and for providing thoughtful comments. We have addressed each one below.

**Reviewer's General Comments**

1. This is a review on the manuscript entitled "Retrieving Vertical Profiles of Cloud Droplet Effective Radius using Multispectral Measurements from MODIS: Examples and Limitations" which deals with the retrieval of vertical cloud droplet effective radius profiles from multispectral measurements based on an adiabatic assumption. First, MODIS measurements are used and the retrieval results are compared to in-situ measurements from the VOCALS-REx campaign. A theoretical study then discusses the implications of the usage of more spectral measurements and measurements with a lower radiometric uncertainty.

2. In general, the paper is well written, mostly clear and good to understand. It also fits well into the scope of AMT and shows nicely how a reduced measurement uncertainty could improve the retrieval of cloud effective radii profiles from space. However, as already stated by the other two reviewers, I also think that the limitations of the retrieval method besides the measurement uncertainty should be discussed further before publication, in particular since this is also subject to the title. Hereby, I am missing the discussion on 3-D radiative transfer and partial cloud cover effects as well, which are known to impact spectral retrievals assuming 1-D plane parallel clouds.

   a. *Authors' response:* We agree that 3-D radiative effects and sub-pixel inhomogeneity should be discussed. See our responses to comments 1 and 2 from Reviewer 2 (Dr. Zhibo Zhang).

   b. *Proposed changes to the manuscript:* We will expand our Discussions and Conclusions section to include a review of previous work, such as Zhang and Platnick (2011) and Zhang et al. (2012), to highlight the drawbacks of using 1-D

vertical cloud profiles and the implications of ignoring horizontal variation in cloud structure. We will discuss future work involving LES-generated cloud fields and 'step-function' cloud fields to more accurately simulate horizontal and vertical cloud inhomogeneity. We will outline the limitations of our method when applied to horizontally heterogeneous cloud fields. A thorough discussion of the biases introduced by sub-pixel inhomogeneity, along with expected impacts on our droplet profile retrieval and future work to mitigate these effects, will be included. This discussion is supported by previous results from Zhang et al. (2016).

3. In addition to that, I would ask the authors to discuss the implications of the constraints made for the optimal estimation method in more detail and further that the "validation" of the retrieval using in-situ measurements is only valid in an idealized world in which the cloud conditions match the ones supported by the retrieval. Particularly, constraining the effective radius at cloud top to be larger than the one at the bottom and both to values smaller than 25 µm limits the retrieval to clouds which do not contain any precipitation formation. Precipitation formation occurs throughout the cloud, and is hence specifically relevant for the here presented retrieval of the vertical cloud effective radius profile. Further, even the cloud top radius can be influenced by drizzle formation. For example, Pörtge et al. (2023) found cloud top effective radii larger than 25 µm for a stratocumulus cloud while simultaneous radar measurements showed precipitating droplets. In addition, the here presented method is also based on the assumption of a relatively narrow monomodal droplet size distribution as it is common in the field. However, the presence of drizzle will lead to the formation of a tail in the distribution (e.g. Zinner et al., 2010; Zhang et al., 2012), which might be an additional factor limiting the retrieval and should be discussed and pointed out more clearly in the conclusions. After consideration of those aspects, I would recommend the publication of the paper.

   a. *Authors' response:* We agree with the reviewer that the retrieval method we developed is only valid for non-precipitating clouds with droplet size increasing from cloud base to cloud top. These two constraints were made for different reasons. For the case of non-precipitating clouds, we are currently limited to retrieving droplet sizes up to 25 $\mu m$ because this is the upper limit of the table of Mie calculations for liquid water droplets provided by libRadtran. To compute radiance for cloudy scenes, libRadtran utilizes a pre-computed table to convert cloud properties into optical properties. The next iteration of this work will use an expanded lookup table that ranges up to 50 microns.

   We assumed droplet size at cloud top was larger than at cloud base because both in-situ measurements and parcel theory show this to be true for non-precipitating stratocumulus clouds. At every iteration, the forward model

computes the top-of-atmosphere reflectance for a cloudy scene with some vertical droplet profile. Since we retrieve only two values from this vertical profile, at cloud top and cloud base, we must make an assumption about how droplets change in size as a function of height. Future iterations of this work will investigate different assumptions, such as sub-adiabatic and linear growth (Platnick, 2000).

Remote measurements of polarized cloud reflectance are often used to retrieve the effective variance of the droplet distribution (Meyer et al., 2025; Pörtge et al., 2023). Since we investigated the retrieval of droplet profiles from measurements taken by MODIS, which measures unpolarized reflectance, we were unable to retrieve the effective variance. Thus, we assumed a standard value used in the community. We agree with the reviewer that this may introduce forward model error if the cloud under observation has a multimodal droplet distribution, such as the example in Pörtge et al. (2023). We will discuss the implications of this in our manuscript, however, multiple studies have shown that the presence of drizzle has only modest impacts on the retrieval of effective radius at various wavelengths (Painemal and Zuidema, 2011; Zhang et al., 2012; Zinner et al., 2010).

b. *Proposed changes to the manuscript:* We will update section 3 to include a discussion on the limited scenarios viable for retrieval with our constraints of non-precipitating adiabatic clouds with droplet sizes less than 25 $\mu m$. We will explain that these constraints stem from using the precomputed table of Mie calculations for liquid water droplets provided by libRadtran, which extends up to 25 microns. Therefore, we were limited to retrieving droplet profiles from non-precipitating clouds only. The next iteration of this work will use precomputed table of Mie calculations for liquid water droplets up to 50 $\mu m$. We note that while drizzle may be present in non-precipitating clouds, several studies have shown the impact on the retrieval of effective radius to be minor. We will also update the Discussion and Conclusions section of the paper to address our choice of an adiabatic droplet profile. Retrieving just two droplet sizes from seven MODIS spectral reflectance measurements is consistent with the results of Platnick (2000), who showed that using three MODIS spectral channels similar to the ones we used led to two unique pieces of information.

Reviewer's Specific Comments

1. L. 122f./Sec. 4.2: I was wondering why the authors did not simulate CPF spectra directly instead of the EMIT spectra? Since this part is a purely theoretical study, I think one could have used the CPF specifications directly to demonstrate how the smaller radiometric uncertainty and the usage of more spectral channels influences the solution space.

   a. *Authors' Response:* The EMIT spectral response functions are freely available, thus we can simulate TOA reflectance spectra without any guesswork. However, after your suggestion, we reached out to the instrument team that developed HySICS (the HyperSpectral Imager for Climate Science) for the CPF mission and asked if we could use the spectral response functions to simulate CPF sampled TOA reflectance.

   b. *Proposed changes to the manuscript:* We were given access to the HySICS spectral response functions and we will update the analysis in section 4.2 such that the TOA reflectance spectra used to generate the contour plots in Figures 7 and 8 will simulate the HySICS spectral channels. We will no longer use simulated EMIT measurements.

2. L. 106f.: In agreement to the first referee, I also think that it would be valuable to introduce the CPF instrument in more detail. In particular, if I have not overseen anything, I am missing the number of spectral channels and the horizontal resolution, please add if possible. And is there a spectral dependence of the measurement uncertainty? If so, the implications for the retrieval should also be addressed in the discussion.

   a. *Authors' Response:* We agree.

   b. *Proposed changes to the manuscript:* Spectral sampling and resolution, orbital geometry, spatial resolution, and swath width will be added to line 108. We will also include the CPF spatial sampling and swath width information in Section 4.1 to provide context to our discussion on comparing sampling volumes between in situ and remote measurements. Lastly, we updated Figure 6 to include an additional histogram with a length scale of $0.5 km$, the spatial sampling of the CPF instrument at nadir. While the result is similar to the $1\ km$ spatial sampling of MODIS when looking nadir, we found it useful to show that, as pixel size decreases, the average variability of effective radius with respect to the horizontal plane decreases. The measurement uncertainty of HySICS, the primary CPF instrument, was reported in Kopp et al. (2017), and showed that neighboring

spectral channels strongly covary with one another. The Discussions and Conclusions section will review the implications of simplifying our model by assuming uncorrelated spectral noise.

3. L. 253: Are the partial derivative fractions presented only valid for the MODIS measurement uncertainty? And are they valid for all wavelengths? Please clarify in the manuscript.

   a. *Authors' Response:* The partial derivative fractions were derived using MODIS measurements. Through trial and error, we determined a set of fractions that would often exceed the MODIS measurement uncertainty of about 2%. We had to strike a balance between precisely estimating the Jacobian, defined as the rate of change of reflectance with respect to some infinitesimal change in one of the state variables, and the measurement uncertainty. For example, we found if $\Delta r_{bot}$ was too small, then $\Delta R(\vec{x}, \lambda_i)$ was small and measurement uncertainty dominated. If $\Delta r_{bot}$ was too large, we no longer accurately estimated the local slope. These fractions were used for all seven spectral channels because the MODIS measurement uncertainty at these channels is roughly constant. Adjustments should be made for use with other instruments. Lower radiometric uncertainty enables the detection of smaller changes in reflectance, which implies that the partial derivatives of the Jacobian can be estimated more precisely.

   b. *Proposed changes to the manuscript:* In section 2.3, we will expand on our description of how the Jacobian is estimated. The percentages listed were used for each spectral channel because the MODIS measurement uncertainty was nearly identical for the channels used. We will also make clear that these percentages must be adjusted when applying this retrieval method to measurements from instruments with different radiometric uncertainty than MODIS.

4. Table 1: In my opinion, it would be very valuable to add the resolution of each MODIS channel and the respective measurement uncertainty here.

   a. *Authors' Response:* We agree.

   b. *Proposed changes to the manuscript:* The table will be updated to include the spectral resolution and measurement uncertainty.

5. L. 323f.: Where do you see the shapes of the distributions from?

a. *Authors' Response:* We normalized the vertical dimension and discretized it into 30 bins for each in-situ profile. After doing this for all in-situ measurements without drizzle or precipitation, we found the median value for each bin, which is what we plotted in Fig. 2. Within each bin, we fit a distribution to the data. We found that for effective radius and liquid water content, a log-normal distribution best fit the measurements for most of the vertical bins, whereas a normal distribution was the best fit for number concentration for most of the bins. This explains why the shading in Fig. 2, which represents the average deviation from the median value, is symmetric for number concentration and asymmetric for the effective radius and liquid water content.

b. *Proposed changes to the manuscript:* In section 3, we will expand on our explanation of Figure 2 to describe how these distribution fits were made.

6. L. 336f.: Perhaps there might be something which I did not understand correctly, but why are you using the 2.13 µm weighting function for the cloud bottom here? To my understanding of Fig. 1, the effective radius derived from that one corresponds to the smallest optical thickness of all channels considered?

a. *Authors' Response:* Thank you for bringing this up, indicating the need for a more detailed explanation in this section. The a priori value for the radius at cloud top was defined as the retrieved effective radius using MODIS measurements, and the a priori uncertainty as the associated retrieval uncertainty. The a priori value for the radius at cloud bottom is defined as 70% of the retrieved effective radius (the a priori value for cloud top radius). This percentage was derived from the median vertical profile of effective radius from the in situ measurements shown in Fig. 1, which shows that the median value of cloud bottom radius was 70% the value of the median cloud top effective radius. The bi-spectral retrieval of effective radius was computed using two MODIS channels centered at 645 $nm$ and 2.13 $µm$, but is predominantly determined from the near-infrared wavelength. As the reviewer points out, most photons at 2.13 $µm$ scatter near cloud top. Therefore, we need to express a higher uncertainty for the a priori value for the radius at cloud bottom. We used the 2.13 $µm$ weighting function to determine that the portion of the total measured signal with a maximum penetration depth within the upper quartile of the cloud was six times that of the portion with a maximum penetration depth within the lower quartile. This is why we defined the a priori uncertainty for the radius at cloud bottom to be 6 times the value of cloud top.

b. *Proposed changes to the manuscript:* In section 3, we will expand our discussion on the a priori values and their uncertainties by outlining how each value is

defined, including a clearer description of the use of the bi-spectral retrieval of effective radius from the MODIS collection 6 cloud products to define the a priori at cloud top and bottom. We will also explain why the weighting function for 2.13 $\mu m$ was used to determine the a priori uncertainty for the cloud bottom radius.

7. L. 375: Are the optical depths stated here derived from the retrieval? Please clarify, where those are derived from.

    a. *Authors' Response:* Those values were derived from the in-situ measurements.

    b. *Proposed changes to the manuscript:* The sentence has been rewritten with the source of the optical depth estimates made clear.

8. Fig. 3: In my opinion, it would be nice to have the corresponding MODIS pictures and an indication where the measurements took place in addition to the profiles. This would give the reader an overall impression of the cloud situation and scenery. Moreover, the measurement times would be interesting to know for the solar geometry for which the comparisons have been made. And how long did it take for the aircraft to sample the profiles, what was the flight distance/spatial coverage of the in-situ measurements?

    a. *Authors' Response:* This is a good suggestion that would provide further context for the results. Instead of including the RGB images of each MODIS scene, we will report the inhomogeneity index, along with the latitude and longitude of the measurements, within a new table that provides information on the MODIS and associated VOCALS-REx measurements used in our analysis.

    b. *Proposed changes to the manuscript:* We included a new table that outlines the time of each MODIS observation, the start and end times of the overlapping VOCALS-REx measurements, the time difference, the geographic location of the MODIS observations, and the sub-pixel inhomogeneity index.

| Figure | MODIS Observation time (UTC) | MODIS Observation latitude and longitude | MODIS Sub-pixel inhomogeneity index $H_\sigma$ | VOCALS-REx in-situ start time (UTC) | VOCALS-REx in-situ end time (UTC) | Time difference (min) |
|---|---|---|---|---|---|---|
| 3a | Nov 11 2008 18:54:28 | -24.0986, -75.0013 | 0.09 | 18:45:20 | 18:45:50 | 8.88 |
| 3b | Nov 11 2008 14:42:29 | -22.8188, -73.0008 | 0.07 | 14:40:59 | 14:41:38 | 1.18 |
| 3c | Nov 9 2008 14:30:20 | -22.8970, -73.0036 | 0.08 | 14:33:33 | 14:34:23 | 3.62 |

9. L. 400f.: One common issue of the bispectral retrieval is the overestimation of the effective radius due to 3-D cloud radiative effects and broken cloudiness. Could this be a reason for the effective radius profile showing larger values than the in-situ measurements? Here, it would also help to have a visualization of the cloud scenery.

   a. *Authors' Response:* We agree that 3-D radiative effects and sub-pixel inhomogeneity should be discussed. See our responses to comments 1 and 2 from Reviewer 2 (Dr. Zhibo Zhang). For the three cases used in our manuscript to retrieve droplet profiles (Figure 3), all had an inhomogeneity index of less than 0.1. According to Zhang and Platnick (2011), these values represent fairly homogeneous clouds and 3-D radiative effects are expected to be insignificant.

   b. *Proposed changes to the manuscript:* We will expand the Discussions and Conclusions section to review how 3-D radiative effects and broken cloudiness impact the retrieval of effective radius. We will expand section 4.1 to review the historical precedent of remote retrievals overestimating in situ measurements and discuss possible sources of this bias.

10. Fig. 5: Please make a comment on the two spikes which are very pronounced in the blue line. Where do they come from? I suspect that they also influence the derived standard deviation and calculated range quite a lot.

   a. *Authors' Response:* The reviewer is correct in pointing out that these two outliers affected the range and standard deviation of this horizontal profile. The in-situ measurements show decreases in the total droplet number concentration at the same moment the effective radius rapidly increases. Likely, these regions are associated with a shift in the droplet distribution towards larger droplets. Below is a figure showing the liquid water content, effective radius, and number concentration for the three horizontal legs shown in Figure 5 of our manuscript.

[Figure]

b. *Proposed changes to the manuscript:* We will replace Figure 5 with the version above that provides additional information on liquid water content and number concentration for three representative profiles. We also included an explanation for the sharp increase in effective radius shown in the blue curve above.

11. L. 484: What is the exact definition of "time difference" here? I guess the vertical profiles were sampled over some time as well, so when did MODIS pass over the scene and between which times were the profiles measured? Please clarify in the manuscript.

a. *Authors' Response:* For each vertically sampled in situ measurement with a start and end time, we used the temporal halfway point to compute the time difference with the MODIS measurement. After reading the reviewer's comment, we revisited this calculation and learned that we could estimate the time difference more precisely by using the MODIS metadata to estimate the time each MODIS pixel within a swath recorded its measurement (thanks to Dr. Larry DiGiorlamo and Dr. Guangyu Zhao). We have updated the time differences using this more precise calculation in the table above (comment 9).

b. *Proposed changes to the manuscript:* Section 4.1 will include a more thorough description of how the time difference between a MODIS measurement and the corresponding VOCALS-REx in-situ sampling is defined.

12. L. 39: "and has been used to verify …"

   a. *Authors' Response:* Thanks for finding this mistake!

   b. *Proposed changes to the manuscript:* This has been fixed.

13. L. 270: "scalar"

   a. *Authors' Response:* Thanks for finding this mistake!

   b. *Proposed changes to the manuscript:* This has been fixed.

14. L. 357: "shown"

   a. *Authors' Response:* Thanks for finding this mistake!

   b. *Proposed changes to the manuscript:* This has been fixed.

**References**

Kopp, G., Smith, P., Belting, C., Castleman, Z., Drake, G., Espejo, J., Heuerman, K., Lanzi, J., and Stuchlik, D.: Radiometric flight results from the HyperSpectral Imager for Climate Science (HySICS), Geoscientific Instrumentation, Methods and Data Systems, 6, 169–191, https://doi.org/10.5194/gi-6-169-2017, 2017.

Meyer, K., Platnick, S., Arnold, G. T., Amarasinghe, N., Miller, D., Small-Griswold, J., Witte, M., Cairns, B., Gupta, S., McFarquhar, G., and O'Brien, J.: Evaluating spectral cloud effective radius retrievals from the Enhanced MODIS Airborne Simulator (eMAS) during ORACLES, Atmospheric Measurement Techniques, 18, 981–1011, https://doi.org/10.5194/amt-18-981-2025, 2025.

Painemal, D. and Zuidema, P.: Assessment of MODIS cloud effective radius and optical thickness retrievals over the Southeast Pacific with VOCALS-REx in situ measurements, Journal of Geophysical Research Atmospheres, 116, 1–16, https://doi.org/10.1029/2011JD016155, 2011.

Platnick, S.: Vertical photon transport in cloud remote sensing problems, Journal of Geophysical Research Atmospheres, 105, 22919–22935, https://doi.org/10.1029/2000JD900333, 2000.

Pörtge, V., Kölling, T., Weber, A., Volkmer, L., Emde, C., Zinner, T., Forster, L., and Mayer, B.: High-spatial-resolution retrieval of cloud droplet size distribution from polarized observations of the cloudbow, Atmos. Meas. Tech., 16, 645–667, https://doi.org/10.5194/amt-16-645-2023, 2023.

Zhang, Z. and Platnick, S.: An assessment of differences between cloud effective particle radius retrievals for marine water clouds from three MODIS spectral bands, Journal of Geophysical Research, 116, D20215, https://doi.org/10.1029/2011JD016216, 2011.

Zhang, Z., Ackerman, A. S., Feingold, G., Platnick, S., Pincus, R., and Xue, H.: Effects of cloud horizontal inhomogeneity and drizzle on remote sensing of cloud droplet effective radius: Case studies based on large-eddy simulations, Journal of Geophysical Research Atmospheres, 117, 1–18, https://doi.org/10.1029/2012JD017655, 2012.

Zinner, T., Wind, G., Platnick, S., and Ackerman, A. S.: Testing remote sensing on artificial observations: impact of drizzle and 3-D cloud structure on effective radius retrievals, Atmospheric Chemistry and Physics, 10, 9535–9549, https://doi.org/10.5194/acp-10-9535-2010, 2010.

---

## Author Comment (AC4)

**Authors' response to comments from reviewer 5**

Andrew J. Buggee & Peter Pilewskie

May 2025

We thank the reviewer for their thorough reading of our paper and for providing thoughtful comments. We have addressed each one below.

**Reviewer's General Comments**

1. This manuscript presents a study on the retrieval of vertical profiles of the effective radius for non-precipitating warm clouds using multispectral solar reflectance measurements in the visible to shortwave infrared regions. Specifically, the study aims to retrieve three parameters: cloud optical thickness, the effective radius at the cloud top, and effective radius at the cloud base.

2. This study presents two key analyses. First, the three parameters are retrieved from the seven MODIS channels using a framework based on the optimal estimation method. The vertical profiles of the effective radius reconstructed from these parameters are then compared with in-situ measurements from the VOCALS-REx campaign. The second key analysis, based on simulations, examines how increasing the number of spectral channels and reducing the radiometric uncertainties can improve the retrieval accuracy of the three parameters. This analysis is particularly relevant to the upcoming CPF instrument, which will provide hyper-spectral imaging measurements.

3. This manuscript is generally well written and falls within the scope of AMT. While several previous studies have addressed similar issues, the results presented in this manuscript, particularly those in Figures 3, 7 and 8, provide valuable contributions to the scientific community, enhancing the understanding of this topic.

4. My major concern, as with other reviewers, is the retrieval bias introduced by subpixel-scale horizontal inhomogeneity and three-dimensional radiative transfer effects. These issues should be addressed in a dedicated section. The relevant previous studies have already been sufficiently cited in other reviewers' comments. Even if the potential

retrieval bias caused by these factors is significant, discussing it should not diminish the value of this study.

    a. *Authors' Response:* We agree that 3-D radiative effects and sub-pixel inhomogeneity should be discussed. See our responses to comments 1 and 2 from Reviewer 2 (Dr. Zhibo Zhang).

    b. Proposed changes to the manuscript: We will expand our manuscript to discuss how 3-D radiative effects impact the retrieval of effective radius. We will discuss how sub-pixel inhomogeneity leads to different 3-D radiative effects, such as illumination and shadowing. The works of Zinner et al. (2010), Zhang and Platnick (2011) and Zhang et al. (2012) are crucial to this discussion as they showed the impact of 3-D radiative effects and sub-pixel inhomogeneity on retrievals of droplet size and discussed the implications for discerning cloud vertical structure. Furthermore, these studies provided thresholds on the sub-pixel inhomogeneity index, $H_\sigma$ that can be used to determine whether 3-D radiative effects need to be considered.

**Minor Comments**

1. Abstract: The abstract should explicitly state that this study focuses on "non-precipitating warm clouds".

    a. *Authors' Response:* We agree.

    b. *Proposed changes to the manuscript:* The suggested phrase was added to the abstract.

2. L15, "near-infrared": This study utilizes the MODIS 1.6 $\mu m$ and 2.1 $\mu m$ channels. While these channels are generally considered part of the near-infrared spectrum, they are often referred to as "shortwave infrared" in research involving MODIS cloud retrievals. Since "MODIS" is mentioned in the title, it would be helpful to clearly specify the wavelength range included in "near-infrared" to avoid potential confusion.

    a. *Authors' Response:* We will change the phrase "near-infrared" to "shortwave infrared" to follow the standard used in the literature.

b. *Proposed changes to the manuscript:* All uses of the phrase "near-infrared" were changed to "shortwave infrared."

3. L150-152, Sect 2.1: Is aerosol scattering and absorption being ignored, or is it excluded from the retrieval variables but still accounted for in the radiative transfer calculations?

   a. *Authors' Response:* We do not retrieve any aerosol properties, but aerosols are accounted for in the forward model. We made the same assumption as the MODIS collection 6 cloud products forward model, which includes an aerosol optical depth of 0.1for cloudy scenes over ocean (Amarasinghe et al., 2017).

   b. *Proposed changes to the manuscript:* We will add a new paragraph at the end of section 3 describing the assumed aerosol type and optical depth, the Cox-Munk bidirectional reflectance model assumptions used to account for the impact of wind speed and direction on the ocean surface, the assumed effective variance of the droplet distribution, the surface albedo, and the US 1976 standard atmosphere that provides vertical profiles of temperature, molecular oxygen, carbon dioxide, and other trace gases.

4. "adiabatic assumption" for Eqs. (3) and (4), Sect 2.2: My understanding might be incorrect, but in the case of a well-known adiabatic cloud model (e.g., Bennartz, 2007; Merk et al., 2016), all supersaturated water vapor is assumed to condense, meaning the number of free parameters is limited to two. Allowing three degrees of freedom, as in Eqs. (3) and (4), would correspond to a sub-adiabatic model, which assumes that the condensation rate is less than 100%. What is important is not the name, but the cloud microphysical reason why three independent parameters are allowed.

   a. *Authors' Response:* Our model is consistent with the Bennartz adiabatic model. Leveraging observational results, Bennartz (2007) assumed the total droplet number concentration, $N_c$, was constant with height. Updrafts move droplets toward the cloud top. Along the way, they grow via water vapor deposition, while $N_c$ remains fixed. Therefore, the total liquid water is distributed over the same number of droplets at any height within the cloud (Bennartz, 2007). Liquid water content is linear with respect to the geometric height: $LWC(z) = c_w z$ where $c_w$ is the condensation rate and $z$ is the geometric height within the cloud (Bennartz, 2007). Bennartz (2007) concludes that the volume-averaged cloud droplet radius depends on the total number concentration and the liquid water content rather than the exact shape

of the droplet distribution: $r_v(z) = \left(\frac{3\,c_w z}{4\,\pi N_c \rho}\right)^{1/3}$. To show our model is
consistent, we start with Eq. (3) from our manuscript and assume the liquid
water content at cloud base is 0, therefore eliminating the $y$-intercept of the
linear equation: $LWC(z) = LWC(H)\frac{z}{H}$, where $H$ is the height at cloud top.
Using this equation, the effective radius profile (Eq. (4)) is now: $r_e(z) =$
$\left(\frac{3}{4\pi N_c \rho}\,LWC(H)\frac{z}{H}\right)^{1/3}$. The condensation rate, with units of $\frac{g}{m^4}$, depends on
temperature and, to a lesser degree, pressure (Rausch et al., 2017). Marine
stratus clouds tend to be shallow. Therefore, we can assume temperature is
constant over the vertical extent of the cloud and, ignoring the pressure
dependence, define the condensation rate as: $c_w = \frac{LWC(H)}{H}$ (Rausch et al.,
2017). Making this substitution, our equation for the effective radius becomes:
$r_e(z) = \left(\frac{3\,c_w z}{4\pi N_c \rho}\right)^{1/3}$, identical to the equation for the volume-averaged radius
in the Bennartz model. The only difference between our two models is that we
account for a non-zero liquid water content at cloud base. We do this because
we found the in-situ measurements of effective radius vary rapidly while
liquid water content (and total number concentration) values are small ($<$
$0.03\,\frac{g}{m^3}$). This is why we defined the cloud base as the altitude where the
liquid water content exceeds $0.03\,\frac{g}{m^3}$ and the total number concentration
exceeds $1\,m^{-3}$.

b. *Proposed changes to the manuscript:* We will mention that our adiabatic
model is identical to the Bennartz model but with a non-zero liquid water
content value at cloud base. We will also explain why a non-zero liquid water
content value is required.

5. L294: Is "the distribution width parameter, $\alpha$" a parameter of the gamma distribution?"

a. *Authors' Response:* Yes. We can add 'gamma' to line 294 to avoid confusion.
Additionally, the alpha parameter is related to the more commonly used
effective variance by: $\alpha = \frac{1}{v_{eff}} - 3$ (Emde et al., 2016) We will include this
definition in the paper and report the values used in the more familiar effective
variance term.

b. *Proposed changes to the manuscript:* We will update this sentence to: "For all
simulations shown, the gamma distribution width parameter, α, was set to 10

based on analysis of in situ measurements of non-precipitating marine stratocumulus clouds from the VOCALS-REx flight campaign."

6. L296, "Cloud geometric thickness was set to 0.5 km": The setting is acceptable in radiative calculations. However, in the (sub)adiabatic cloud models, the cloud geometric thickness H should be determined uniquely from the set of $\tau_c$, $r_{top}$, and $r_{bot}$.

   a. *Authors' Response:* Using the retrievals of $\tau_c$, $r_{top}$, and $r_{bot}$, we estimate liquid water path by assuming the total number concentration, $N_c$, is constant with height. Furthermore, at any height within the cloud, there is a monodispersed distribution represented by $r_e(z)$. Since the effective radius depends on the altitude within cloud, the liquid water path is $LWP = \frac{4}{3}\pi\rho N_c \int r_e^3(z)\,dz$. Assuming $\frac{2\pi r_e}{\lambda} \gg 1$: $\tau_c^\lambda = 2\pi N_c \int r_e^2(z)\,dz$. To estimate liquid water path, we solve for the total number concentration using the equation for optical depth and plug this into the equation for liquid water path: $LWP = \frac{2}{3}\rho\tau_c \frac{\int r_e^3(z)\,dz}{\int r_e^2(z)\,dz}$. Bennartz (2007) assumed an adiabatic cloud with an effective radius and liquid water content of 0 at cloud base to drive an equation for the cloud thickness: $H = \left(\frac{2\,LWP}{c_w C_F}\right)^{1/2}$ where $c_w$ is the condensation rate and $C_F$ is the cloud fraction. We could use our estimate of LWP to estimate the geometric thickness by assuming some cloud temperature to estimate $c_w$, however, our forward model used to retrieve $\tau_c$, $r_{top}$, and $r_{bot}$ assumed a cloud geometric thickness of 0.5 $km$. We expect an estimate of the cloud thickness to be close to our forward model assumption. Furthermore, the geometric thickness is not a radiatively relevant quantity. We found little change to the simulated TOA reflectance when using cloud geometric thicknesses of 0.5 and 1 $km$.

   b. *Proposed changes to the manuscript:* None.

7. L331, "the first seven spectral channels of MODIS": Are the response functions of these channels taken into account in the forward calculation?

   a. *Authors' Response:* Yes. We will include a sentence clarifying the source of the MODIS spectral response functions used to simulate top-of-atmosphere reflectance.

b. *Proposed changes to the manuscript:* We will add a sentence to section 3 that describes the source of the MODIS spectral response functions used to simulate top-of-atmosphere reflectance.

8. L332, "because they deliberately avoid water vapor absorption, simplifying the forward model": Are water vapor absorption and Rayleigh scattering taken into account in the forward calculation?

   a. *Authors' Response:* Yes, both water vapor absorption and molecular scattering are included within the forward model. The purpose of line 332 was to emphasize that the spectral channels used in our multispectral retrieval were chosen in part because, at those wavelengths, the bulk absorption coefficients for water vapor are negligible.

   b. *Proposed changes to the manuscript:* We will more detail on our forward model assumptions (see response to comment 3).

9. L376: Is it correct that 0.55 μm is being used?

   a. *Authors' Response:* No! Thank you for catching this mistake.

   b. *Proposed changes to the manuscript:* Line 376 has been updated to reflect the correct MODIS visible channel used in the bi-spectral estimate of $r_e$ and $\tau_c$, which was 0.65 $\mu m$.

10. Figure 3: To verify horizontal inhomogeneity, it would be preferable to include the corresponding RGB images for these MODIS retrievals. At the very least, the latitude and longitude of the MODIS retrievals should be provided, allowing readers to check the images themselves.

    a. *Authors' Response:* We agree that a discussion on horizontal inhomogeneity is needed. Instead of including the RGB images of each MODIS scene, we will report the inhomogeneity index, along with the latitude and longitude of the measurements, within a new table that provides information on the MODIS and associated VOCALS-REx measurements used in our analysis. Zhang and Platnick (2011) showed that retrievals of effective radius are biased from 3-D radiative effects such as illumination and shadowing when the cloud under observation has an inhomogeneity index greater than 0.3. This results in a significant difference between retrieved effective radii using 2.1$\mu m$ and 3.7$\mu m$ shortwave infrared measurements. For the three cases used in our

manuscript to retrieve droplet profiles (Figure 3), all had an inhomogeneity index of less than 0.1. According to Zhang and Platnick (2011), these values represent fairly homogeneous clouds and 3-D radiative effects are expected to be insignificant.

b. *Proposed changes to the manuscript:* We will include a new table that outlines the time of each MODIS observation, the start and end times of the overlapping VOCALS-REx measurements, the time difference, the geographic location of the MODIS observations, and the sub-pixel inhomogeneity index.

| Figure | MODIS Observation time (UTC) | MODIS Observation latitude and longitude | MODIS Sub-pixel inhomogeneity index $H_\sigma$ | VOCALS-REx in-situ start time (UTC) | VOCALS-REx in-situ end time (UTC) | Time difference (min) |
|---|---|---|---|---|---|---|
| 3a | Nov 11 2008 18:54:28 | -24.0986, -75.0013 | 0.09 | 18:45:20 | 18:45:50 | 8.88 |
| 3b | Nov 11 2008 14:42:29 | -22.8188, -73.0008 | 0.07 | 14:40:59 | 14:41:38 | 1.18 |
| 3c | Nov 9 2008 14:30:20 | -22.8970, -73.0036 | 0.08 | 14:33:33 | 14:34:23 | 3.62 |

11. Figure 3: I recommend also showing the other effective radius ($r_{e,1.6}$) retrieved using 1.6 μm instead of 2.1 μm ($r_{e,2.1}$), which is included in MOD06, in Figure 3. $r_{e,1.6}$ may be able to sense the cloud particle size in a deeper depth than $r_{e,2.1}$.

a. *Authors' Response:* We agree that for warm, non-precipitating, adiabatic marine stratus clouds, $r_{e,1.6}$ should be smaller than $r_{e,2.1}$ because photons at 1.6 $\mu m$ have deeper average penetration depths due to a larger single scattering albedo than photons at 2.1 $\mu m$ (Platnick, 2000). However, we showed $r_{e,2.1}$ along with our multi-spectral retrieval because this value was used for the a priori value of $r_{top}$, and for estimating the a priori value of $r_{bot}$.

b. *Proposed changes to the manuscript:* None.

12. L401-405: To investigate why the case in Figure 3b performs worse than the other two, have you considered conducting a remote sensing simulation using the VOCALS-REx in-situ measurements? That is, simulating MODIS reflectance measurements using the droplet size distribution obtained from the VOCALS-REx as input, and then retrieving $\tau_c$, $r_{top}$, and $r_{bot}$ using your algorithm.

a. *Authors' Response:* We thank the reviewer for this suggestion. We will investigate our solution to the case shown in Figure 3b by using the VOCALS-REx in-situ measurements to simulate MODIS TOA reflectances. However, as Figure 7 shows, the number of possible state vectors that lead to convergence is large when using the first seven MODIS spectral channels. With a large solution space (the area within the isopleth of one), the a priori guess strongly influences the final state vector because the iterative Gauss-Newton technique pushes each state vector along the direction of greatest change. The solution shown in Figure 3.b may suffer from a more inaccurate prior than the other two cases.

b. *Proposed changes to the manuscript:* We will update our manuscript with the findings from this suggestion.

13. Section 4.2: Why are the EMIT specifications and wavelengths used in the simulation instead of CPF?

a. *Authors' Response:* The EMIT spectral response functions are freely available. Thus, we can simulate TOA reflectance spectra without any guesswork. We used EMIT measurements as a surrogate for CPF measurements because they have a similar spectral range and resolution. However, multiple reviewers asked a similar question, so we reached out to the instrument team that developed HySICS (the HyperSpectral Imager for Climate Science), the spectral instrument on board CPF, and asked if we could obtain the spectral response functions so that we could simulate CPF-sampled TOA reflectance.

b. *Proposed changes to the manuscript:* We were given access to the HySICS spectral response functions and have altered the analysis in section 4.2 such that the TOA reflectance spectra used to generate the contour plots in Figures 7 and 8 now simulate the HySICS spectral channels. We no longer use simulated EMIT measurements.

14. Figure 7: Has it been discussed why this contour pattern appears, particularly why the uncertainties of $\tau_c$ and $r_{top} - r_{bot}$ exhibit a negatively correlated pattern?

a. *Authors' Response:* For this project, we did not investigate the reasons for the particular contour pattern found in Figure 7. In addition, we would argue that the retrieval uncertainty does not exhibit a negative correlation. Our iterative

method terminates when the relative $\ell^2$-norm difference between the forward modeled reflectances and the MODIS observations is less than one, the inner-most contour in Figure 7. The retrieval uncertainty of optical depth is the width of this contour along the $x$-axis, and the uncertainty in $r^*_{top} - r_{bot}$ is the width along the $y$-axis. A negative correlation would imply that retrieval uncertainty of $\tau_c$ decreases as the uncertainty of $r^*_{top} - r_{bot}$ grows, which does not appear to be true. Figure 7 suggests that many values for the radius at cloud bottom will lead to convergence. In addition, we can conclude that the vector normal to the contours of Figure 7, the direction of greatest change, consistently had a larger component along the $\tau_c$ axis than the $r^*_{top} - r_{bot}$.

b. *Proposed changes to the manuscript:* None.

15. L509: Please list the wavelengths of the 35 spectral channels used. It would be even better if they were presented along with the transmittance of atmospheric gases.

   a. *Authors' Response:* We agree that this would be helpful for readers.

   b. *Proposed changes to the manuscript:* We will include a plot of atmospheric transmittance over the spectral range of the CPF instrument and overlay the 35 spectral channels used in section 4.2 of our analysis.

16. L529-531, Sect. 4.2: Is assuming a radiometric uncertainty of 0.3% still reasonable, even when considering potential uncertainty in forward calculation, including uncertainties in given parameters such as gas absorption, surface albedo, and aerosols? Additionally, is this 0.3% uncertainty fairly defined in comparison to the 2% uncertainty of MODIS L1B?

   a. *Authors' Response:* We acknowledge the lack of discussion on different sources of retrieval uncertainty and agree that they should be discussed. See our response to comment 11 from reviewer 1.

   b. *Proposed changes to the manuscript:* We will update section 3 to include a description on sources of forward model uncertainty, following previous work by Poulsen et al. (2012). In section 4.2, we will adjust the uncertainty added to the simulated TOA reflectance spectra to include both measurement and forward model uncertainty. Instead of explicitly estimating the uncertainty of each source within the forward model, we leverage previous work by Watts et al. (1998) and Platnick et al. (2017) to describe the fraction of the total uncertainty due to forward model uncertainty. We also make it clear that forward model uncertainty can never be reduced entirely. Additionally, we

will expand section 4.2 with a comparison of our multi-spectral retrieval uncertainty estimate using simulated CPF TOA reflectances with the MODIS collection 6 cloud products retrieval uncertainty.

**References**

Albert, P., Bennartz, R., and Fischer, J.: Remote Sensing of Atmospheric Water Vapor from Backscattered Sunlight in Cloudy Atmospheres, 2001.

Amarasinghe, N., Platnick, S., and Meyer, K.: Overview of the MODIS Collection 6 Cloud Optical Property (MOD06) Retrieval Look-up Tables, 2017.

Bennartz, R.: Global assessment of marine boundary layer cloud droplet number concentration from satellite, Journal of Geophysical Research: Atmospheres, 112, https://doi.org/10.1029/2006JD007547, 2007.

Buchwitz, M. and Burrows, J. P.: Retrieval of CH4, CO, and CO2 total column amounts from SCIAMACHY near-infrared nadir spectra: retrieval algorithm and first results, in: Remote Sensing of Clouds and the Atmosphere VIII, Remote Sensing of Clouds and the Atmosphere VIII, 375–388, https://doi.org/10.1117/12.514219, 2004.

Mauceri, S., Kindel, B., Massie, S., and Pilewskie, P.: Neural network for aerosol retrieval from hyperspectral imagery, Atmospheric Measurement Techniques, 12, 6017–6036, https://doi.org/10.5194/amt-12-6017-2019, 2019.

Pilewskie, P. and Twomey, S.: Cloud Phase Discrimination by Reflectance Measurements near 1.6 and 2.2 µm, Journal of the Atmospheric Sciences, 44, 3419–3420, 1987.

Platnick, S.: Vertical photon transport in cloud remote sensing problems, Journal of Geophysical Research Atmospheres, 105, 22919–22935, https://doi.org/10.1029/2000JD900333, 2000.

Platnick, S., Meyer, K. G., King, M. D., Wind, G., Amarasinghe, N., Marchant, B., Arnold, G. T., Zhang, Z., Hubanks, P. A., Holz, R. E., Yang, P., Ridgway, W. L., and Riedi, J.: The MODIS Cloud Optical and Microphysical Products: Collection 6 Updates and Examples from Terra and Aqua, IEEE Transactions on Geoscience and Remote Sensing, 55, 502–525, https://doi.org/10.1109/TGRS.2016.2610522, 2017.

Poulsen, C. A., Siddans, R., Thomas, G. E., Sayer, A. M., Grainger, R. G., Campmany, E., Dean, S. M., Arnold, C., and Watts, P. D.: Cloud retrievals from satellite data using optimal estimation: evaluation and application to ATSR, Atmospheric Measurement Techniques, 5, 1889–1910, https://doi.org/10.5194/amt-5-1889-2012, 2012.

Rausch, J., Meyer, K., Bennartz, R., and Platnick, S.: Differences in liquid cloud droplet effective radius and number concentration estimates between MODIS collections 5.1 and 6 over global oceans, Atmospheric Measurement Techniques, 10, 2105–2116, https://doi.org/10.5194/amt-10-2105-2017, 2017.

Rozanov, V. V. and Kokhanovsky, A. A.: Semianalytical cloud retrieval algorithm as applied to the cloud top altitude and the cloud geometrical thickness determination from top-of-atmosphere reflectance

measurements in the oxygen A band, Journal of Geophysical Research: Atmospheres, 109, https://doi.org/10.1029/2003JD004104, 2004.

Watts, P. D., Mutlow, C. T., Baran, A. J., and Zavody, A. M.: Study on Cloud Properties derived from Meteosat Second Generation Observations, Rutherford Appleton Lab, 1998.

Zhang, Z. and Platnick, S.: An assessment of differences between cloud effective particle radius retrievals for marine water clouds from three MODIS spectral bands, Journal of Geophysical Research, 116, D20215, https://doi.org/10.1029/2011JD016216, 2011.

Zhang, Z., Ackerman, A. S., Feingold, G., Platnick, S., Pincus, R., and Xue, H.: Effects of cloud horizontal inhomogeneity and drizzle on remote sensing of cloud droplet effective radius: Case studies based on large-eddy simulations, Journal of Geophysical Research Atmospheres, 117, 1–18, https://doi.org/10.1029/2012JD017655, 2012.

---

## Author Comment (AC5)

**Author's response to comments from reviewer 4**

Andrew J. Buggee & Peter Pilewskie

May 2025

We thank the reviewer for their thorough reading of our paper and for providing thoughtful comments. We have addressed each one below.

**Reviewer's General Comments**

1.  The manuscript discusses the retrieval of vertical droplet size profiles from multispectral solar reflectance observations with high radiometric accuracy using a constrained optimal estimation inversion technique applied to MODIS observations. The study leverages VOCALS-REx field campaign data to develop the forward model constraints, which improves retrievals from, in particular, the lower optical depth levels of moderately thick liquid phase clouds. The high radiometric accuracy and spectral sampling follows from the design specifications for the upcoming CLARREO Pathfinder (CPF) instrument to be flown on the ISS in the 2026-27 timeframe. The findings highlight the value of high radiometric accuracy compared with current state of the art satellite imagers, as well as the challenges in comparing retrievals against in situ measurements in heterogenous clouds due to the profound differences in sampling volume.

2.  The study contributes to a better understanding of future cloud microphysical profile retrieval information content from solar reflectance observations and nicely expands on previous efforts. The manuscript is very well-written, successfully capturing the history of previous studies as well as appropriate details of the author's work. I characterized one comment as major but all others are minor.

**Major Comments**

Fig. 4: This is a very important figure in terms of the study findings but I was confused.

1.  I interpret the y-axis to be absolute reflectance, not relative reflectance like the accuracy specs for MODIS or CPF. If that's correct, the choice of r_bottom and dr_bottom will

scale the y-axis value Jacobians without changing the MODIS or CPF lines. If that's the case, I don't know how to interpret the results (e.g., if dr_bottom is effectively zero, all bars will be zero on the y-axis). If not the case, please elaborate.

a. *Authors' Response:* The y-axis is absolute reflectance, and indeed the y-axis values would change for different values of $r_{bot}$ and $\Delta r_{bot}$. The y-axis is the change in reflectance due to a small perturbation in the radius at cloud bottom. For the $i^{th}$ iteration and the $j^{th}$ spectral channel, we estimate the y-axis values using the following equation:

$$\Delta R(x_i, \lambda_j) = R\big((r_i^{top}, r_i^{bot} + \Delta r_i^{bot}, \tau_{c_i}), \lambda_j\big) - R\big((r_i^{top}, r_i^{bot}, \tau_{c_i}), \lambda_j\big)$$

The measurement uncertainties shown in Figure 4 for MODIS and CPF are also displayed in absolute reflectance. We used the reported radiometric uncertainties for the MODIS and CPF instruments and multiplied these percentages with the original reflectance, $R\big((r_i^{top}, r_i^{bot}, \tau_{c_i}), \lambda_j\big)$, for each spectral channel. Therefore, these curves would also change for different values of $r_{bot}$.

With all that said, we acknowledge the reviewer's point. The figure may be more useful if shown in relative values. The measurement uncertainties for MODIS and CPF are typically reported as a percentage. Therefore, it may be more useful to readers to show the percent change in reflectance along the y-axis. This has the added benefit of normalizing the change in reflectance with the initial reflectance, which depends on the current value of the radius at cloud bottom.

The following provides more details on how Figure 4 was created. We sought to use a representative cloud example, defined as having the median droplet profile found in Figure 1, except with varying optical depth. $\Delta r_{bot}$ is defined as $\Delta r_{bot} = 0.35 \, r_{bot}$ (L253). As described in the paper, this value was chosen so that the Jacobian terms with respect to the radius at cloud bottom, $\frac{\Delta R_\lambda}{\Delta r_{bot}}$, would not be dwarfed by the measurement uncertainty. However, we had to strike a balance between estimating the Jacobian, defined as the rate of change of reflectance with respect to some infinitesimal change in $r_{bot}$, with the measurement uncertainty. We found if $\Delta r_{bot}$ was too small, measurement uncertainty dominated. If $\Delta r_{bot}$ was too large, we no longer accurately estimated the local slope. We determined $\Delta r_{bot} = 0.35 \, r_{bot}$ through trial and error. The phrasing in our manuscript may be misleading because we cannot always guarantee our estimate of the Jacobian exceeds the measurement uncertainty, because reflectance depends on the state vector. We decided to use a single value that worked for a broad set of state vectors.

b. *Proposed changes to the manuscript:* We will add additional information to the description of Figure 4 that more thoroughly explains how the y-axis values and the measurement uncertainty, which are reported as absolute reflectance, are computed. We will also explain the sensitive nature of reflectance with the state vector.

2. While the text mentions the spectral dependence of the Jacobians, it's not clear which channel(s) are being used in the figure.

   a. *Authors' Response:* The channels used are listed by their center wavelengths along the x-axis of the plot. All seven channels used in the multi-spectral retrieval (Table 1) are shown. Note that they are not in sequential order according to channel number but rather increasing in the value of the center wavelength.

   b. *Proposed changes to the manuscript:* None.

3. (3) The y-axis for the Jacobians should be labeled delta reflectance, delta reflectance/drbottom, or something similar unless I'm mistaken about (1).

   c. *Authors' Response:* Yes, we agree.

   d. *Proposed changes to the manuscript:* The y-axis label will be updated to reflect the changes from comment (1).

**Minor Comments**

1. L34: "effective droplet radius" or "effective droplet absorption" is proportional to 1-ssa? While it's true that $kr_e \sim 1 - ssa$ for an absorbing wavelength, it's an ill-defined definition for $r_e$ when $ssa = unity$ (i.e., reduces to $r_e = 0/0$).

   a. *Authors' Response:* We thank the reviewer for pointing out this mistake. Our intention was to highlight the relationship between the effective radius and the fractional absorption of incident light due to multiple scattering within warm clouds. We will correct this sentence.

b. *Proposed changes to the manuscript:* We changed this sentence to say the following: "The fraction of incident light absorbed by optically thick warm clouds is proportional to the effective droplet radius over the solar spectrum."

2. L49, 60: While Twomey and Cocks (1982) provides a nice overview of the retrieval theory, a more focused retrieval study was done in the follow-up Twomey and Cocks (1989, Beitr. Phys. Atmosph.), which used 5 spectral channels simultaneously in the retrieval (not bispectral) and presented the solution space in terms of residual contour plots similar to your Figs. 7, 8. I'm not suggesting you include the following relevant historic $\tau$, $r_e$ retrievals references but just for awareness: Other airborne retrievals (Foot (1988), Rawlins and Foot (1990)); AVHRR (Arking and Childs, 1985), Platnick and Twomey (1994).

a. *Authors' Response:* We thank the reviewer for suggesting the study by Twomey and Cocks (1989) as an early example of a multispectral retrieval of effective radius and optical depth. While the paper was hard to track down, it has proved insightful. We will review all suggested papers for potential incorporation into the historical section of our manuscript.

b. *Proposed changes to the manuscript:* The introduction has been updated to include several of the suggested papers, all of which were early examples of airborne retrievals of cloud microphysics.

3. L60: suggest adding the qualifier "nearly independent from one another for optically thicker clouds …"

a. *Authors' Response:* We agree.

b. *Proposed changes to the manuscript:* The suggested qualifier has been added.

4. L64: "… radius, cloud optical depth, and various surface spectral reflectance assumptions."

a. *Authors' Response:* We appreciate the suggestion.

b. *Proposed changes to the manuscript:* We included the above phrase about surface spectral reflectance in our sentence that mentions the free parameters varied to produce look-up tables for the MODIS Collection 6 cloud products algorithm.

5. L123, Sect. 4.2: As a simulation, it doesn't make a difference for present purposes, but I'm curious why the simulations were done for EMIT spectra instead of CPF, which is mentioned prominently as the motivation for the study (including the abstract). Was it in anticipation of doing EMIT retrievals as a follow-on? It would be useful to explain the rationale.

   a. *Authors' Response:* Multiple reviewers asked a similar question, so we reached out to the instrument team that developed HySICS (the HyperSpectral Imager for Climate Science), the spectral instrument on board CPF, and asked if we could obtain the spectral response functions so that we could simulate CPF-sampled TOA reflectance.

   b. *Proposed changes to the manuscript:* We were given access to the HySICS spectral response functions and have altered the analysis in section 4.2 such that the TOA reflectance spectra used to generate the contour plots in Figures 7 and 8 now simulate the HySICS spectral channels.

6. L184: What effective variance ($v_e$) is used? The alpha "width parameter" is mentioned on L294 but would be helpful to put it in terms of $v_e$. Are the same value(s) used for all 100 layers?

   a. *Authors' Response:* The relationship between the alpha parameter and the effective variance is defined by Emde et al. (2016): $\alpha = \frac{1}{v_{eff}} - 3$. We will include this definition in the paper and report the alpha values in the more familiar effective variance form. All 100 layers use the same effective variance. This parameter could vary with the vertical dimension in future iterations of our forward model.

   b. *Proposed changes to the manuscript:* The manuscript will be updated to include the relationship between the alpha width parameter and the effective variance. We will also report the effective variance used and make clear that this is constant for all 100 layers of the discretized cloud.

7. L188, 193: Eq. 5 is an approximation, though a reasonably good one, for the retrieved re since an exact weighting function is confounded by multiple scattering. I.e., suggest "represents the approximate retrieved …"

   a. *Authors' Response:* Yes, we agree.

   b. *Proposed changes to the manuscript:* This sentence has been updated to the following: "The wavelength-dependent column-weighted retrieved effective radius is approximated by:"

8. L91: A nice summary of the previous work. Platnick (2000) also did an information content study for MODIS-like imager, including the effect of calibration uncertainty, to help understand the number of independent parameters that can be retrieved for vertical profile inversions. Hard to make apple-to-apple comparisons but do your results seem somewhat consistent? Similar question with respect Fig. 8 accuracy sensitivity.

   a. *Authors' Response:* We thank the reviewer for bringing up this question. The information content study in Platnick (2000) highlights several key points relevant to our work that should be discussed in our manuscript. The number of independent pieces of information that can be retrieved to determine a droplet profile is, at most, equal to the number of spectral channels used. In the analysis by Platnick (2000), the three retrievals of effective radius using near-infrared wavelengths of 1.6 $\mu m$, 2.1 $\mu m$ and 3.7 $\mu m$ were found to provide only two pieces of information. The reason is that the difference between the retrieved $r_{1.6\mu m}$ and $r_{2.1\mu m}$ is less than the retrieval uncertainties for each.
      We expect the retrieval uncertainty to be the same or less than that assumed by Platnick (2000) because of the increased number of spectral channels. Therefore, our results appear in line with those of Platnick (2000) because we are only retrieving two pieces of information, the effective radius at cloud top and bottom, which was deemed possible with just three wavelengths by Platnick (2000). However, we have not explicitly computed the minimum eigenvalue of the scaled covariance matrix.

   b. *Proposed changes to the manuscript:* We will expand the Discussions and Conclusions section to include a discussion of our results and their consistency with those of Platnick (2000). In addition, the historical background within the introduction has been expanded to include the information content results from Platnick (2000).

9. Fig. 1: Please try to add some contrast to the line plot colors as some are hard to distinguish (esp. for color blind readers).

   a. *Authors' Response:* We will do so.

   b. *Proposed changes to the manuscript:* The colors of the different curves in Figure 1 have been updated to increase contrast and readability for color-blind readers.

10. L253, 254: Good idea.

    a. *Authors' Response:* Thanks!

11. L295: The MODIS retrieval wouldn't correspond exactly to the upper boundary re according to Fig. 1. Likely a small difference but worth a comment.

    a. *Authors' Response:* We agree with this clarification. The retrieval of effective radius is representative of droplet size below but near the cloud top. We will clarify this in the manuscript.

    b. *Proposed changes to the manuscript:* The manuscript will be updated to include the following clarification: "The retrieved effective radius does not represent the droplet size at cloud top. As the weighting functions in Figure 1 demonstrate, the retrieval represents the droplet size near but likely below cloud top. Nevertheless, we found this was an effective value to use for the a priori at cloud top."

12. L361/Sect. 4.1: For further context on the confounding effects that uncertainties in situ probes have on retrieval validation, including sampling issues associated with vertical and horizontal heterogeneity, I suggest looking at the recent Meyer et al. ORACLES study (amt.copernicus.org/articles/18/981/2025/). The paper discusses airborne spectral retrievals compared against two in situ cloud probes (CAS, PDI) having different measurement approaches in addition to some retrieval forward model errors. Retrieval evaluation with airborne probes continues to be an inherently challenging problem for the community. Nice discussion here and in Sect. 4.1.

    a. *Authors' Response:* We thank the reviewer for sharing this paper. The nuanced discussion on comparing remote retrievals with in-situ observations focused on different aspects than our own discussion, and we will include it in our

manuscript. Consistent with other findings, Meyer et al. (2025) found remote retrievals of cloud effective radius to be larger, on average, than the coincident in-situ derived effective radii. This study attempted to reduce the differences between remote retrievals and in situ measurements by adjusting the complex index of refraction for liquid water and the effective variance of the droplet distribution within the forward radiative transfer model. This paper also has a great overview on the difficult nature of comparing remote retrievals and in-situ measurements and we will incorporate these results into our discussion in section 4.1.

b. *Proposed changes to the manuscript:* Section 4.1 will be updated to include the results of Meyer et al. (2025).

13. L377: Not sure that the cloud-top re retrievals "validate" use of the 2.1 μm MODIS bispectral retreival as a prior as much as demonstrates consistency with its use as a prior. I.e., much of the upper cloud re information content is coming from the 2.1 μm channel, regardless of which algorithm is used.

a. *Authors' Response:* We agree with the reviewer's point. We included the $2.1\mu m$ MODIS bispectral retrieval of effective radius in Figure 3 to demonstrate that this value was consistently found to be near the in-situ derived values at cloud top. This result shows it is a suitable choice for the a priori at cloud top. This is not 'validation' in the technical sense, and we will adjust our phrasing accordingly.

b. *Proposed changes to the manuscript:* Line 377 will be updated to the following: "The bi-spectral retrieval of effective radius was within range of the cloud top in situ measurement for each case, demonstrating consistency with its use as the a priori value for the radius at cloud top."

14. Fig. 3a and 3c have the same MODIS retrieval values (blue dashed lines). One must be incorrect.

a. *Authors' Response:* Thanks for catching this!

b. *Proposed changes to the manuscript:* Figure 3a has been corrected.

15. L409: I think this often gets lost on those who use gradient searches as part of inversion algorithms, especially in higher dimensional spaces. So, good to make this point, as obvious as it may seem. Is there an example solution contour plot associated with Fig. 3 that you could show to illustrate this point (i.e., similar to Figs. 7, 8)?

   a. *Authors' Response:* We thank the review for the suggestion. Figures 7 and 8 show the $\ell^2$-norm of the difference between the forward modeled reflectances and the measurements in two-dimensional space. In actuality, this 'residual space' occupies three dimensions, with a residual associated with each point in the $r_{top}$ $r_{bot}, \tau_c$ space. There is a region within this residual space that meets our convergence requirements (the area within the isopleths of 1 in Figures 7 and 8 of our manuscript). We found the gradient to be large outside the convergence region, but once inside, the gradient was quite small. Even if we allow the iterations to continue within the isopleth of one, the slopes are so small that the algorithm quickly converges at one of the local minima.

   b. *Proposed changes to the manuscript:* We will add a figure highlighting how steep the solution space is outside the convergence region and how shallow it is within the convergence region.

16. L440: suggest "… approximately 1 km2". The effective pixel shape in the across track direction suffers from the finite integration time and so has a ~2 km triangular wide spatial weighting function for most MODIS channels though a bit less so for "1 km" channels aggregated from the native 250 m (bands 1, 2) and 500 m (bands 3-7) detector arrays. That said, L462 is correct that the across track sampling is 1 km.

   a. *Authors' Response:* Thank you for this clarification!

   b. *Proposed changes to the manuscript:* The adverb "approximately" was incorporated as suggested.

17. L446: Interesting number. Thanks for making the calculation.

18. Fig. 5 caption, L454, 455, and later text/captions.: Constant altitude flight lines aren't usually considered a "profile" in airborne sampling vernacular (at least in the cloud and aerosol community). Also, elsewhere in the manuscript profiles is used, without qualification, to describer vertical sizes only so it will be a source of confusion. Try "horizontal legs" or just "legs". I realize that constant altitude across three different

clouds during the campaign may end up sampling different depths relative to cloud top and so have some vertical profile information (e.g., the yellow curve in Fig. 5).

a. *Authors' Response:* We agree with this point and want to ensure readers have a clear distinction between vertically and horizontally sampled in-situ measurements.

b. *Proposed changes to the manuscript:* The manuscript will be updated so that the term 'profile' is used only when referring to vertical in-situ measurements or a retrieved vertical profile of droplet size. The term 'horizontal leg' is used exclusively for in-situ measurements within clouds at a near-constant altitude.

19. L458: "… and 6 µm (yellow)"

a. *Authors' Response:* Thanks for catching this!

b. *Proposed changes to the manuscript:* The manuscript has been updated so that the units are before the parenthetical descriptor.

20. Figs. 7, 8: Nice demonstration of more channels v. better accuracy, with the latter being the only way to dramatically reduce the delta radius solution space uncertainty. That's an important result. (1) Initially, I didn't notice that the y-axis had both positive and negative values. Would be helpful to add a horizontal line to the zero value so readers can quickly appreciate that a large region of the space is outside the constraint. Or add a slight shading to the negative regions. (2) Add a point on the plots to indicate the modeled cloud optical depth and delta effective radius that was used in the simulation (didn't see it mentioned in the text, nor the cloud top effective radius).

a. *Authors' Response:* We agree that both suggestions provide useful information to readers. As the reviewer correctly noticed, the left panel of Figure 7 shows that some of the state vectors within the isopleth of one (the convergence region) have values where $r_{top} - r_{bot} < 0$, which is outside the constraint we defined.

b. *Proposed changes to the manuscript:* We will update Figures 7 and 8 to include a shading that highlights the negative regions. We will also add points to indicate the modeled cloud optical depth and $r_{top} - r_{bot}$ values used in our simulation.

21. Data Availability: If MODIS L2 cloud data was used, please also mention that these files were obtained from LAADS. I strongly suggest providing a doi for both the L1B and L2 files, which should be available on the LAADS product information pages.

    a. *Authors' Response:* We agree.

    b. *Proposed changes to the manuscript:* The Data Availability section will be updated to include DOI's for the L1B, L2 and the MODIS geolocation files. We will cite LAADS as the source of all MODIS data used in our analysis.

**References**

Emde, C., Buras-Schnell, R., Kylling, A., Mayer, B., Gasteiger, J., Hamann, U., Kylling, J., Richter, B., Pause, C., Dowling, T., and Bugliaro, L.: The libRadtran software package for radiative transfer calculations (version 2.0.1), Geoscientific Model Development, 9, 1647–1672, https://doi.org/10.5194/gmd-9-1647-2016, 2016.

King, N. J. and Vaughan, G.: Using passive remote sensing to retrieve the vertical variation of cloud droplet size in marine stratocumulus: An assessment of information content and the potential for improved retrievals from hyperspectral measurements, Journal of Geophysical Research Atmospheres, 117, https://doi.org/10.1029/2012JD017896, 2012.

Meyer, K., Platnick, S., Arnold, G. T., Amarasinghe, N., Miller, D., Small-Griswold, J., Witte, M., Cairns, B., Gupta, S., McFarquhar, G., and O'Brien, J.: Evaluating spectral cloud effective radius retrievals from the Enhanced MODIS Airborne Simulator (eMAS) during ORACLES, Atmospheric Measurement Techniques, 18, 981–1011, https://doi.org/10.5194/amt-18-981-2025, 2025.

Platnick, S.: Vertical photon transport in cloud remote sensing problems, Journal of Geophysical Research Atmospheres, 105, 22919–22935, https://doi.org/10.1029/2000JD900333, 2000.

Twomey, S. and Cocks, T.: Remote sensing of cloud parameters from spectral reflectance in the near-infrared, Beiträge zur Physik der Atmosphäre, 62, 172–179, 1989.

---

## Author Comment (AC6)

**Authors' response to comments from Dr. Zhibo Zhang**

Andrew J. Buggee & Peter Pilewskie

May 2025

We thank the reviewer for their thorough reading of our paper and for providing thoughtful comments. We have addressed each one below.

**Reviewer's General Comments**

1. The other reviewer has provided an excellent summary of this study. Overall, I find this to be a meaningful contribution that explores the potential of CLARREO Pathfinder (PF) observations for advanced cloud remote sensing. However, in addition to the concerns raised by other reviewers, this study has a critical issue that must be addressed before publication: the failure to consider compounding factors—particularly sub-pixel inhomogeneity and three-dimensional (3D) radiative effects—that can significantly impact the retrieval of cloud effective radius (Re) profiles.

2. As outlined below, the influence of 3D radiative transfer effects and sub-pixel inhomogeneity on bi-spectral retrievals, and their implications for effective radius retrievals at different spectral bands (e.g., Re 2.1 µm vs. Re 3.7 µm), have been extensively studied and documented in the literature. Given these well-established issues, I do not believe the paper should be published unless they are thoroughly addressed.

**Reviewer's Major Concerns: Compounding Factors Affecting Retrievals**

The fundamental principle underlying the retrieval algorithm in this study is that different spectral bands are sensitive to different vertical portions of a cloud layer due to their distinct vertical weighting functions, which arise from spectral-dependent absorption. However, spectral differences in retrieved Re values can also be attributed to other factors, such as sub-pixel cloud inhomogeneity and 3D radiative effects, which have not been adequately considered in this paper.

For example, **Zhang and Platnick (2011)** systematically examined the discrepancies in Re retrievals across different spectral bands. A key finding was that Re values retrieved using the 2.1 μm band tend to be significantly larger than those retrieved using the 3.7 μm band. This contradicts expectations based on vertical weighting arguments alone, as the 3.7 μm band, being more absorptive, should produce a larger Re value than the 2.1 μm band. However, actual MODIS retrievals show the opposite pattern. While CLARREO PF does not include the 3.7 μm band, the same biases due to sub-pixel inhomogeneity and 3D effects can still affect retrievals using the 2.1 μm and other bands.

Further, **Zhang et al. (2012, 2016)** demonstrated the impact of sub-pixel inhomogeneity on spectral Re differences (Re 3.7 μm vs. Re 2.1 μm). As shown in **Figure 1 of Zhang et al. (2012)**, the retrieval look-up table (LUT) for Re 3.7 μm is more orthogonal and, therefore, less susceptible to sub-pixel inhomogeneity compared to the Re 2.1 μm retrieval. These findings highlight the need for this study to account for similar effects when evaluating CLARREO PF retrievals.

**Reviewer's Recommendations**

To strengthen the study, I recommend the following:

1. **Use more realistic cloud fields in radiative transfer simulations.**
   - The study currently focuses only on single vertical profiles of Re without considering horizontal cloud variations within and beyond a given pixel. This approach oversimplifies real-world cloud structures.
   - Ideally, large eddy simulation (LES)-generated cloud fields should be used as input for radiative transfer simulations.
   - At a minimum, simple "toy models," such as step clouds or randomly varying cloud fields, should be employed for sensitivity studies. For example, using a step-cloud case and applying a moving average with a 0.5 km resolution pixel would help emulate CLARREO PF observations and test whether the Re profile retrieval algorithm remains robust under spatially averaged radiances.

     a. *Authors' Response:* We agree with the reviewer that our forward model simplifies real cloud structures, which often exhibit horizontal variation within a single pixel. For our study, we performed single-pixel analysis on real MODIS measurements, which precludes any knowledge of sub-pixel information other than the reflectivity at 855 $nm$, which is used to compute the sub-pixel inhomogeneity. In the future, we will investigate

using more sophisticated, high spatial-resolution cloud models to study the impacts of sub-pixel inhomogeneity and 3-D radiative biases on our retrieval.

Regarding the use of "more realistic cloud fields", we believe our best approach is to cite the reviewer's papers on developing a MODIS retrieval simulator on LES cloud fields to demonstrate the potential for 3-D biases. Applying a similar approach for this study would not produce cloud fields any more *real* than the assumed simple cloud structure since the 3-D in situ data needed to initialize the LES was unavailable. We will discuss the benefits of using an LES model to simulate cloud fields and the results from the reviewer's papers in the Discussion and Conclusion section of our manuscript.

A 'step-function' cloud field is a simple method for testing the impacts of horizontal variability on our droplet profile retrieval. Zhang and Platnick (2011) used this method to test how spatial variations in cloud optical thickness cause 3-D radiative biases and impact effective radius retrievals using different MODIS spectral channels. We will cite this work in our discussion of biases introduced by sub-pixel horizontal inhomogeneity in our Discussions and Conclusions section.

Section 4.1 of our manuscript describes the horizontal variation of effective radius computed from in situ measurements along horizontal legs during the VOCALS-REx field campaign (Wood et al., 2011). At nadir viewing, the MODIS cross-track pixel length is about 1 $km$, whereas at the maximum scan angle of 55°, it is about 5 $km$. For these two cross-track pixel length extremes, we found the median horizontal variability of droplet size to be 0.47 $\mu m$ and 0.57 $\mu m$, respectively. These statistics are mentioned because we wish to highlight the relatively small horizontal variations observed in the marine stratocumulus cases used in our analysis.

All pixels used in the development of our algorithm, including the three cases shown in our manuscript, had an inhomogeneity index of less than 0.1. According to Zhang and Platnick (2011), these values represent fairly homogeneous clouds, and 3-D radiative effects are expected to be insignificant.

We should note that Zhang et al. (2012) found horizontal variations in cloud optical thickness were primarily responsible for large differences in retrieved effective radii using different shortwave infrared wavelengths. Due to the flight path characteristics, we are not able to estimate the horizontal variation of optical depth from the VOCALS-REx data at a higher spatial resolution than MODIS.

b. *Proposed changes to the manuscript:* We will expand our Discussions and Conclusions section to include a review of previous work, such as Zhang and Platnick (2011) and Zhang et al. (2012), to highlight the drawbacks of using 1-D vertical cloud profiles and the implications of ignoring horizontal variation in cloud structure. We will discuss future work involving LES-generated cloud fields and 'step-function' cloud fields to more accurately simulate horizontal and vertical cloud inhomogeneity.

2. **Include a dedicated section discussing compounding factors that introduce retrieval errors.**
   o A thorough discussion should be added to explicitly address the effects of sub-pixel inhomogeneity and 3D radiative transfer.
   o The paper should explain how these issues could affect retrieval accuracy and describe potential strategies to mitigate them in the proposed retrieval algorithm.

   *Authors' Response:* The authors acknowledge the lack of discussion about 3-D cloud radiative effects and sub-pixel inhomogeneity in our submitted draft. We limited ourselves to cases where these effects would be small, but that does not negate them entirely. We agree that sub-pixel horizontal inhomogeneities likely impact our retrieval and should be discussed. Zhang and Platnick (2011) showed that both cloud vertical structure and 3-D radiative effects can cause differences in effective radii retrieved using shortwave infrared measurements at 2.1 $\mu m$ and 3.7 $\mu m$. They concluded that it may be possible to determine the cloud droplet profile using different shortwave infrared measurements with an inhomogeneity index of less than 0.1 (Zhang and Platnick, 2011). In a follow-up study, Zhang et al. (2012) used Large Eddy Simulations of cloud fields to show that retrievals of droplet size from pixels with high sub-pixel inhomogeneity were affected by small-scale variations in cloud optical thickness (Zhang et al., 2012). The authors concluded that 3-D radiative effects like illumination and shadowing tend to cancel one another out at MODIS-like spatial scales (Zhang et al., 2012). An additional study by Zinner et al. (2010) also used LES-generated cloud fields to investigate the impact of 3-D radiative effects on retrievals of effective radius and found them to be pronounced only for scattered cumulus scenes. In our analysis, we used MODIS observations of marine stratocumulus with a sub-pixel inhomogeneity index of less than 0.1 to limit the 3-D biases on the retrieval of effective radius.

We agree with the reviewer that a discussion of 3-D biases and sub-pixel inhomogeneity is required if our retrieval is to have broader appeal. Zhang et al. (2012) showed that as sub-pixel inhomogeneity increases, so does the retrieval of effective radius using measurements at 2.1 $\mu m$. We expect our droplet profile retrieval is susceptible to the same bias when using the first seven MODIS spectral channels or measurements from CPF. Zhang et al. (2016) outlined a mathematical framework that can be used to estimate the retrieval uncertainty of effective radius and optical depth when sub-pixel reflectance variations are large. However, mitigation of 3-D effects on traditional 1-D retrievals is an ongoing field of research. Several have shown that machine-learning techniques trained on LES data are capable of overcoming 3-D biases (Nataraja et al., 2022; Okamura et al., 2017). We will include these results in our discussion.

   a. *Proposed changes to the manuscript:* We will expand the Discussion and Conclusion section of our manuscript to outline the limitations of our method when applied to horizontally heterogeneous cloud fields. A thorough discussion of the biases introduced by sub-pixel inhomogeneity, along with expected impacts on our droplet profile retrieval and future work to mitigate these effects, will be included. This discussion is supported by previous results from Zhang et al. (2016).

3. **Expand the discussion on key factors influencing even 1D retrievals.**
   o The current study does not sufficiently account for several important factors that impact retrieval accuracy, including:
      ▪ **Sun-viewing geometry**, which affects radiative transfer and retrieval sensitivity.
      ▪ **Errors in ancillary data**, which are necessary for atmospheric corrections.
      ▪ **Surface reflectance effects**, particularly over land and sun-glint regions, which can introduce additional uncertainties.
   o These factors should be explicitly discussed, along with their potential impact on retrieval performance.

   a. *Authors' Response:* We acknowledge the lack of discussion on different sources of retrieval uncertainty and agree that they should be discussed. See our response to comment 11 from reviewer 1.
      Platnick (2000) demonstrated the retrieval of effective radius for vertically inhomogeneous clouds depends on the solar-viewing geometry

by showing that weighting functions increasingly sample the upper region of the cloud as viewing angle increases. Accordingly, we expect our droplet profile retrieval to estimate larger values at cloud top and bottom as viewing angle increases, if the cloud under observation has a non-homogeneous vertical droplet profile. Furthermore, Grosvenor and Wood (2014) investigated how solar zenith angle affects the MODIS-derived retrieval of effective radius. The authors found that the three effective radius retrievals using the 1.6, 2.1 and 3.7 $\mu m$ MODIS spectral channels closely agreed with one another for small solar zenith angles (Grosvenor and Wood, 2014). We will include these papers in our discussion of how solar and viewing geometry affects the retrieval of effective radius.

Errors in ancillary data, such as vertical profiles of temperature, water vapor and aerosols, the assumed effective variance of the modeled gamma distribution, and the Cox-Munk ocean surface reflectance model, all contribute to the retrieval uncertainty of our droplet profile. Platnick et al. (2017) estimated the uncertainty of these components in order to estimate the uncertainty of MODIS-derived cloud retrievals. The authors estimated a 20% uncertainty for the amount of precipitable water above cloud, the transmittance through ozone-absorbing regions, and the surface wind speed, which greatly affects ocean surface reflectance (Platnick et al., 2017). Lastly, they estimated the uncertainty due to their assumption on the effective variance of the size distribution, which they claim is equal to the standard deviation of the same distribution. These assumptions are valid for our retrieval as well, and we will discuss each one in our manuscript.

b. *Proposed changes to the manuscript:* We will expand section 3 to include a discussion on sources of forward model uncertainty, such as assumed spectral channel independence, assumed retrieved variable independence, errors in vertical profiles of temperature, water vapor and aerosols, surface reflectance uncertainty, horizontal and vertical cloud structure, and the assumed droplet size distribution. In section 4.2, we will adjust the uncertainty added to the simulated TOA reflectance spectra to include measurement uncertainty and an estimate for the forward model uncertainty. We will leverage previous work by Watts et al. (1998), Poulsen et al. (2012), and Platnick et al. (2017) to estimate the fraction of the total uncertainty due to forward model uncertainty. We will include new analysis in section 4 that shows the impact of solar-viewing geometry on droplet profile retrievals using simulated CPF measurements.

**Reviewer's conclusive comment**

While this study explores an important topic, its current approach oversimplifies real-world cloud conditions and neglects key retrieval challenges. Addressing sub-pixel inhomogeneity and 3D radiative transfer effects is crucial for ensuring the validity of the retrieval algorithm. Without such considerations, the conclusions drawn from the study may be misleading. I strongly recommend that these issues be thoroughly addressed before the paper is considered for publication.

**References**

Grosvenor, D. P. and Wood, R.: The effect of solar zenith angle on MODIS cloud optical and microphysical retrievals within marine liquid water clouds, Atmospheric Chem. Phys., 14, 7291–7321, https://doi.org/10.5194/acp-14-7291-2014, 2014.

Li, X.-Y., Wang, H., Chen, J., Endo, S., George, G., Cairns, B., Chellappan, S., Zeng, X., Kirschler, S., Voigt, C., Sorooshian, A., Crosbie, E., Chen, G., Ferrare, R. A., Gustafson, W. I., Hair, J. W., Kleb, M. M., Liu, H., Moore, R., Painemal, D., Robinson, C., Scarino, A. J., Shook, M., Shingler, T. J., Thornhill, K. L., Tornow, F., Xiao, H., Ziemba, L. D., and Zuidema, P.: Large-Eddy Simulations of Marine Boundary Layer Clouds Associated with Cold-Air Outbreaks during the ACTIVATE Campaign. Part I: Case Setup and Sensitivities to Large-Scale Forcings, https://doi.org/10.1175/JAS-D-21-0123.1, 2021.

Nataraja, V., Schmidt, S., Chen, H., Yamaguchi, T., Kazil, J., Feingold, G., Wolf, K., and Iwabuchi, H.: Segmentation-based multi-pixel cloud optical thickness retrieval using a convolutional neural network, Atmospheric Meas. Tech., 15, 5181–5205, https://doi.org/10.5194/amt-15-5181-2022, 2022.

Okamura, R., Iwabuchi, H., and Schmidt, K. S.: Feasibility study of multi-pixel retrieval of optical thickness and droplet effective radius of inhomogeneous clouds using deep learning, Atmospheric Meas. Tech., 10, 4747–4759, https://doi.org/10.5194/amt-10-4747-2017, 2017.

Platnick, S.: Vertical photon transport in cloud remote sensing problems, J. Geophys. Res. Atmospheres, 105, 22919–22935, https://doi.org/10.1029/2000JD900333, 2000.

Platnick, S., Meyer, K. G., King, M. D., Wind, G., Amarasinghe, N., Marchant, B., Arnold, G. T., Zhang, Z., Hubanks, P. A., Holz, R. E., Yang, P., Ridgway, W. L., and Riedi, J.: The MODIS Cloud Optical and Microphysical Products: Collection 6 Updates and Examples from Terra and Aqua, IEEE Trans. Geosci. Remote Sens., 55, 502–525, https://doi.org/10.1109/TGRS.2016.2610522, 2017.

Poulsen, C. A., Siddans, R., Thomas, G. E., Sayer, A. M., Grainger, R. G., Campmany, E., Dean, S. M., Arnold, C., and Watts, P. D.: Cloud retrievals from satellite data using optimal estimation: evaluation and application to ATSR, Atmospheric Meas. Tech., 5, 1889–1910, https://doi.org/10.5194/amt-5-1889-2012, 2012.

Watts, P. D., Mutlow, C. T., Baran, A. J., and Zavody, A. M.: Study on Cloud Properties derived from Meteosat Second Generation Observations, Rutherford Appleton Lab, 1998.

Wood, R., Mechoso, C. R., Bretherton, C. S., Weller, R. A., Huebert, B., Straneo, F., Albrecht, B. A., Coe, H., Allen, G., Vaughan, G., Daum, P., Fairall, C., Chand, D., Gallardo Klenner, L., Garreaud, R., Grados, C., Covert, D. S., Bates, T. S., Krejci, R., Russell, L. M., De Szoeke, S., Brewer, A., Yuter, S. E., Springston, S. R., Chaigneau, A., Toniazzo, T., Minnis, P., Palikonda, R., Abel, S. J., Brown, W. O. J., Williams, S., Fochesatto, J., Brioude, J., and Bower, K. N.: The VAMOS ocean-cloud-atmosphere-land study regional experiment (VOCALS-REx): Goals, platforms, and field operations, Atmospheric Chem. Phys., 11, 627–654, https://doi.org/10.5194/acp-11-627-2011, 2011.

Zhang, Z. and Platnick, S.: An assessment of differences between cloud effective particle radius retrievals for marine water clouds from three MODIS spectral bands, J. Geophys. Res., 116, D20215, https://doi.org/10.1029/2011JD016216, 2011.

Zhang, Z., Ackerman, A. S., Feingold, G., Platnick, S., Pincus, R., and Xue, H.: Effects of cloud horizontal inhomogeneity and drizzle on remote sensing of cloud droplet effective radius: Case studies based on large-eddy simulations, J. Geophys. Res. Atmospheres, 117, 1–18, https://doi.org/10.1029/2012JD017655, 2012.

Zhang, Z., Werner, F., Cho, H.-M., Wind, G., Platnick, S., Ackerman, A. S., Di Girolamo, L., Marshak, A., and Meyer, K.: A framework based on 2-D Taylor expansion for quantifying the impacts of subpixel reflectance variance and covariance on cloud optical thickness and effective radius retrievals based on the bispectral method, J. Geophys. Res. Atmospheres, 121, 7007–7025, https://doi.org/10.1002/2016JD024837, 2016.

Zinner, T., Wind, G., Platnick, S., and Ackerman, A. S.: Testing remote sensing on artificial observations: impact of drizzle and 3-D cloud structure on effective radius retrievals, Atmospheric Chem. Phys., 10, 9535–9549, https://doi.org/10.5194/acp-10-9535-2010, 2010.

---

## Author Response (AR1)

**Authors' response to comments from anonymous reviewer 1**

Andrew J. Buggee & Peter Pilewskie

May 2025

We thank the reviewer for their thorough reading of our paper and for providing thoughtful comments. We have addressed each one below.

Please note that for all changes to the manuscript listed below, the line numbers correspond to the "Authors' track changes" PDF file.

**Reviewer's General Comments**

- 1. This paper is a study on the potential of the upcoming CLARREO Pathfinder (CPF) mission to provide more detailed retrievals of cloud properties than heritage (MODIS-like) imaging sensors thanks to a combination of decreased radiometric uncertainty and increased spectral sampling. The specific geophysical situation studied is joint retrieval of cloud optical depth (COD) and effective radius at the top and bottom of the cloud, for marine stratocumulus scenes. In contrast, one of the major current large-scale approaches (MODIS-like bispectral) retrieves COD and near-top effective radius using a pair of bands, and makes multiple bispectral retrievals with their differences being semi-informative on cloud structure.
- 2. There are two main parts to the analysis. First is use of VOCALS-REX field campaign data to set up some case studies for the proposed retrieval method using MODIS. This has the advantage of being something which can be tested now. The second is a sensitivity study, comparing the capabilities of a MODIS-like sensor with the EMIT instrument as a surrogate for CPF. Together these provide a starting point for moving towards this next level of detail in passive imager cloud retrieval algorithms.
- 3. The manuscript is in scope for AMT. There is a lot to like about this paper: it tackles an important problem, is clearly written, and has some nuance to the discussion. I particularly appreciated the discussion of sampling scales in the VOCALS-REX part of the discussion. The quality of writing and presentation are good. It is (mostly) well-

- referenced. I also appreciate the authors quickly noticing and fixing the incorrect panel of Figure 3. I think it is worth consideration for an AMT science highlight.
- 4. That said, there are points where I think clarification and deeper discussion with respect to realistic performance are needed. As the paper does not claim to be a fully operational approach it does not need to be the final word on the matter, but as a case study and example of what can be done, I think there are sections where more caveats should be discussed, and there are a few things I was not certain about. I recommend minor revisions before publication. I would be willing to review the revision, if the Editor would like.

**Reviewer's Specific Comments**

- 1. Line 31: I'm not sure I'd seen COD described as mean photon free paths through the cloud before, although I can see this framing makes sense as it is the integral of extinction coefficient which is units e.g. km-1 (extinction events per unit distance). Normally it is just referred to as vertical integral of extinction coefficient. I'm curious if there's a reason the authors picked this particular framing for COD.
  - a. *Authors' Response:* We find it useful to describe optical depth as the number of photon mean free paths because it is an intuitive way to think about a quantity that depends on how the number density of particles and their scattering and absorption cross sections vary along a particular path. Bohren and Clothiaux (2006) show that the probability of a photon traveling a geometric distance x before being scattered or absorbed (assuming no multiple scattering), is  $p(x) = (\kappa + \beta) \exp(-(\kappa + \beta)x)$ , where  $\kappa$  is the bulk absorption coefficient and  $\beta$  is the bulk scattering coefficient. The mean free path is then  $\langle x \rangle = \int_0^\infty x \, p(x) dx = \frac{1}{\kappa + \beta} = \ell$ . Since  $\tau_{\lambda} = \int (\kappa + \beta) dz$ , we can also define optical depth as  $\tau_{\lambda} = \int \frac{dz}{\ell}$ , the number of mean free paths.
  - b. Changes to the manuscript: None.
- 2. Line 31: not sure I'd describe effective radius retrievals as "extinction-weighted" but maybe "photon-penetration-weighted"? For a really deep convective cloud, for example, the photons seen from space are still mostly coming from near the cloud top even if the water/extinction would be somewhat further down. And this is in line with e.g. the

Platnick (2000) reference cited and weighting functions shown in the paper. To me "extinction-weighted" implies an optical center of mass.

- a. *Authors' Response:* We used the term 'extinction-weighted' to convey that the retrieved effective radius is a vertically-weighted average that depends on the extinction properties of liquid water at the set of wavelengths used in the retrieval. However, it is clear that our term may lead to some confusion, while the reviewer's proposed term is strictly correct.
- b. *Changes to the manuscript:* Line 41: Changed "extinction-weighted" to "photon-penetration-weighted".
- 3. Introduction, general: I like the historical discussion, but there are a few omissions that I think are quite relevant. One is the ORAC retrieval which came out of the same lab as Clive Rodgers who put down the Optimal Estimation (OE) formalism used here, applied mostly to European sensors (ATSRs and successors). See Sayer et al (2011) and Poulsen et al (2012). This isn't an explicitly bispectral approach (uses all bands together) but only retrieved a single effective radius (sensitivity from 1.6 and 3.7 micron bands) as opposed to attempting a profile. Another is the VISST algorithm applied to cloud properties from MODIS observations within CERES pixels (as part of the CERES data processing chain), which is also not bispectral but again retrieving a single effective radius from visible and multiple SWIR bands (0.65, 1.6, 2.1, 3.7 micron). The reference I use for this is Minnis et al (2011) – that paper cites some earlier AVHRR work using that algorithm from the late 1990s, but it's in conference proceedings that don't seem to be broadly available, so I can't say for sure what was done. There is also earlier OE work by e.g. Heidinger (2003) applied to the AVHRRs (a lot of later work from that NOAA team focuses on the infrared, but the above algorithm also used solar radiances and is more conceptually similar to bispectral). All of these approaches (ORAC, CERES, AVHRR) have been applied to multi-decadal multi-sensor records and approach the question of effective radius parameterization a bit differently from either the bispectral method or the profiling method, so I think merit some discussion in the manuscript. Also, I think all of these methods were applied somewhat earlier than the publications describing them were written (otherwise mostly documented in proceedings and technical reports) so they are not such newcomers as the paper dates might imply.
  - a. *Authors' Response:* We appreciate the reviewer sharing these relevant papers. Poulsen et al. (2012) was particularly illuminating with its thorough outline of the ORAC retrieval methodology and the description of forward model uncertainty. The results of Sayer et al. (2011) suggest that the multispectral retrieval of effective radius estimates effective radii deeper in the cloud where droplet sizes

tend to be smaller. The paper concludes with an endorsement for the retrieval of vertical droplet profiles. Heidinger (2003) applied the Rodgers optimal estimation technique to retrieve effective radius and optical depth using one channel in the visible, one in the near-infrared, and two in the infrared. Minnis et al. (2011) describes an iterative technique to retrieve cloud phase, optical depth, and effective radius from MODIS and VIIRS observations to support the CERES data products. We will highlight all of these papers in the introduction.

- b. *Changes to the manuscript:* Lines 134-143: Descriptions of previous work by Heidinger (2003), Minnis et al. (2011), Sayer et al. (2011), and Poulsen et al. (2012) have been added to the historical background section. Lines 304-308, 440-445, 535, and 774: We cite Poulsen et al. (2012) numerous times when discussing the optimal estimation retrieval method and forward model uncertainty.
- 4. Line 106: I see there is a paper reference there but for ease it would be good to detail the expected pixel size, orbital geometry, swath width, and spectral sampling/bandwidth of the CPF mission as well. This should be recapped in the conclusion as well, where relevant (e.g. in the discussion of scales of variability in marine stratocumulus clouds).
  - a. Authors' Response: We agree.
  - b. Changes to the manuscript: Lines 168-174: Descriptions of CLARREO Pathfinder (CPF) spectral sampling and resolution, orbital geometry, spatial resolution, and swath width were added. Figure 6: An additional histogram with a length scale of 0.5km, the spatial sampling of the CPF instrument at nadir was added. While the result is similar to the 1 km spatial sampling of MODIS when looking nadir, we found it useful to show that, as pixel size decreases, the average variability of effective radius with respect to the horizontal plane decreases.
- 5. Section 2.1: I would suggest renaming this "the bispectral method" instead of "the standard method". What does "standard" mean? From a polar-orbiting viewpoint, yes, this method has been applied routinely to MODIS and VIIRS. But that in my view implies it's the only way things are done, despite e.g. the ATSR, AVHRR, CERES references I provided which have similar (or longer) time series of data.
  - a. Authors' Response: We agree and will adopt "bispectral method".

- b. *Changes to the manuscript:* Line 196: Section 2.1 heading and elsewhere: We updated our phrasing throughout the paper to use 'bispectral' instead of 'standard'.
- 6. Line 233: In practical terms  $S_{\epsilon}$  tends to be used not just for measurement uncertainty but the combination of measurement plus forward model uncertainty covariance. This may be worth noting. Mathematically, it doesn't make a difference whether one puts only measurement error in  $S_{\epsilon}$  (in which forward model parameterization uncertainty is normally put in another matrix often called  $S_b$  in Rodgers notation), or combines both measurement and forward model uncertainty. This is omitted from the equations and discussions here. See also my comment 11, which is my main issue with the paper as written.
  - a. *Authors' Response:* We agree with the reviewer's comment that there should be some discussion on forward model errors. Indeed, our forward model deviates from the true nature of clouds and the atmosphere due to the many simplifications, which deserve scrutiny. The recommended paper by Poulsen et al. (2012) was particularly illuminating in this regard, thanks to its thorough discussion of forward model uncertainties.
  - b. Changes to the manuscript: Lines 309-314: We updated the definition of the measurement covariance matrix,  $S_{\epsilon}$ , so that it is now the sum of measurement and forward model uncertainty, citing Poulsen et al. (2012).
- 7. Line 260 and elsewhere: the paper often refers to the "constrained" OE approach, kind of making it seem like the constraints are unusual or an innovation. In reality though every algorithm (including OE ones) are putting in constraints similar to this (state bounds). I'm not sure that the word "constrained" needs to be emphasized in the paper very much as it makes the reader focus more on that while in my view the novel aspect is getting at radius profiles in adiabatic clouds.
  - a. Authors' Response: We do not claim that constrained optimal estimation is an innovation of our own. We chose to repeat that phrase to emphasize the importance of the constraint. Without it, using MOIDS measurements with ~ 2% measurement uncertainty can lead to retrieved profiles where the droplet size at cloud top is smaller than cloud bottom, violating our forward model assumption. That being said, we appreciate the reviewer's comment because we do not wish to distract readers from the more important result of retrieving droplet profiles.

- b. *Changes to the manuscript:* Line 290 and elsewhere: The qualifier *constrained* was removed from the section 2.3 heading and from nearly all mentions of the optimal estimation method. Where the constraints are first introduced, we emphasized their importance.
- 8. Line 276 and elsewhere: the residual/left side of L2 norm is most commonly referred to as the "cost function" and often denoted capital italic *J* in the Rodgers formalism. For ease of readers comparing different references, I think it would be good to note these notation/terminology differences somewhere around here.
  - a. *Authors' Response*: We will adopt the terminology and notation that are commonly used in the retrieval community. However, we would like to point out that the left side of equation 11 in our manuscript is not the cost function in the sense outlined by Rodgers (the first two terms on the right-hand side of equation 5.3 in Rodgers, 2000) or Poulsen (equation 1 of Poulsen et al. (2012)). This is why originally defined the L2-norm of the difference between the forward-modeled reflectances and the measurements as the 'residual'.
  - b. Changes to the manuscript: Line 368 and elsewhere: We defined the cost function as the  $L^2$ -norm of the difference between the forward-modeled reflectances and the measurements. Instead of continually referring to the  $L^2$ -norm throughout the paper, we used 'cost function', as is custom in the retrieval community.
- 9. Line 286: for completeness, I'd add the equation for uncertainty estimate on the retrieved state here. Unless I missed it, it seems to not be included, and as part of the paper is talking about expected improvements from CPF I think it is worth including explicitly how this is calculated.
  - a. Authors' Response: We agree.
  - b. *Changes to the manuscript:* Line 381: We added the equation for computing the posterior covariance matrix (Equation 13).
- 10. Line 335: the MODIS retrieval uncertainties used as the a priori uncertainty should be stated here, and a citation to where they came from added.

- a. *Authors' Response:* The a priori uncertainty for optical depth and effective radius at cloud top was defined as the MODIS retrieval uncertainty for optical depth and effective radius, respectively. The MODIS retrievals and their respective uncertainties vary between pixels. Therefore, there is no single number to report. For the three MODIS scenes used in our paper, the mean retrieval uncertainty for cloud effective radius over ocean with an optical depth of at least three was 10.6% ( $\sim 0.89 \ \mu m$ ). For optical depth, the mean retrieval uncertainty was 5.9% ( $\sim 0.56$ ). These values align with the expected retrieval uncertainty of the MODIS Collection 6 cloud products (Platnick et al., 2017).
- b. *Changes to the manuscript:* Lines 507-513: We added a citation for the retrieval uncertainty of the MODIS Collection 6 cloud products (Platnick et al., 2017). We also included the statistics mentioned above for the retrieval uncertainty of cloudy pixels over the ocean with an optical depth of at least three to provide readers with an idea of the values used in our analysis.
- 11. Sections 3 onwards: my main technical issue with the MODIS retrievals and simulated CPF uncertainties is that they are a realistic "best case" performance and this is kind of skirted over. The discussion more or less takes the only relevant uncertainty source as radiometric (sensor absolute calibration uncertainty and shot noise). Even if that were true, from my reading the calibration uncertainty is taken as spectrally independent. In reality it may be spectrally correlated (based on experiences with various space-based sensors) which affects downstream uncertainty characterization. But really, the main issue is the implicit assumption that the forward model (including its numerical implementation) is perfect which is inherently false (and semi-acknowledged by the fact the section 3 title includes "forward model assumptions"). These assumptions, as well as e.g. factors like lookup table interpolation precision, uncertainties in ancillary data (surface reflectance/albedo, gas columns), and non-calibration image artefacts (e.g. 3D radiative transfer effects, image ghosts, delayed impulse response after bright pixels), are often similar to or larger than absolute calibration uncertainty. And these can all have e.g. angular dependence and spectral covariation as well. So this is a big reason why retrievals are never as good as idealized sensitivity studies (as they rarely can take into account these factors). I understand this paper is a proof of concept and not a full operational algorithm. But I think it is necessary to acknowledge these issues seriously (I really doubt we can make our forward models good enough to take advantage of CPF's radiometric calibration quality). Otherwise it feels like it is misleadingly over-hyping the CPF mission as folks who don't work in algorithm development may well not be aware that radiometric quality is only one of the determining factors in retrieval quality. I wonder if somehow this discussion could be tied into the existing sensitivity studies (or

new sensitivity studies). Maybe this could involve comparing MODIS retrieval uncertainties with the width of contours in figures 7 and 8 - I will leave this to the authors to decide how best to respond.

a. *Authors' Response:* We acknowledge the lack of discussion on sources of forward model uncertainty. Forward model uncertainty is difficult to quantify but should not be ignored. We agree with the reviewer that our discussion in sections 4.2 and 5 should focus on total uncertainty. Minimizing forward model uncertainty leads to a measurement-limited solution that, the reviewer points out, may be unachievable with CPF measurements. Assuming a droplet profile is just one assumption that reduces forward model uncertainty because the assumption of a vertically homogeneous cloud is known to be a simplification for certain types of clouds (Platnick, 2000). In the future, an optimal estimation algorithm may be able to leverage the full spectrum of CPF to simultaneously estimate cloud phase (Pilewskie and Twomey, 1987), cloud top height (Rozanov and Kokhanovsky, 2004), above-cloud column water vapor (Albert et al., 2001), *CO*2 column amount (Buchwitz and Burrows, 2004), and aerosol optical depth (Mauceri et al., 2019), reducing forward model uncertainty by limiting the number of assumptions.

We assumed the radiometric uncertainty of the instrument was uncorrelated, and the reviewer is correct in noting that this is not the best representation of real space-based spectrometers. Kopp et al. (2017) computed the relative total radiometric uncertainty for the CPF instrument, HySICS, as a function of spectral channel for bright (cloud-filled) Earth viewing scenes. Flat field uncertainty dominates at short wavelengths, while shot noise and brightness offset dominate at longer wavelengths. For each channel, the total relative uncertainty appears to strongly covary with neighboring channels (Kopp et al., 2017). Future iterations of this work will leverage these findings to define the off-diagonal elements of the measurement covariance matrix. That said, we will emphasize that the assumption of uncorrelated measurement uncertainty between spectral channels is a simplification of the true instrument.

Lastly, we do not want the framing of our results to overstate our findings. We will adjust the wording in sections 4.2 and 5 to provide the necessary context for our results. We appreciate the reviewer's suggestion to include a discussion on how our multi-spectral retrieval uncertainty compares with the well-documented MODIS Collection 6 effective radius retrieval uncertainty (Platnick et al., 2017). We will incorporate this into section 4.2.

b. *Changes to the manuscript:* Lines 312-314, 445-451, 541-544, 789-794, 923-986, 1050-1059: We updated section 3 to include a description on sources of forward

model uncertainty, following previous work by Poulsen et al. (2012). In section 4.2, we adjusted the uncertainty added to the simulated TOA reflectance spectra to include both measurement and forward model uncertainty. Instead of explicitly estimating the uncertainty of each source within the forward model, we leveraged previous work by Poulsen et al. (2012) to provide reasonable estimates for the fraction of the total uncertainty due to forward model uncertainty. We also emphasized that forward model uncertainty can never be reduced entirely. Additionally, we compared our multispectral retrieval uncertainty estimate using simulated CPF TOA reflectances with the MODIS collection 6 cloud products retrieval uncertainty.


- The study currently focuses only on single vertical profiles of Re without considering horizontal cloud variations within and beyond a given pixel. This approach oversimplifies real-world cloud structures.
- o Ideally, large eddy simulation (LES)-generated cloud fields should be used as input for radiative transfer simulations.
- At a minimum, simple "toy models," such as step clouds or randomly varying cloud fields, should be employed for sensitivity studies. For example, using a step-cloud case and applying a moving average with a 0.5 km resolution pixel would help emulate CLARREO PF observations and test whether the Re profile retrieval algorithm remains robust under spatially averaged radiances.
  - a. *Authors' Response:* We agree with the reviewer that our forward model simplifies real cloud structures, which often exhibit horizontal variation within a single pixel. For our study, we performed single-pixel analysis on real MODIS measurements, which precludes any knowledge of sub-pixel information other than the reflectivity at 855 nm, which is used to compute the sub-pixel inhomogeneity. In the future, we will investigate

using more sophisticated, high spatial-resolution cloud models to study the impacts of sub-pixel inhomogeneity and 3-D radiative biases on our retrieval.

Regarding the use of "more realistic cloud fields", we believe our best approach is to cite the reviewer's papers on developing a MODIS retrieval simulator on LES cloud fields to demonstrate the potential for 3-D biases. Applying a similar approach for this study would not produce cloud fields any more *real* than the assumed simple cloud structure since the 3-D in situ data needed to initialize the LES was unavailable. We will discuss the benefits of using an LES model to simulate cloud fields and the results from the reviewer's papers in the Discussion and Conclusion section of our manuscript.

A 'step-function' cloud field is a simple method for testing the impacts of horizontal variability on our droplet profile retrieval. Zhang and Platnick (2011) used this method to test how spatial variations in cloud optical thickness cause 3-D radiative biases and impact effective radius retrievals using different MODIS spectral channels. We will cite this work in our discussion of biases introduced by sub-pixel horizontal inhomogeneity in our Discussions and Conclusions section.

Section 4.1 of our manuscript describes the horizontal variation of effective radius computed from in situ measurements along horizontal legs during the VOCALS-REx field campaign (Wood et al., 2011). At nadir viewing, the MODIS cross-track pixel length is about 1 km, whereas at the maximum scan angle of 55°, it is about 5 km. For these two cross-track pixel length extremes, we found the median horizontal variability of droplet size to be 0.47  $\mu m$  and 0.57  $\mu m$ , respectively. These statistics are mentioned because we wish to highlight the relatively small horizontal variations observed in the marine stratocumulus cases used in our analysis.

All pixels used in the development of our algorithm, including the three cases shown in our manuscript, had an inhomogeneity index of less than 0.1. According to Zhang and Platnick (2011), these values represent fairly homogeneous clouds, and 3-D radiative effects are expected to be insignificant.

We should note that Zhang et al. (2012) found horizontal variations in cloud optical thickness were primarily responsible for large differences in retrieved effective radii using different shortwave infrared wavelengths. Due to the flight path characteristics, we are not able to estimate the horizontal variation of optical depth from the VOCALS-REx data at a higher spatial resolution than MODIS.

b. Changes to the manuscript: Lines 1067-1089: We added two new paragraphs to our Discussions and Conclusions section to include a review of previous work highlighting the drawbacks of using 1-D clouds and the implications of ignoring horizontal variation in cloud structure, citing Zhang and Platnick (2011), Zhang et al. (2012) and Zinner et al. (2010). We discussed future work involving LES-generated cloud fields to more accurately simulate horizontal and vertical cloud inhomogeneity.

**2. Include a dedicated section discussing compounding factors that introduce retrieval errors.**

- A thorough discussion should be added to explicitly address the effects of subpixel inhomogeneity and 3D radiative transfer.
- The paper should explain how these issues could affect retrieval accuracy and describe potential strategies to mitigate them in the proposed retrieval algorithm.

Authors' Response: The authors acknowledge the lack of discussion about 3-D cloud radiative effects and sub-pixel inhomogeneity in our submitted draft. We limited ourselves to cases where these effects would be small, but that does not negate them entirely. We agree that sub-pixel horizontal inhomogeneities likely impact our retrieval and should be discussed. Zhang and Platnick (2011) showed that both cloud vertical structure and 3-D radiative effects can cause differences in effective radii retrieved using shortwave infrared measurements at 2.1 µm and 3.7 µm. They concluded that it may be possible to determine the cloud droplet profile using different shortwave infrared measurements with an inhomogeneity index of less than 0.1 (Zhang and Platnick, 2011). In a follow-up study, Zhang et al. (2012) used Large Eddy Simulations of cloud fields to show that retrievals of droplet size from pixels with high sub-pixel inhomogeneity were affected by small-scale variations in cloud optical thickness (Zhang et al., 2012). The authors concluded that 3-D radiative effects like illumination and shadowing tend to cancel one another out at MODIS-like spatial scales (Zhang et al., 2012). An additional study by Zinner et al. (2010) also used LES-generated cloud fields to investigate the impact of 3-D radiative effects on retrievals of effective radius and found them to be pronounced only for scattered cumulus scenes. In our analysis, we used MODIS observations of marine stratocumulus with a sub-pixel inhomogeneity index of less than 0.1 to limit the 3-D biases on the retrieval of effective radius.

We agree with the reviewer that a discussion of 3-D biases and sub-pixel inhomogeneity is required if our retrieval is to have broader appeal. Zhang et al. (2012) showed that as sub-pixel inhomogeneity increases, so does the retrieval of effective radius using measurements at 2.1  $\mu$ m. We expect our droplet profile retrieval is susceptible to the same bias when using the first seven MODIS spectral channels or measurements from CPF. Zhang et al. (2016) outlined a mathematical framework that can be used to estimate the retrieval uncertainty of effective radius and optical depth when sub-pixel reflectance variations are large. However, mitigation of 3-D effects on traditional 1-D retrievals is an ongoing field of research. Several have shown that machine-learning techniques trained on LES data are capable of overcoming 3-D biases (Nataraja et al., 2022; Okamura et al., 2017). We will include these results in our discussion.

a. Changes to the manuscript: Lines 453-467, 1077-1089: We included a new paragraph in Section 3 and in the Discussions and Conclusions section that reviews the limitations of our method when applied to horizontally heterogeneous cloud fields. We discussed the biases introduced by sub-pixel inhomogeneity, along with the expected impacts on our droplet profile retrieval, and highlighted studies addressing the mitigation of these effects.

**3. Expand the discussion on key factors influencing even 1D retrievals.**

- The current study does not sufficiently account for several important factors that impact retrieval accuracy, including:
  - **Sun-viewing geometry**, which affects radiative transfer and retrieval sensitivity.
  - Errors in ancillary data, which are necessary for atmospheric corrections.
  - Surface reflectance effects, particularly over land and sun-glint regions, which can introduce additional uncertainties.
- These factors should be explicitly discussed, along with their potential impact on retrieval performance.
  - a. *Authors' Response:* We acknowledge the lack of discussion on different sources of retrieval uncertainty and agree that they should be discussed. See our response to comment 11 from reviewer 1.

Platnick (2000) demonstrated the retrieval of effective radius for vertically inhomogeneous clouds depends on the solar-viewing geometry by showing that weighting functions increasingly sample the upper region of the cloud as viewing angle increases. Accordingly, we expect our droplet profile retrieval to estimate larger values at cloud top and bottom as viewing angle increases, if the cloud under observation has a non-homogeneous vertical droplet profile. Furthermore, Grosvenor and Wood (2014) investigated how solar zenith angle affects the MODIS-derived retrieval of effective radius. The authors found that the three effective radius retrievals using the 1.6, 2.1 and 3.7  $\mu m$  MODIS spectral channels closely agreed with one another for small solar zenith angles (Grosvenor and Wood, 2014). We will include these papers in our discussion of how solar and viewing geometry affects the retrieval of effective radius.

Errors in ancillary data, such as vertical profiles of temperature, water vapor and aerosols, the assumed effective variance of the modeled gamma distribution, and the Cox-Munk ocean surface reflectance model, all contribute to the retrieval uncertainty of our droplet profile. Platnick et al. (2017) estimated the uncertainty of these components in order to estimate the uncertainty of MODIS-derived cloud retrievals. The authors estimated a 20% uncertainty for the amount of precipitable water above cloud, the transmittance through ozone-absorbing regions, and the surface wind speed, which greatly affects ocean surface reflectance (Platnick et al., 2017). Lastly, they estimated the uncertainty due to their assumption on the effective variance of the size distribution, which they claim is equal to the standard deviation of the same distribution. These assumptions are valid for our retrieval as well, and we will discuss each one in our manuscript.

b. Changes to the manuscript: Lines 445-451, 789-794, 1091-1111: We included a discussion on sources of forward model uncertainty, such as assumed spectral channel independence, assumed retrieved variable independence, errors in vertical profiles of temperature, water vapor and aerosol uncertainty, surface reflectance uncertainty, horizontal and vertical cloud structure, and the assumed droplet size distribution. In Section 4.2, we adjusted the uncertainty added to the simulated TOA reflectance spectra to account for measurement uncertainty and an estimate of the forward model uncertainty. We leveraged previous studies by Watts et al. (1998), Poulsen et al. (2012), and Platnick et al. (2017) to estimate the fraction of the total uncertainty due to forward model uncertainty. We also discussed how solar-viewing geometry, errors in ancillary data, and errors in surface reflectance can impact our 1-D retrieval in the Discussions and Conclusions section.

**Reviewer's conclusive comment**

While this study explores an important topic, its current approach oversimplifies real-world cloud conditions and neglects key retrieval challenges. Addressing sub-pixel inhomogeneity and 3D radiative transfer effects is crucial for ensuring the validity of the retrieval algorithm. Without such considerations, the conclusions drawn from the study may be misleading. I strongly recommend that these issues be thoroughly addressed before the paper is considered for publication.

  - a. Authors' response: We agree with the reviewer that the retrieval method we developed is only valid for non-precipitating clouds with droplet size increasing from cloud base to cloud top. These two constraints were made for different reasons. For the case of non-precipitating clouds, we are currently limited to retrieving droplet sizes up to 25 μm because this is the upper limit of the table of Mie calculations for liquid water droplets provided by libRadtran. To compute radiance for cloudy scenes, libRadtran utilizes a pre-computed table to convert cloud properties into optical properties. The next iteration of this work will use an expanded lookup table that ranges up to 50 microns.

We assumed droplet size at cloud top was larger than at cloud base because both in-situ measurements and parcel theory show this to be true for nonprecipitating stratocumulus clouds. At every iteration, the forward model computes the top-of-atmosphere reflectance for a cloudy scene with some vertical droplet profile. Since we retrieve only two values from this vertical profile, at cloud top and cloud base, we must make an assumption about how droplets change in size as a function of height. Future iterations of this work will investigate different assumptions, such as sub-adiabatic and linear growth (Platnick, 2000).

Remote measurements of polarized cloud reflectance are often used to retrieve the effective variance of the droplet distribution (Meyer et al., 2025; Pörtge et al., 2023). Since we investigated the retrieval of droplet profiles from measurements taken by MODIS, which measures unpolarized reflectance, we were unable to retrieve the effective variance. Thus, we assumed a standard value used in the community. We agree with the reviewer that this may introduce forward model error if the cloud under observation has a multimodal droplet distribution, such as the example in Pörtge et al. (2023). We will discuss the implications of this in our manuscript, however, multiple studies have shown that the presence of drizzle has only modest impacts on the retrieval of effective radius at various wavelengths (Painemal and Zuidema, 2011; Zhang et al., 2012; Zinner et al., 2010).

b. Changes to the manuscript: Lines 350-354, 404-409, 1095-1100: We updated sections 2.3 and 3 with discussions of the limitations of our droplet profile retrieval, considering the constraints of non-precipitating adiabatic clouds with droplet sizes less than 25 μm. We explained that the upper limit of 25 μm stems from using the precomputed table of Mie calculations for liquid water droplets provided by libRadtran, which only extends up to 25 microns. Therefore, we were limited to retrieving droplet profiles from non-precipitating clouds only. The next iteration of this work will utilize a precomputed table of Mie calculations with a maximum droplet size exceeding 25 microns. We expanded the Discussions and Conclusions section to mention that while drizzle may be present in non-precipitating clouds, several studies have shown the impact on the retrieval of effective radius to be minor.

**Reviewer's Specific Comments**

1. L. 122f./Sec. 4.2: I was wondering why the authors did not simulate CPF spectra directly instead of the EMIT spectra? Since this part is a purely theoretical study, I think one could have used the CPF specifications directly to demonstrate how the smaller

radiometric uncertainty and the usage of more spectral channels influences the solution space.

- a. *Authors' Response:* The EMIT spectral response functions are freely available, thus we can simulate TOA reflectance spectra without any guesswork. However, after your suggestion, we reached out to the instrument team that developed HySICS (the HyperSpectral Imager for Climate Science) for the CPF mission and asked if we could use the spectral response functions to simulate CPF sampled TOA reflectance.
- b. *Changes to the manuscript:* Section 4.2, Figures 8 and 9: We were given access to the HySICS spectral response functions and we have updated the analysis in section 4.2 such that the TOA reflectance spectra used to generate the contour plots in Figures 8 and 9 simulated the HySICS spectral channels. We no longer use simulated EMIT measurements.
- 2. L. 106f.: In agreement to the first referee, I also think that it would be valuable to introduce the CPF instrument in more detail. In particular, if I have not overseen anything, I am missing the number of spectral channels and the horizontal resolution, please add if possible. And is there a spectral dependence of the measurement uncertainty? If so, the implications for the retrieval should also be addressed in the discussion.
  - a. Authors' Response: We agree.
  - b. Changes to the manuscript: Lines 168-174: Descriptions of CLARREO Pathfinder (CPF) spectral sampling and resolution, orbital geometry, spatial resolution, and swath width were added. Figure 6: An additional histogram with a length scale of 0.5km, the spatial sampling of the CPF instrument at nadir was added. While the result is similar to the 1 km spatial sampling of MODIS when looking nadir, we found it useful to show that, as pixel size decreases, the average variability of effective radius with respect to the horizontal plane decreases. Results from the study by Kopp et al. (2017) are included in the Discussions and Conclusions section. This study found that neighboring spectral channels strongly covary with one another.
- 3. L. 253: Are the partial derivative fractions presented only valid for the MODIS measurement uncertainty? And are they valid for all wavelengths? Please clarify in the manuscript.

- a. Authors' Response: The partial derivative fractions were derived using MODIS measurements. Through trial and error, we determined a set of fractions that would often exceed the MODIS measurement uncertainty of about 2%. We had to strike a balance between precisely estimating the Jacobian, defined as the rate of change of reflectance with respect to some infinitesimal change in one of the state variables, and the measurement uncertainty. For example, we found if  $\Delta r_{bot}$  was too small, then  $\Delta R(\vec{x}, \lambda_i)$  was small and measurement uncertainty dominated. If  $\Delta r_{bot}$  was too large, we no longer accurately estimated the local slope. These fractions were used for all seven spectral channels because the MODIS measurement uncertainty at these channels is roughly constant. Adjustments should be made for use with other instruments. Lower radiometric uncertainty enables the detection of smaller changes in reflectance, which implies that the partial derivatives of the Jacobian can be estimated more precisely.
- b. *Changes to the manuscript:* Lines 335-341: We expanded on our description of how the Jacobian was estimated, including information on its purpose and applicability.
- 4. Table 1: In my opinion, it would be very valuable to add the resolution of each MODIS channel and the respective measurement uncertainty here.
  - a. Authors' Response: We agree.
  - b. *Changes to the manuscript:* Table 1: The table was updated to include spectral resolution and measurement uncertainty.
- 5. L. 323f.: Where do you see the shapes of the distributions from?
  - a. *Authors' Response:* We normalized the vertical dimension and discretized it into 30 bins for each in-situ profile. After doing this for all in-situ measurements without drizzle or precipitation, we found the median value for each bin, which is what we plotted in Fig. 2. Within each bin, we fit a distribution to the data. We found that for effective radius and liquid water content, a log-normal distribution best fit the measurements for most of the vertical bins, whereas a normal distribution was the best fit for number concentration for most of the bins. This explains why the shading in Fig. 2, which represents the average deviation from the median value, is symmetric for number concentration and asymmetric for the effective radius and liquid water content.

- b. *Changes to the manuscript:* Lines 489-495: We expanded our description of Figure 2 to include information about the data and best distribution fits.
- 6. L. 336f.: Perhaps there might be something which I did not understand correctly, but why are you using the 2.13 μm weighting function for the cloud bottom here? To my understanding of Fig. 1, the effective radius derived from that one corresponds to the smallest optical thickness of all channels considered?
  - a. Authors' Response: Thank you for bringing this up, indicating the need for a more detailed explanation in this section. The a priori value for the radius at cloud top was defined as the retrieved effective radius using MODIS measurements, and the a priori uncertainty as the associated retrieval uncertainty. The a priori value for the radius at cloud bottom is defined as 70% of the retrieved effective radius (the a priori value for cloud top radius). This percentage was derived from the median vertical profile of effective radius from the in situ measurements shown in Fig. 1, which shows that the median value of cloud bottom radius was 70% the value of the median cloud top effective radius. The bi-spectral retrieval of effective radius was computed using two MODIS channels centered at 645 nm and 2.13  $\mu$ m, but is predominantly determined from the near-infrared wavelength. As the reviewer points out, most photons at 2.13 µm scatter near cloud top. Therefore, we need to express a higher uncertainty for the a priori value for the radius at cloud bottom. We used the 2.13  $\mu m$  weighting function to determine that the portion of the total measured signal with a maximum penetration depth within the upper quartile of the cloud was six times that of the portion with a maximum penetration depth within the lower quartile. This is why we defined the a priori uncertainty for the radius at cloud bottom to be 6 times the value of cloud top.
  - b. Changes to the manuscript: Lines 502-513, 534-538: We expanded on our discussion of the a priori values and their uncertainties by outlining how each value is defined, including a clearer description of the use of the bi-spectral retrieval of effective radius from the MODIS collection 6 cloud products to define the a priori at cloud top and bottom. We also provided an explanation on why the weighting function for 2.13 μm was used to determine the a priori uncertainty for the cloud bottom radius.
- 7. L. 375: Are the optical depths stated here derived from the retrieval? Please clarify, where those are derived from.
  - a. Authors' Response: Those values were derived from the in situ measurements.

- b. *Changes to the manuscript:* Lines 586-587: The sentence has been rewritten with the source of the optical depth estimates made clear.
- 8. Fig. 3: In my opinion, it would be nice to have the corresponding MODIS pictures and an indication where the measurements took place in addition to the profiles. This would give the reader an overall impression of the cloud situation and scenery. Moreover, the measurement times would be interesting to know for the solar geometry for which the comparisons have been made. And how long did it take for the aircraft to sample the profiles, what was the flight distance/spatial coverage of the in-situ measurements?
  - a. *Authors' Response:* This is a good suggestion that would provide further context for the results. Instead of including the RGB images of each MODIS scene, we will report the inhomogeneity index, along with the latitude and longitude of the measurements, within a new table that provides information on the MODIS and associated VOCALS-REx measurements used in our analysis.
  - b. *Changes to the manuscript:* Table 2: We included a new table that outlines the time of each MODIS observation, the start and end times of the overlapping VOCALS-REx measurements, the time difference, the geographic location of the MODIS observations, and the sub-pixel inhomogeneity index.

|        | MODIC                              | MODIS        | MODIS Sub-         | VOCALS-     | VOCALS-     | Time       |
|--------|------------------------------------|--------------|--------------------|-------------|-------------|------------|
| Figure | MODIS
Observation
time (UTC) | Observation  | pixel              | REx in-situ | REx in-situ | difference |
|        |                                    | latitude and | inhomogeneity      | start time  | end time    | (min)      |
|        |                                    | longitude    | index $H_{\sigma}$ | (UTC)       | (UTC)       |            |
| 3a     | Nov 11 2008                        | -24.0986,    | 0.09               | 18:45:20    | 18:45:50    | 8.88       |
|        | 18:54:28                           | -75.0013     |                    |             |             |            |
| 3b     | Nov 11 2008                        | -22.8188,    | 0.07               | 14:40:59    | 14:41:38    | 1.18       |
|        | 14:42:29                           | -73.0008     |                    |             |             |            |
| 3c     | Nov 9 2008                         | -22.8970,    | 0.08               | 14:33:33    | 14:34:23    | 3.62       |
|        | 14:30:20                           | -73.0036     |                    |             |             |            |

- 9. L. 400f.: One common issue of the bispectral retrieval is the overestimation of the effective radius due to 3-D cloud radiative effects and broken cloudiness. Could this be a reason for the effective radius profile showing larger values than the in-situ measurements? Here, it would also help to have a visualization of the cloud scenery.
  - a. *Authors' Response:* We agree that 3-D radiative effects and sub-pixel inhomogeneity should be discussed. See our responses to comments 1 and 2 from Reviewer 2 (Dr. Zhibo Zhang). For the three cases used in our manuscript to

- retrieve droplet profiles (Figure 3), all had an inhomogeneity index of less than 0.1. According to Zhang and Platnick (2011), these values represent fairly homogeneous clouds and 3-D radiative effects are expected to be insignificant.
- b. Changes to the manuscript: Lines 453-467, 668-674, 1067-1089: We expanded our Discussions and Conclusions section to include a review of previous work highlighting the drawbacks of using 1-D clouds and the implications of ignoring horizontal variation in cloud structure, citing Zhang and Platnick (2011), Zhang et al. (2012) and Zinner et al. (2010). We discussed future work involving LES-generated cloud fields to more accurately simulate horizontal and vertical cloud inhomogeneity. We included a discussion on the limitations of our method when applied to horizontally heterogeneous cloud fields. We reviewed the biases introduced by sub-pixel inhomogeneity, along with expected impacts on our droplet profile retrieval and future work to mitigate these effects. We noted the historical precedent of remote retrievals overestimating in situ measurements and discussed one recent study by Meyer et al. (2025) that highlights this bias.
- 10. Fig. 5: Please make a comment on the two spikes which are very pronounced in the blue line. Where do they come from? I suspect that they also influence the derived standard deviation and calculated range quite a lot.
  - a. *Authors' Response*: The reviewer is correct in pointing out that these two outliers affected the range and standard deviation of this horizontal profile. The in-situ measurements show decreases in the total droplet number concentration at the same moment the effective radius rapidly increases. Likely, these regions are associated with a shift in the droplet distribution towards larger droplets. Below is a figure showing the liquid water content, effective radius, and number concentration for the three horizontal legs shown in Figure 5 of our manuscript.

- b. *Changes to the manuscript:* Figure 5, Lines 697-701: We replaced Figure 5 with the version above that provides additional information on liquid water content and number concentration for three representative profiles. We also included an explanation for the sharp increase in effective radius shown in the blue curve above.
- 11. L. 484: What is the exact definition of "time difference" here? I guess the vertical profiles were sampled over some time as well, so when did MODIS pass over the scene and between which times were the profiles measured? Please clarify in the manuscript.
  - a. *Authors' Response:* For each vertically sampled in situ measurement with a start and end time, we used the temporal halfway point to compute the time difference with the MODIS measurement. After reading the reviewer's comment, we revisited this calculation and learned that we could estimate the time difference more precisely by using the MODIS metadata to estimate the time each MODIS pixel within a swath recorded its measurement (thanks to Dr. Larry DiGiorlamo and Dr. Guangyu Zhao). We have updated the time differences using this more precise calculation in the table above (comment 9).
  - b. *Changes to the manuscript*: Lines 759-762: We included a description of how the time difference between a MODIS measurement and the corresponding VOCALS-REx in-situ measurement was defined.

- 12. L. 39: "and has been used to verify ..."
  - a. Authors' Response: Thanks for finding this mistake!
  - b. *Changes to the manuscript:* Line 50: We fixed this mistake.
- 13. L. 270: "scalar"
  - a. Authors' Response: Thanks for finding this mistake!
  - b. Changes to the manuscript: Line 362: We fixed this mistake.
- 14. L. 357: "shown"
  - a. Authors' Response: Thanks for finding this mistake!
  - b. *Changes to the manuscript*: Line 562: We fixed this mistake.


scale the y-axis value Jacobians without changing the MODIS or CPF lines. If that's the case, I don't know how to interpret the results (e.g., if dr\_bottom is effectively zero, all bars will be zero on the y-axis). If not the case, please elaborate.

a. Authors' Response: The y-axis is absolute reflectance, and indeed the y-axis values would change for different values of  $r_{bot}$  and  $\Delta r_{bot}$ . The y-axis is the change in reflectance due to a small perturbation in the radius at cloud bottom. For the  $i^{th}$  iteration and the  $j^{th}$  spectral channel, we estimate the y-axis values using the following equation:

$$\Delta R\left(\boldsymbol{x}_{i}, \boldsymbol{\lambda}_{j}\right) = R\left((r_{i}^{top}, r_{i}^{bot} + \Delta r_{i}^{bot}, \boldsymbol{\tau}_{c_{i}}), \boldsymbol{\lambda}_{j}\right) - R\left((r_{i}^{top}, r_{i}^{bot}, \boldsymbol{\tau}_{c_{i}}), \boldsymbol{\lambda}_{j}\right)$$

The measurement uncertainties shown in Figure 4 for MODIS and CPF are also displayed in absolute reflectance. We used the reported radiometric uncertainties for the MODIS and CPF instruments and multiplied these percentages with the original reflectance,  $R((r_i^{top}, r_i^{bot}, \tau_{c_i}), \lambda_j)$ , for each spectral channel. Therefore, these curves would also change for different values of  $r_{bot}$ .

With all that said, we acknowledge the reviewer's point. The figure may be more useful if shown in relative values. The measurement uncertainties for MODIS and CPF are typically reported as a percentage. Therefore, it may be more useful to readers to show the percent change in reflectance along the y-axis. This has the added benefit of normalizing the change in reflectance with the initial reflectance, which depends on the current value of the radius at cloud bottom.

The following provides more details on how Figure 4 was created. We sought to use a representative cloud example, defined as having the median droplet profile found in Figure 1, except with varying optical depth.  $\Delta r_{bot}$  is defined as  $\Delta r_{bot} = 0.35 \ r_{bot}$  (L253). As described in the paper, this value was chosen so that the Jacobian terms with respect to the radius at cloud bottom,  $\frac{\Delta R_{\lambda}}{\Delta r_{bot}}$ , would not be dwarfed by the measurement uncertainty. However, we had to strike a balance between estimating the Jacobian, defined as the rate of change of reflectance with respect to some infinitesimal change in  $r_{bot}$ , with the measurement uncertainty. We found if  $\Delta r_{bot}$  was too small, measurement uncertainty dominated. If  $\Delta r_{bot}$  was too large, we no longer accurately estimated the local slope. We determined  $\Delta r_{bot} = 0.35 \ r_{bot}$  through trial and error. The phrasing in our manuscript may be misleading because we cannot always guarantee our estimate of the Jacobian exceeds the measurement uncertainty, because reflectance depends on the state vector. We decided to use a single value that worked for a broad set of state vectors.

- b. *Changes to the manuscript:* Lines 637-640, 653-658: We added additional information on Figure 4 that more thoroughly explains how the y-axis values and the measurement uncertainty, which are reported as absolute reflectances, are computed. We also noted the sensitive nature of reflectance with the state vector.
- 2. While the text mentions the spectral dependence of the Jacobians, it's not clear which channel(s) are being used in the figure.
  - a. *Authors' Response*: The channels used are listed by their center wavelengths along the x-axis of the plot. All seven channels used in the multi-spectral retrieval (Table 1) are shown. Note that they are not in sequential order according to channel number but rather increasing in the value of the center wavelength.
  - b. *Changes to the manuscript:* None.
- 3. (3) The y-axis for the Jacobians should be labeled delta reflectance, delta reflectance/drbottom, or something similar unless I'm mistaken about (1).
  - c. Authors' Response: Yes, we agree.
  - d. Changes to the manuscript: Figure 4: The y-axis label has been changed to  $\Delta Reflectance$ .

**Minor Comments**

- 1. L34: "effective droplet radius" or "effective droplet absorption" is proportional to 1-ssa? While it's true that  $kr_e \sim 1 ssa$  for an absorbing wavelength, it's an ill-defined definition for  $r_e$  when ssa = unity (i.e., reduces to  $r_e = 0/0$ ).
  - a. *Authors' Response:* We thank the reviewer for pointing out this mistake. Our intention was to highlight the relationship between the effective radius and the fractional absorption of incident light due to multiple scattering within warm clouds. We will correct this sentence.

- b. *Changes to the manuscript:* Lines 44-45: This sentence has been updated to: "The fraction of incident light absorbed by optically thick warm clouds is proportional to the effective droplet radius over the solar spectrum."
- 2. L49, 60: While Twomey and Cocks (1982) provides a nice overview of the retrieval theory, a more focused retrieval study was done in the follow-up Twomey and Cocks (1989, Beitr. Phys. Atmosph.), which used 5 spectral channels simultaneously in the retrieval (not bispectral) and presented the solution space in terms of residual contour plots similar to your Figs. 7, 8. I'm not suggesting you include the following relevant historic τ, re retrievals references but just for awareness: Other airborne retrievals (Foot (1988), Rawlins and Foot (1990)); AVHRR (Arking and Childs, 1985), Platnick and Twomey (1994).
  - a. *Authors' Response:* We thank the reviewer for suggesting the study by Twomey and Cocks (1989) as an early example of a multispectral retrieval of effective radius and optical depth. While the paper was hard to track down, it has proved insightful. We will review all suggested papers for potential incorporation into the historical section of our manuscript.
  - b. *Changes to the manuscript*: Lines 61-63: The introduction has been updated to include the results of Twomey and Cocks (1989) and Rawlins and Foot (1990) as relevant historical studies. Twomey and Cocks (1989) is cited several other times throughout the paper (Lines 177, 389, 664).
- 3. L60: suggest adding the qualifier "nearly independent from one another for optically thicker clouds ..."
  - a. Authors' Response: We agree.
  - b. *Changes to the manuscript:* Line 73: The sentence has been updated to: "Reflectances in these two spectral regions are nearly independent from one another, especially for clouds with an optical thickness greater than about ten;...".
- 4. L64: "... radius, cloud optical depth, and various surface spectral reflectance assumptions."
  - a. Authors' Response: We appreciate the suggestion.

- b. *Changes to the manuscript:* Lines 85-88: The sentence now reads: "This bispectral method is employed to compute the MODIS Collection 6 cloud products by computing extensive lookup tables of cloud reflectance with varying solar and viewing geometry, effective cloud droplet radius, cloud optical depth and various surface spectral reflectance assumptions (Amarasinghe et al., 2017)."
- 5. L123, Sect. 4.2: As a simulation, it doesn't make a difference for present purposes, but I'm curious why the simulations were done for EMIT spectra instead of CPF, which is mentioned prominently as the motivation for the study (including the abstract). Was it in anticipation of doing EMIT retrievals as a follow-on? It would be useful to explain the rationale.
  - a. Authors' Response: Multiple reviewers asked a similar question, so we reached out to the instrument team that developed HySICS (the HyperSpectral Imager for Climate Science), the spectral instrument on board CPF, and asked if we could obtain the spectral response functions so that we could simulate CPF-sampled TOA reflectance.
  - b. *Changes to the manuscript:* Section 4.2, Figures 8 and 9: We were given access to the HySICS spectral response functions and we have updated the analysis in section 4.2 such that the TOA reflectance spectra used to generate the contour plots in Figures 8 and 9 simulated the HySICS spectral channels. We no longer use simulated EMIT measurements.
- 6. L184: What effective variance ( $\nu_e$ ) is used? The alpha "width parameter" is mentioned on L294 but would be helpful to put it in terms of  $\nu_e$ . Are the same value(s) used for all 100 layers?
  - a. Authors' Response: The relationship between the alpha parameter and the effective variance is defined by Emde et al. (2016):  $\alpha = \frac{1}{v_{eff}} 3$ . We will include this definition in the paper and report the alpha values in the more familiar effective variance form. All 100 layers use the same effective variance. This parameter could vary with the vertical dimension in future iterations of our forward model.

- b. *Changes to the manuscript:* Lines 255-257: We used the effective variance to define the width of the distribution and provided the relationship between libRadtran's alpha width parameter and the effective variance.
- 7. L188, 193: Eq. 5 is an approximation, though a reasonably good one, for the retrieved re since an exact weighting function is confounded by multiple scattering. I.e., suggest "represents the approximate retrieved ..."
  - a. Authors' Response: Yes, we agree.
  - b. *Changes to the manuscript*: Line 261: This sentence has been updated to the following: "The wavelength-dependent column-weighted retrieved effective radius is approximated by:"
- 8. L91: A nice summary of the previous work. Platnick (2000) also did an information content study for MODIS-like imager, including the effect of calibration uncertainty, to help understand the number of independent parameters that can be retrieved for vertical profile inversions. Hard to make apple-to-apple comparisons but do your results seem somewhat consistent? Similar question with respect Fig. 8 accuracy sensitivity.
  - a. Authors' Response: We thank the reviewer for bringing up this question. The information content study in Platnick (2000) highlights several key points relevant to our work that should be discussed in our manuscript. The number of independent pieces of information that can be retrieved to determine a droplet profile is, at most, equal to the number of spectral channels used. In the analysis by Platnick (2000), the three retrievals of effective radius using near-infrared wavelengths of 1.6  $\mu m$ , 2.1  $\mu m$  and 3.7  $\mu m$  were found to provide only two pieces of information. The reason is that the difference between the retrieved  $r_{1.6\mu m}$  and  $r_{2.1\mu m}$  is less than the retrieval uncertainties for each.

We expect the retrieval uncertainty to be the same or less than that assumed by Platnick (2000) because of the increased number of spectral channels. Therefore, our results appear in line with those of Platnick (2000) because we are only retrieving two pieces of information, the effective radius at cloud top and bottom, which was deemed possible with just three wavelengths by Platnick (2000). However, we have not explicitly computed the minimum eigenvalue of the scaled covariance matrix.

- b. *Changes to the manuscript:* Lines 104-110: The historical background within the introduction has been expanded to include the information content results from Platnick (2000).
- 9. Fig. 1: Please try to add some contrast to the line plot colors as some are hard to distinguish (esp. for color blind readers).
  - a. Authors' Response: We will do so.
  - b. *Changes to the manuscript:* Figure 1: The colors of the different curves in Figure 1 have been updated to increase contrast and readability for color-blind readers using the Coblis color blindness simulator. In addition, we checked all figures using the Coblis simulator and made color adjustments as needed.
- 10. L253, 254: Good idea.
  - a. Authors' Response: Thanks!
- 11. L295: The MODIS retrieval wouldn't correspond exactly to the upper boundary re according to Fig. 1. Likely a small difference but worth a comment.
  - a. *Authors' Response:* We agree with this clarification. The retrieval of effective radius is representative of droplet size below but near the cloud top. We will clarify this in the manuscript.
  - b. *Changes to the manuscript:* Lines 409-411: The manuscript has been updated to the following: "We used the MODIS retrieval of cloud top height to define the upper boundary of the cloud, but this value is likely to be imperfectly aligned with the cloud top effective radius that we retrieved due to retrieval uncertainties in both."
- 12. L361/Sect. 4.1: For further context on the confounding effects that uncertainties in situ probes have on retrieval validation, including sampling issues associated with vertical and horizontal heterogeneity, I suggest looking at the recent Meyer et al. ORACLES study (amt.copernicus.org/articles/18/981/2025/). The paper discusses airborne spectral retrievals compared against two in situ cloud probes (CAS, PDI) having different measurement approaches in addition to some retrieval forward model errors. Retrieval

evaluation with airborne probes continues to be an inherently challenging problem for the community. Nice discussion here and in Sect. 4.1.

- a. Authors' Response: We thank the reviewer for sharing this paper. The nuanced discussion on comparing remote retrievals with in-situ observations focused on different aspects than our own discussion, and we will include it in our manuscript. Consistent with other findings, Meyer et al. (2025) found remote retrievals of cloud effective radius to be larger, on average, than the coincident insitu derived effective radii. This study attempted to reduce the differences between remote retrievals and in situ measurements by adjusting the complex index of refraction for liquid water and the effective variance of the droplet distribution within the forward radiative transfer model. This paper also has a great overview on the difficult nature of comparing remote retrievals and in-situ measurements and we will incorporate these results into our discussion in section 4.1.
- b. *Changes to the manuscript:* Lines 406-409, 668-674, 734-737, 1054-1056: The results of Meyer et al. (2025) were included as a recent example of the difficult nature of comparing remote retrievals with in situ observations. We also discussed the efforts by Meyer et al. (2025) to reduce the difference between remote retrievals and in situ measurements by tuning forward model parameters.
- 13. L377: Not sure that the cloud-top re retrievals "validate" use of the 2.1  $\mu$ m MODIS bispectral retrieval as a prior as much as demonstrates consistency with its use as a prior. I.e., much of the upper cloud re information content is coming from the 2.1  $\mu$ m channel, regardless of which algorithm is used.
  - a. Authors' Response: We agree with the reviewer's point. We included the  $2.1\mu m$  MODIS bispectral retrieval of effective radius in Figure 3 to demonstrate that this value was consistently found to be near the in-situ derived values at cloud top. This result shows it is a suitable choice for the a priori at cloud top. This is not 'validation' in the technical sense, and we will adjust our phrasing accordingly.
  - b. *Changes to the manuscript:* Lines 590-591: The manuscript has been updated to the following: "The bispectral retrieval of effective radius was within range of the cloud top in situ measurement for each case, demonstrating consistency with its use as the a priori value for the radius at cloud top."

- 14. Fig. 3a and 3c have the same MODIS retrieval values (blue dashed lines). One must be incorrect.
  - a. Authors' Response: Thanks for catching this!
  - b. Changes to the manuscript: Figure 3a has been corrected.
- 15. L409: I think this often gets lost on those who use gradient searches as part of inversion algorithms, especially in higher dimensional spaces. So, good to make this point, as obvious as it may seem. Is there an example solution contour plot associated with Fig. 3 that you could show to illustrate this point (i.e., similar to Figs. 7, 8)?
  - a. Authors' Response: We thank the review for the suggestion. Figures 7 and 8 show the  $\ell^2$ -norm of the difference between the forward modeled reflectances and the measurements in two-dimensional space. In actuality, this 'residual space' occupies three dimensions, with a residual associated with each point in the  $r_{top}$   $r_{bot}$ ,  $\tau_c$  space. There is a region within this residual space that meets our convergence requirements (the area within the isopleths of 1 in Figures 7 and 8 of our manuscript). We found the gradient to be large outside the convergence region, but once inside, the gradient was quite small. Even if we allow the iterations to continue within the isopleth of one, the slopes are so small that the algorithm quickly converges at one of the local minima.
  - b. *Changes to the manuscript:* None. We explored several options for depicting the higher-dimensional solution space, but we were not satisfied with any of them. In our follow-up study, which explores retrieving droplet profiles with the entire HySICS sampled spectrum, we will continue to explore ways to view the higher-dimensional solution space.
- 16. L440: suggest "... approximately 1 km2". The effective pixel shape in the across track direction suffers from the finite integration time and so has a ~2 km triangular wide spatial weighting function for most MODIS channels though a bit less so for "1 km" channels aggregated from the native 250 m (bands 1, 2) and 500 m (bands 3-7) detector arrays. That said, L462 is correct that the across track sampling is 1 km.
  - a. Authors' Response: Thank you for this clarification!

- b. *Changes to the manuscript:* Line 676: The adverb "approximately" was incorporated as suggested.
- 17. L446: Interesting number. Thanks for making the calculation.
- 18. Fig. 5 caption, L454, 455, and later text/captions.: Constant altitude flight lines aren't usually considered a "profile" in airborne sampling vernacular (at least in the cloud and aerosol community). Also, elsewhere in the manuscript profiles is used, without qualification, to describer vertical sizes only so it will be a source of confusion. Try "horizontal legs" or just "legs". I realize that constant altitude across three different clouds during the campaign may end up sampling different depths relative to cloud top and so have some vertical profile information (e.g., the yellow curve in Fig. 5).
  - a. *Authors' Response:* We agree with this point and want to ensure readers have a clear distinction between vertically and horizontally sampled in-situ measurements.
  - b. *Changes to the manuscript:* Figure 5 caption, Figure 6 caption, Lines 694, 696 and elsewhere: The manuscript has been updated so that the term 'profile' is used only when referring to vertical in-situ measurements or a retrieved vertical profile. The term 'horizontal leg' is used exclusively for in-situ measurements within clouds at a near-constant altitude.
- 19. L458: "... and 6 μm (yellow)"
  - a. Authors' Response: Thanks for catching this!
  - b. *Changes to the manuscript:* Line 697: The manuscript has been updated so that the units are before the parenthetical descriptor.
- 20. Figs. 7, 8: Nice demonstration of more channels v. better accuracy, with the latter being the only way to dramatically reduce the delta radius solution space uncertainty. That's an important result. (1) Initially, I didn't notice that the y-axis had both positive and negative values. Would be helpful to add a horizontal line to the zero value so readers can quickly appreciate that a large region of the space is outside the constraint. Or add a slight shading to the negative regions. (2) Add a point on the plots to indicate the modeled

cloud optical depth and delta effective radius that was used in the simulation (didn't see it mentioned in the text, nor the cloud top effective radius).

- a. Authors' Response: We agree that both suggestions provide useful information to readers. As the reviewer correctly noticed, the left panel of Figure 7 shows that some of the state vectors within the isopleth of one (the convergence region) have values where  $r_{top} r_{bot} < 0$ , which is outside the constraint we defined.
- b. *Changes to the manuscript:* Figures 8 and 9: A shading has been added that highlights the negative regions of each plot. We will also add points to indicate the true state vector used in our simulation.
- 21. Data Availability: If MODIS L2 cloud data was used, please also mention that these files were obtained from LAADS. I strongly suggest providing a doi for both the L1B and L2 files, which should be available on the LAADS product information pages.
  - a. Authors' Response: We agree.
  - b. *Changes to the manuscript:* Lines 1125-1130: The Data Availability section was updated to include DOI's for the MODIS L1B, L2 and geolocation files. We also cited LAADS as the source of all MODIS data used in our analysis.

  - a. *Authors' Response:* We agree that a discussion on horizontal inhomogeneity is needed. Instead of including the RGB images of each MODIS scene, we will

report the inhomogeneity index, along with the latitude and longitude of the measurements, within a new table that provides information on the MODIS and associated VOCALS-REx measurements used in our analysis. Zhang and Platnick (2011) showed that retrievals of effective radius are biased from 3-D radiative effects such as illumination and shadowing when the cloud under observation has an inhomogeneity index greater than 0.3. This results in a significant difference between retrieved effective radii using  $2.1\mu m$  and  $3.7\mu m$  shortwave infrared measurements. For the three cases used in our manuscript to retrieve droplet profiles (Figure 3), all had an inhomogeneity index of less than 0.1. According to Zhang and Platnick (2011), these values represent fairly homogeneous clouds and 3-D radiative effects are expected to be insignificant.

b. *Changes to the manuscript:* Table 2: We included a new table that outlines the time of each MODIS observation, the start and end times of the overlapping VOCALS-REx measurements, the time difference, the geographic location of the MODIS observations, and the sub-pixel inhomogeneity index.

|        | MODIS                  | MODIS        | MODIS Sub-         | VOCALS-     | VOCALS-     | Time       |
|--------|------------------------|--------------|--------------------|-------------|-------------|------------|
| Figure | Observation time (UTC) | Observation  | pixel              | REx in-situ | REx in-situ | difference |
|        |                        | latitude and | inhomogeneity      | start time  | end time    | (min)      |
|        | time (OTC)             | longitude    | index $H_{\sigma}$ | (UTC)       | (UTC)       |            |
| 3a     | Nov 11 2008            | -24.0986,    | 0.09               | 18:45:20    | 18:45:50    | 8.88       |
|        | 18:54:28               | -75.0013     |                    |             |             |            |
| 3b     | Nov 11 2008            | -22.8188,    | 0.07               | 14:40:59    | 14:41:38    | 1.18       |
|        | 14:42:29               | -73.0008     |                    |             |             |            |
| 3c     | Nov 9 2008             | -22.8970,    | 0.08               | 14:33:33    | 14:34:23    | 3.62       |
|        | 14:30:20               | -73.0036     |                    |             |             |            |

- 11. Figure 3: I recommend also showing the other effective radius  $(r_{e,1.6})$  retrieved using 1.6  $\mu$ m instead of 2.1  $\mu$ m  $(r_{e,2.1})$ , which is included in MOD06, in Figure 3.  $r_{e,1.6}$  may be able to sense the cloud particle size in a deeper depth than  $r_{e,2.1}$ .
  - a. Authors' Response: We agree that for warm, non-precipitating, adiabatic marine stratus clouds,  $r_{e,1.6}$  should be smaller than  $r_{e,2.1}$  because photons at 1.6  $\mu m$  have deeper average penetration depths due to a larger single scattering albedo than photons at 2.1  $\mu m$  (Platnick, 2000). However, we showed  $r_{e,2.1}$  along with our multi-spectral retrieval because this value was used for the a priori value of  $r_{top}$ , and for estimating the a priori value of  $r_{bot}$ .
  - b. Changes to the manuscript: None.

- 12. L401-405: To investigate why the case in Figure 3b performs worse than the other two, have you considered conducting a remote sensing simulation using the VOCALS-REx insitu measurements? That is, simulating MODIS reflectance measurements using the droplet size distribution obtained from the VOCALS-REx as input, and then retrieving  $\tau_c$ ,  $r_{top}$ , and  $r_{bot}$  using your algorithm.
  - a. *Authors' Response:* We thank the reviewer for this suggestion. We will investigate our solution to the case shown in Figure 3b by using the VOCALS-REx in-situ measurements to simulate MODIS TOA reflectances. However, as Figure 7 shows, the number of possible state vectors that lead to convergence is large when using the first seven MODIS spectral channels. With a large solution space (the area within the isopleth of one), the a priori guess strongly influences the final state vector because the iterative Gauss-Newton technique pushes each state vector along the direction of greatest change. The solution shown in Figure 3.b may suffer from a more inaccurate prior than the other two cases.
  - b. *Changes to the manuscript:* None. There were no major changes to the results using the VOCALS-REx in situ measured droplet profile to define the forward model cloud microphysics.
- 13. Section 4.2: Why are the EMIT specifications and wavelengths used in the simulation instead of CPF?
  - a. *Authors' Response:* The EMIT spectral response functions are freely available. Thus, we can simulate TOA reflectance spectra without any guesswork. We used EMIT measurements as a surrogate for CPF measurements because they have a similar spectral range and resolution. However, multiple reviewers asked a similar question, so we reached out to the instrument team that developed HySICS (the HyperSpectral Imager for Climate Science), the spectral instrument on board CPF, and asked if we could obtain the spectral response functions so that we could simulate CPF-sampled TOA reflectance.
  - b. *Changes to the manuscript:* Section 4.2, Figures 8 and 9: We were given access to the HySICS spectral response functions and we have updated the analysis in section 4.2 such that the TOA reflectance spectra used to generate the contour plots in Figures 8 and 9 simulated the HySICS spectral channels. We no longer use simulated EMIT measurements.

- 14. Figure 7: Has it been discussed why this contour pattern appears, particularly why the uncertainties of  $\tau_c$  and  $r_{top} r_{bot}$  exhibit a negatively correlated pattern?
  - a. Authors' Response: For this project, we did not investigate the reasons for the particular contour pattern found in Figure 7. In addition, we would argue that the retrieval uncertainty does not exhibit a negative correlation. Our iterative method terminates when the relative  $\ell^2$ -norm difference between the forward modeled reflectances and the MODIS observations is less than one, the innermost contour in Figure 7. The retrieval uncertainty of optical depth is the width of this contour along the x-axis, and the uncertainty in  $r_{top}^* r_{bot}$  is the width along the y-axis. A negative correlation would imply that retrieval uncertainty of  $\tau_c$  decreases as the uncertainty of  $r_{top}^* r_{bot}$  grows, which does not appear to be true. Figure 7 suggests that many values for the radius at cloud bottom will lead to convergence. In addition, we can conclude that the vector normal to the contours of Figure 7, the direction of greatest change, consistently had a larger component along the  $\tau_c$  axis than the  $r_{top}^* r_{bot}$ .
  - b. Changes to the manuscript: None.
- 15. L509: Please list the wavelengths of the 35 spectral channels used. It would be even better if they were presented along with the transmittance of atmospheric gases.
  - a. *Authors' Response*: We agree that this would be helpful for readers.
  - b. *Changes to the manuscript:* Figure 7: We added a plot of simulated TOA reflectance over the spectral range of the CPF instrument and overlaid the 35 spectral channels used in section 4.2 of our analysis.
- 16. L529-531, Sect. 4.2: Is assuming a radiometric uncertainty of 0.3% still reasonable, even when considering potential uncertainty in forward calculation, including uncertainties in given parameters such as gas absorption, surface albedo, and aerosols? Additionally, is this 0.3% uncertainty fairly defined in comparison to the 2% uncertainty of MODIS L1B?
  - a. *Authors' Response:* We acknowledge the lack of discussion on different sources of retrieval uncertainty and agree that they should be discussed. See our response to comment 11 from reviewer 1.
  - b. *Changes to the manuscript:* Lines 312-314, 445-451, 541-544, 789-794, 923-986, 1050-1059: We updated section 3 to include a description on sources of

forward model uncertainty, following previous work by Poulsen et al. (2012). In section 4.2, we adjusted the uncertainty added to the simulated TOA reflectance spectra to include both measurement and forward model uncertainty. Instead of explicitly estimating the uncertainty of each source within the forward model, we leveraged previous work by Poulsen et al. (2012) to provide reasonable estimates for the fraction of the total uncertainty due to forward model uncertainty. We also emphasized that forward model uncertainty can never be reduced entirely. Additionally, we compared our multispectral retrieval uncertainty estimate using simulated CPF TOA reflectances with the MODIS collection 6 cloud products retrieval uncertainty.

[revised manuscript text omitted]